# Effective Probabilistic Time Series Forecasting with Fourier Adaptive Noise-Separated Diffusion

## Abstract

Existing diffusion-based time series forecasting methods often target on mixed temporal patterns or undifferentiated residuals, limiting the potential of distinct temporal components. In this paper, we propose the **F**ourier **A**daptive **L**ite **D**iffusion **A**rchitecture (**FALDA**), a novel probabilistic framework for time series forecasting. FALDA leverages Fourier-based decomposition to incorporate a component-specific architecture, enabling tailored modeling of individual temporal components. A conditional diffusion model is utilized to estimate the future noise term, while our proposed lightweight denoiser, DEMA (Decomposition MLP with AdaLN), conditions on the historical noise term to enhance denoising performance. Grounded in rigorous mathematical proof, we introduce the Diffusion Model for Residual Regression (DMRR), a framework which methodologically unifies diffusion-based probabilistic regression method and theoretically demonstrate that FALDA effectively reduces epistemic uncertainty, allowing probabilistic learning to primarily focus on aleatoric uncertainty through further probabilistic analysis. Experiments on six real-world benchmarks demonstrate that FALDA consistently outperforms existing probabilistic forecasting approaches across most datasets for long-term time series forecasting while achieving enhanced computational efficiency without compromising accuracy. Notably, FALDA also achieves superior overall performance compared to state-of-the-art (SOTA) point forecasting approaches, with improvements of up to 9%. The code will be made publicly available.

## 1 Introduction

Time series forecasting (TSF) is crucial for decision-making systems in domains like finance (Li et al., 2020), healthcare (Festag & Spreckelsen, 2023), and transportation (Lv et al., 2014; Dai et al., 2020). Recent developments in deep learning have yielded various effective approaches for TSF (Wu et al., 2021; Zeng et al., 2023; Liu et al., 2024). These deterministic models process historical time series data to generate future predictions and exhibit strong capabilities in point forecasting tasks.

Diffusion models have demonstrated significant success across various generative tasks, including image generation (Esser et al., 2024; Rombach et al., 2022; Peebles & Xie, 2023; Chu et al., 2024; Liu et al., 2023; Lan et al., 2025; Ramesh et al., 2021; Labs, 2024; Chu et al., 2025) and video generation (Zhang et al., 2024; 2025; Zheng et al., 2024; Bar-Tal et al., 2024; Hu, 2024; Blattmann et al., 2023; Yang et al., 2024b; Lin et al., 2024). However, when applied to probabilistic time series forecasting, the progressive noise injection mechanism of diffusion models tends to disrupt the inherent temporal structures. Consequently, many previous works (Fan et al., 2024; Tashiro et al., 2021; Shen et al., 2024) that attempt to reconstruct complete temporal patterns (encompassing seasonality, trend, and noise) from pure noise often achieve inferior point estimation accuracy compared to deterministic models. This challenge becomes especially pronounced when handling non-stationary time series, as their statistical properties (e.g., mean, variance, autocorrelation, etc.) evolve over time (Yang et al., 2024a; Yuan & Qiao, 2024; Liu et al., 2022b; Ye et al., 2024).

Recent studies have explored hybrid approaches combining point estimation with diffusion models. TMDM (Li et al., 2024b) incorporates predictions from point estimation models into both forward and backward diffusion processes to enhance future predictions. $D^3U$ (Li et al., 2025) attempts to

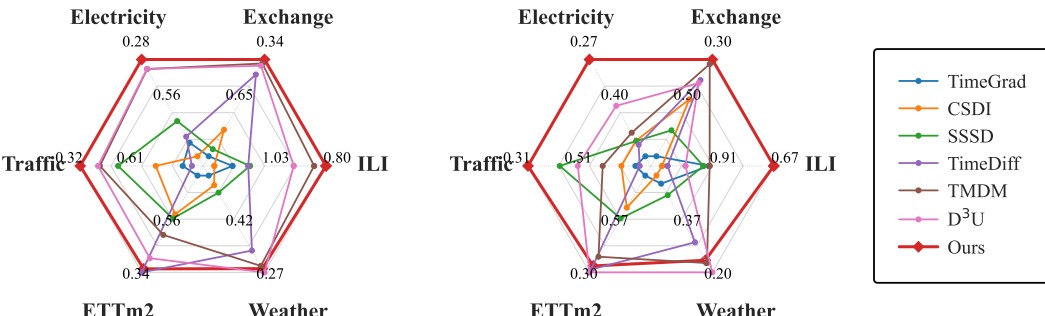

Figure 1: Performance of FALDA in point estimation (MAE, left) and probabilistic prediction (CRPS, right). All three plug-and-play methods (TMDM, D³U, and FALDA) utilize NSformer as the same backbone network for fair comparison.

decouple deterministic and uncertainty learning by leveraging embedded representations from point estimation to guide the diffusion model in capturing residual patterns, thereby avoiding the need to reconstruct complete temporal components through diffusion. Although demonstrating superior point estimation capability compared to previous diffusion-based approaches (Tashiro et al., 2021; Rasul et al., 2021), these approaches (1) are limited by their generic architecture designs, lacking explicit inductive biases to capture distinctive temporal structures, such as non-stationary patterns, and (2) employ diffusion to model an undifferentiated residual, which inevitably entangles **epistemic uncertainty** (from the limited capability of the guide model) with **aleatoric uncertainty** (inherent data noise) (Hüllermeier & Waegeman, 2021). As a result, these architectures fail to fully exploit the potential of different temporal components, undermining their capacity to contribute to further accuracy gains, particularly when integrated with strong backbone models.

In this paper, we first analyze the decoupling mechanisms for deterministic and uncertain components in Li et al. (2024b; 2025); Ho et al. (2020); Han et al. (2022), and introduce a unified generalized diffusion learning framework called DMRR (Diffusion Model for Residual Regression). Building on DMRR, we develop FALDA, a novel diffusion-based time series forecasting framework that employs Fourier decomposition to decouple time series into three distinct components: non-stationary trends, stationary patterns, and noise patterns. Through tailored modeling of each component, FALDA effectively separates epistemic uncertainty and aleatoric uncertainty (Gawlikowski et al., 2023), allowing the probabilistic modeling component to focus exclusively on aleatoric uncertainty. A lightweight denoiser DEMA is designed to handle multi-scale residuals. As a non-autoregressive diffusion model, FALDA avoids the common issue of error accumulation and demonstrates superior performance in long-range prediction tasks. Unlike conventional approaches that predict diffusion noise (Li et al., 2024b; Tashiro et al., 2021), our denoiser directly constructs the target series, thereby reducing the learning complexity for temporal patterns (Shen & Kwok, 2023). By integrating DDIM (Song et al., 2021) and DEMA, FALDA achieves both training and sampling efficiency. As illustrated in Figure 1, FALDA outperforms existing methods in both point estimation and probabilistic forecasting.

In summary, our main contributions are:

- We propose the Fourier Adaptive Lite Diffusion Architecture (FALDA), a diffusion-based probabilistic time series forecasting framework that leverages Fourier decomposition to decouple and model different time-series components. We design DEMA (Decomposition MLP with AdaLN), a lightweight denoiser that integrates adaptive layer normalization and trend-seasonality decomposition to handle multi-scale residuals. Combined with DDIM, DEMA improves computational efficiency while maintaining performance.

- We introduce the Diffusion Model for Residual Regression (DMRR), a theoretical framework that methodologically unifies diffusion-based probabilistic regression methods. DMRR not only establishes the equivalence of their underlying mechanisms, but also provides a solid theoretical foundation for FALDA's uncertainty modeling capabilities.

- FALDA supports plug-and-play deployment through a phase-adaptive training schedule, enabling seamless integration (e.g., processing the stationary term with SOTA deterministic

models). We evaluate our model on six real-world datasets, and the results demonstrate that FALDA achieves superior overall performance on both point forecasting and probabilistic forecasting.

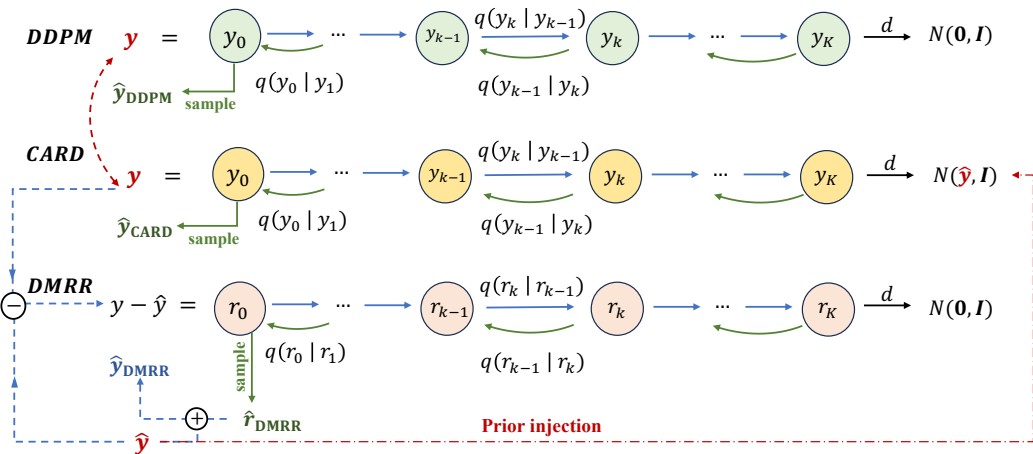

Figure 2: Comparison of three diffusion frameworks: DDPM, CARD, and DMRR, where $\hat{y}_{\text{DDPM}}$, $\hat{y}_{\text{CARD}}$, and $\hat{y}_{\text{DMRR}}$ represent their respective final estimates.

## 2 DIFFUSION MODEL FOR RESIDUAL REGRESSION (DMRR)

Diffusion models are increasingly applied to probabilistic regression, including TSF. While some recent probabilistic regression methods have demonstrated strong performances (Han et al., 2022; Li et al., 2024b; 2025), they inherently conform to a unified framework that refines residual errors through Denoising Diffusion Probabilistic Models (DDPM) (Ho et al., 2020). In this work, we term this framework Diffusion Model for Residual Regression (DMRR). This section begins with a formal review of the Classification and Regression Diffusion (CARD) (Han et al., 2022), which establishes a generalized framework extending DDPM, where DDPM can be viewed as a special case with zero prior knowledge. Through the lens of the DMRR framework, we subsequently demonstrate that CARD essentially applies standard DDPM to perform residual fitting, establishing a conceptual unification across these seemingly disparate approaches (Li et al., 2024b; 2025).

**CARD** CARD extends Denoising Diffusion Probabilistic Models (DDPM) by incorporating prior knowledge into both forward and reverse diffusion processes (see Appendix A.1 for DDPM fundamentals). Formally, given a target variable $y_0 \sim q(y)$ with covariate $x$, CARD utilizes prior knowledge $f_\phi(x)$ to guide the generation, where $f_\phi$ can be a pretrained network as demonstrated in Han et al. (2022). This yields the following forward diffusion process:

$$
\begin{aligned}
y_k &= \sqrt{\alpha_k} y_{k-1} + (1 - \sqrt{1 - \beta_k}) f_\phi(x) + \sqrt{\beta_k} z_k, \quad z_k \sim \mathcal{N}(0,1), \quad \text{(one-step)} \\
y_k &= \sqrt{\bar{\alpha}_k} y_0 + (1 - \sqrt{\bar{\alpha}_k}) f_\phi(x) + \sqrt{1 - \bar{\alpha}_k} \bar{z}_k, \quad \bar{z}_k \sim \mathcal{N}(0,1), \quad \text{(multi-step)}
\end{aligned}
\tag{1}
$$

where $\alpha_k = 1 - \beta_k \in (0,1)$ and $\bar{\alpha}_k = \prod_{s=1}^{k} \alpha_s$ denote the noise schedule parameters for $k = 1, 2, \ldots, K$. This process converges to a Gaussian limit distribution: $\mathcal{N}(f_\phi(x), I)$. The corresponding reverse process posterior distribution is given by:

$$
q(y_{k-1}|y_k, y_0) = \mathcal{N}(y_{k-1}; \tilde{m}_k, \tilde{\beta}_k I), \text{ where}
$$

$$
\tilde{m}_k = \frac{\beta_k \sqrt{\bar{\alpha}_{k-1}}}{1 - \bar{\alpha}_k} y_0 + \frac{(1 - \bar{\alpha}_{k-1})\sqrt{\alpha_k}}{1 - \bar{\alpha}_k} y_k + (1 + \frac{(\sqrt{\bar{\alpha}_k} - 1)(\sqrt{\alpha_k} + \sqrt{\bar{\alpha}_{k-1}})}{1 - \bar{\alpha}_k}) f_\phi(x), \tag{2}
$$

$$
\tilde{\beta}_k = \frac{1 - \bar{\alpha}_{k-1}}{1 - \bar{\alpha}_k} \beta_k.
$$

The residual $l_k = y_k - f_\phi(x)$ exhibits the same convergence behavior as DDPM, with a standard Gaussian distribution as its limit distribution. This equivalence underpins our DMRR framework, which systematically formalizes this residual learning paradigm within a unified diffusion framework.

**The unified framework**  As illustrated in Figure 2, our proposed DMRR framework introduces a residual learning paradigm that decouples prior knowledge from the limit distribution in CARD diffusion process. Given the target $y$, the framework first generates a preliminary estimate $\hat{y}$ (for CARD $\hat{y} = f_\phi(x)$). Unlike CARD, which learns the full data distribution $y$ guided by $\hat{y}$, DMRR focuses on learning the residual distribution $q(r)$, where $r = y - \hat{y}$. This is implemented through a DDPM process, where the forward diffusion follows the Markov chain $\{r_0 = r, r_1, \ldots, r_k, \ldots\}$ with $r_k$ denoting the noise sample at step $k$. The reverse process generates residual predictions: $\hat{r}_{\text{DMRR}}$ via the denoising network. The final output, which can be considered as a refinement of the preliminary estimate $\hat{y}$, combines both components:

$$\hat{y}_{\text{DMRR}} = \hat{y} + \hat{r}_{\text{DMRR}}. \tag{3}$$

Mathematically, we prove that $l_k = y_k - \hat{y}$ in CARD and $r_k$ in DMRR possess identical conditional and posterior distributions (see Appendix A for rigorous proofs). And it should be noted that when the preliminary estimate $\hat{y} = 0$, CARD and DMRR degenerate to the standard DDPM.

In Section 3.3, we comprehensively discuss the diffusion framework underlying state-of-the-art TSF models. We further analyze how different framework designs affect the performance of time series prediction tasks and illustrate the advantages of DMRR framework in TSF tasks.

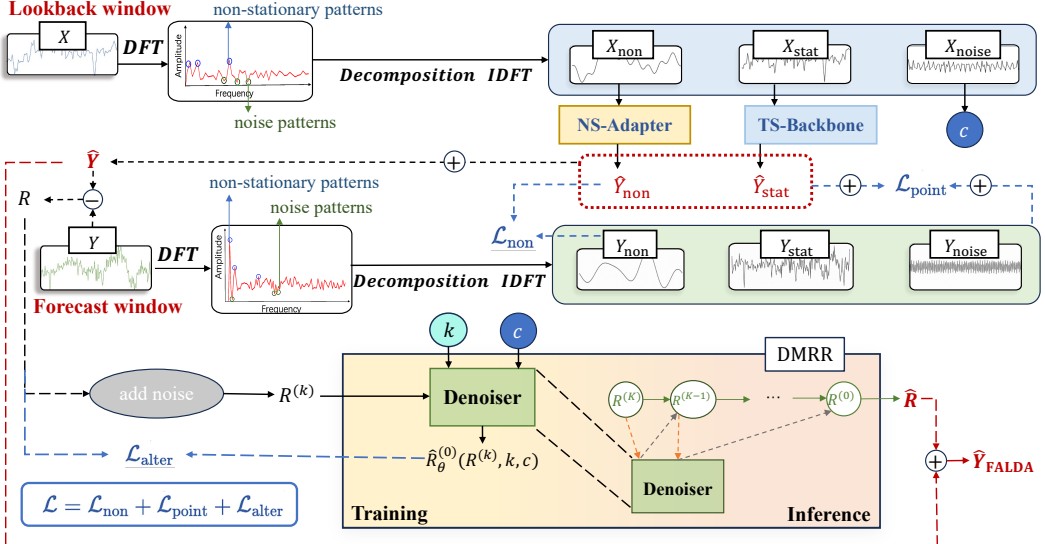

Figure 3: An illustration of the proposed FALDA framework. By leveraging Fourier decomposition, NS-Adapter and TS-Backbone generate the preliminary estimation, $\hat{Y}$. The prediction residual $R = Y - \hat{Y}$ is then input into the denoiser for subsequent probabilistic learning and refinement of the preliminary estimation.

# 3  FOURIER ADAPTIVE LITE DIFFUSION ARCHITECTURE (FALDA)

From a methodological perspective, probabilistic time series forecasting is a specialized form of probabilistic regression applied to temporal data, necessitating explicit modeling of sequential dependencies. Within the DMRR framework, we propose the Fourier Adaptive Lite Diffusion Architecture (FALDA), which leverages point-guided diffusion models for TSF while reducing the influence of non-stationarity and noise on probabilistic learning. We further analyze the underlying mechanism through a comparative discussion of diffusion-based TSF models.

### 3.1 PROBLEM STATEMENT

In the time series forecasting task, let $X = \{X^t\}_{t=1}^T \in \mathbb{R}^{T \times D}$ represent an observed multivariate time series with $T$ historical time steps, where each $X^t \in \mathbb{R}^D$ denotes the $D$-dimensional observation vector at time $t$. Given this lookback window $X$, the objective is to forecast the subsequent $S$ time steps, denoted as $Y = \{Y^t\}_{t=1}^S \in \mathbb{R}^{S \times D}$. Our approach specifically addresses real-world time series data where significant non-stationarity arises from trend and seasonality. Furthermore, these series typically possess an inherently sparse frequency domain, with energy concentrated in a few main frequencies.

### 3.2 MAIN FRAMEWORK

**FALDA**   As shown in Figure 3, the time series is first decomposed into three components: a **non-stationary term**, $Y_{\text{non}}$, representing temporal components that exhibit time-varying statistical properties; a **stationary term**, $Y_{\text{stat}}$, comprising components whose statistical properties remain invariant over time; and a **noise term**, $Y_{\text{noise}}$, reflecting inherent stochastic disturbances within the time series. Following Yuan & Qiao (2024); Ye et al. (2024), this decomposition is performed using the Fourier transform. Specifically, the non-stationary component is extracted by reconstructing the time series from the frequencies corresponding to the $K_1$ largest amplitudes, while the noise component is obtained by reconstructing the time series from the frequencies associated with the $K_2$ smallest amplitudes:

$$Y_{\text{non}} = \mathcal{F}^{-1}(\text{Top}(\mathcal{F}(Y), K_1)), \quad Y_{\text{noise}} = \mathcal{F}^{-1}(\text{Bottom}(\mathcal{F}(Y), K_2)). \tag{4}$$

Here, $\mathcal{F}$ denotes the Fourier transform and $\mathcal{F}^{-1}$ denotes the inverse Fourier transform. The operators $\text{Top}(\cdot, K_1)$ and $\text{Bottom}(\cdot, K_2)$ select the frequency components with the $K_1$ largest and the $K_2$ smallest amplitudes, respectively. Since different datasets exhibit varying levels of non-stationarity and inherent noise, the values of the hyperparameters $K_1$ and $K_2$ may change accordingly. An adaptive selection strategy is used to determine these hyperparameters. Specifically, $K_1$ is set as the average number of frequency components whose amplitudes exceed ratio $p_1$ of the maximum amplitude, while $K_2$ is determined as the average number of frequency components whose amplitudes are below ratio $p_2$ of the maximum amplitude. The ratios $p_1$ and $p_2$ are uniformly set across different datasets. For further details, please refer to Appendix E.5. The stationary term is defined as:

$$Y_{\text{stat}} = Y - Y_{\text{non}} - Y_{\text{noise}}. \tag{5}$$

Similarly, the decomposition for $X$ is given by $X = X_{\text{non}} + X_{\text{stat}} + X_{\text{noise}}$. Based on this decomposition, FALDA integrates three key components: (1) a non-stationary adapter (**NS-Adapter**) $f_w$, which models the non-stationary term $Y_{\text{non}}$ by addressing evolving temporal patterns and mitigating epistemic uncertainty; (2) a time series backbone (**TS-Backbone**) $g_\phi$, which captures temporally invariant patterns to model the stationary component $Y_{\text{stat}}$; (3) a conditional diffusion process with a lightweight denoiser $\hat{R}_\theta^{(0)}$, which specializes in handling aleatoric uncertainty by modeling the inherent noise component $Y_{\text{noise}}$ in the data. The predictions for the non-stationary and stationary components are given by:

$$\hat{Y}_{\text{non}} = f_w(X_{\text{non}}), \quad \hat{Y}_{\text{stat}} = g_\phi(X_{\text{stat}}). \tag{6}$$

Here $f_w$ is implemented as a multi-layer perceptron (MLP) to effectively capture non-stationary patterns, while $g_\phi$ serves as a flexible backbone that can be substituted by conventional point forecasting models. For further details on the implementation of $f_w$, please refer to Appendix E.3.

Eq. 6 gives a preliminary estimation $\hat{Y} = \hat{Y}_{\text{non}} + \hat{Y}_{\text{stat}}$. We use DDPM to model the residual component, which is defined as $R = Y - \hat{Y}$. During the reverse process, the posterior mean is parameterized as: $\tilde{\mu}_\theta(R^{(k)}, k) = \frac{\sqrt{\bar{\alpha}_{k-1}}\beta_k}{1-\bar{\alpha}_k}\hat{R}_\theta^{(0)}(R^{(k)}, k, c) + \frac{\sqrt{\alpha_k}(1-\bar{\alpha}_{k-1})}{1-\bar{\alpha}_k}R^{(k)}, k = K, K-1, ..., 1$. $R^{(k)}$ represents the noise sample at step $k$, and condition $c$ is set to the noise term of the lookback window, $X_{\text{noise}}$. The denoiser $\hat{R}_\theta^{(0)}(R^{(k)}, k, c)$ directly reconstructs the target $R = R^{(0)}$ instead of learning the diffusion noise at each step. This approach alleviates the learning difficulty of time series data (Shen & Kwok, 2023; Yuan & Qiao, 2024). An estimate of the residuals is generated through reverse sampling: $\hat{R}^{(K)} \rightarrow \hat{R}^{(K-1)} \rightarrow \cdots \rightarrow \hat{R}^{(0)} = \hat{R}$. The final output is the sum of the three component outputs in FALDA:

$$\hat{Y}_{\text{FALDA}} = \hat{Y}_{\text{non}} + \hat{Y}_{\text{stat}} + \hat{R}. \tag{7}$$

In alignment with the multi-component decomposition framework of FALDA, we propose a tailored loss function designed to facilitate multi-task optimization. To effectively capture non-stationary patterns, we define the non-stationary term loss $\mathcal{L}_{\text{non}}$ to provide prior guidance. Simultaneously, to ensure the overall accuracy of the preliminary point estimations, we define the overall point estimation loss $\mathcal{L}_{\text{point}}$. These two loss functions can be expressed as:

$$\mathcal{L}_{\text{non}} = \ell(Y_{\text{non}}, \hat{Y}_{\text{non}}), \quad \mathcal{L}_{\text{point}} = \ell(Y, \hat{Y}), \tag{8}$$

where $\ell$ is the $L_1$ loss. The alternative loss $\mathcal{L}_{\text{alter}}$ simultaneously optimizes the denoiser and fine-tunes the point estimate model through two terms:

$$\mathcal{L}_{\text{alter}} = \lambda_s \underbrace{\|\text{sg}(R) - \hat{R}_\theta^{(0)}(R^{(k)}, k, c)\|^2}_{\mathcal{L}_{\text{diffusion}}} + \eta_s \underbrace{\|R - \text{sg}(\hat{R}_\theta^{(0)}(R^{(k')}, k', c))\|^2}_{\mathcal{L}_{\text{finetune}}}. \tag{9}$$

Here, $R = Y - \hat{Y}$. The first term $\mathcal{L}_{\text{diffusion}}$ targets the optimization of the denoiser, where the stop-gradient operation $\text{sg}(\cdot)$ ensures no interference with the point estimate model's training. The second term $\mathcal{L}_{\text{finetune}}$ fine-tunes the point estimate models, improving them alongside the denoiser. Here, $k'$ is a hyperparameter that enables flexible selection of the diffusion step during the fine-tuning process. Additionally, two scheduling hyperparameters, $\lambda_s$ and $\eta_s$, are introduced to control the alternating optimization of the two losses in $\mathcal{L}_{\text{alter}}$. These parameters depend on the current training epoch $s$, and are governed by a threshold $\delta$ and a period $\Delta$:

$$\lambda_s = \begin{cases} 1, & s \geq \delta \text{ and } s \bmod \Delta \neq 0 \\ 0, & \text{otherwise} \end{cases}, \quad \eta_s = \begin{cases} 1, & s \geq \delta \text{ and } s \bmod \Delta = 0 \\ 0, & \text{otherwise} \end{cases}, \tag{10}$$

where the hyperparameter $\delta$ determines the pretraining duration (in epochs) for the point forecasting models, while $\Delta$ controls the alternating intervals between denoiser training and fine-tuning. The final loss function is given by:

$$\mathcal{L} = \mathcal{L}_{\text{non}} + \mathcal{L}_{\text{point}} + \mathcal{L}_{\text{alter}}. \tag{11}$$

For complete training and inference algorithm of FALDA, please refer to Appendix D.

**DEMA**  We design DEMA (Decomposition MLP with AdaLN), a lightweight denoiser denoted as $\hat{R}_\theta^{(0)}$, to effectively predict the future time series noise term $Y_{\text{noise}}$. As a conditional denoiser, $\hat{R}_\theta^{(0)}(\cdot)$ takes the $k$-step noise sample $R^{(k)} \in \mathbb{R}^{S \times D}$, the diffusion step $k$, and condition $c = X_{\text{noise}} \in \mathbb{R}^{T \times D}$ as input. The input $R^{(k)}$ and condition $c$ are projected into a latent space with dimension $H_d$ through the following embedding process:

$$h_k^{[0]} = \text{Linear}(R^{(k)}) \in \mathbb{R}^{H_d \times D}, \quad e_k = \text{Linear}(\text{PE}(k)) + \text{Linear}(c) \in \mathbb{R}^{H_d \times D}, \tag{12}$$

where $\text{PE}(\cdot)$ is sinusoidal embedding (Vaswani et al., 2017; Li et al., 2024a). The embedding $h_k$ and $e_k$ are then processed by an $L$-layer encoder. At each layer $l \in \{0, 1, ..., L-1\}$, the encoder performs the following computations:

$$\left[\tau_{\text{season}}^{[l]}, \tau_{\text{trend}}^{[l]}\right] = \left[h_k^{[l]} - \text{MA}_a(h_k^{[l]}), \text{MA}_a(h_k^{[l]})\right], \tag{13}$$

$$\left[\gamma_i^{[l]}, \beta_i^{[l]}, o_i^{[l]}\right] = \text{Linear}(\text{SiLU}(e_k)), \tag{14}$$

$$\bar{\tau}_i^{[l]} = (\gamma_i^{[l]} + 1) \odot \text{LayerNorm}(\tau_i^{[l]}) + \beta_i^{[l]}, \tag{15}$$

where $\gamma_i^{[l]}$, $\beta_i^{[l]}$ and $o_i^{[l]}$ represent the scale factor, shift factor, and gating factor, respectively, with $i \in \{\text{season}, \text{trend}\}$. $\text{MA}_a$ denotes the moving average operation with kernel size $a$. The output of an encoder layer is computed as:

$$h_k^{[l+1]} = h_k^{[l]} + (o_{\text{season}}^{[l]} + o_{\text{trend}}^{[l]}) \odot \text{Linear}(\bar{\tau}_{\text{season}}^{[l]} + \bar{\tau}_{\text{trend}}^{[l]}). \tag{16}$$

After processing through an adaptive layer normalization decoder, the denoiser generates its final output $\hat{R}_\theta^{(0)}(R^{(k)}, k, c) \in \mathbb{R}^{S \times D}$, where $\theta$ represents all trainable parameters in the network.

### 3.3 ANALYSIS OF DIFFERENT DIFFUSION-BASED TIME SERIES MODELS WITH RESIDUAL LEARNING

TMDM and $D^3U$ are representative diffusion-based time series forecasting models that incorporate residual learning. Specifically, TMDM employs CARD as its underlying diffusion mechanism, while $D^3U$ and FALDA utilize DMRR (see Appendix B for detailed mathematical formulations). As discussed in Section 2, DMRR and CARD share identical transition probabilities and posterior distributions, indicating that their stochastic dynamics are mathematically equivalent. Despite theoretical equivalence, DMRR offers crucial modeling advantages and is inherently more suitable for TSF tasks compared to CARD. Real-world time series typically consist of multiple components (trend, seasonality, and inherent noise) that are often corrupted during the diffusion process due to gradual noise addition. This corruption makes it challenging to recover the time series distribution from the noise data (Yuan & Qiao, 2024). Although the preliminary estimate partially captures temporal patterns, it remains difficult for CARD framework to learn the residual distribution from the noisy full time series $Y^{(k)}$, which also represents a limitation of TMDM. In contrast, $D^3U$ and FALDA, which are based on DMRR, alleviate this limitation through their residual learning paradigm. This paradigm explicitly decouples the preliminary estimation from the limiting distribution in CARD and focuses exclusively on modeling the residual between the preliminary estimate and the ground truth. The residual components encompass both epistemic and aleatoric uncertainties (Gawlikowski et al., 2023). While $D^3U$ demonstrates promising performance by utilizing latent representations from the encoder as the condition in the reverse process, its generalized modeling approach primarily captures epistemic uncertainty due to the lack of explicit consideration for distinct temporal components. This architectural characteristic limits its ability to explicitly model the pure underlying probability distribution, especially the aleatoric uncertainty component. Furthermore, this limitation may result in diminishing returns when applied to backbone models that already exhibit strong predictive capabilities. An elaborate analysis of this phenomenon is provided in Appendix C. Our framework extends this approach by introducing dedicated network architectures designed to capture three key temporal components. This enhanced modeling capability enables more balanced learning of both epistemic and aleatoric uncertainties, thereby contributing to improved point estimation accuracy.

Recent work (Ye et al., 2025) highlights non-stationary uncertainty modeling as a critical challenge. NsDiff (Ye et al., 2025) incorporates time-varying variance as prior knowledge into the diffusion forward process endpoint, enhancing non-stationary uncertainty capability compared to TMDM. However, during our experiments, we observed that FALDA inherently possesses a strong capability for capturing non-stationary uncertainty. As visualized in Figure 8, we compare the predictions of TMDM and FALDA on the Exchange dataset. While TMDM's predicted variance does not exhibit significant time-varying characteristics, FALDA's predicted variance increases over time, aligning with the characteristics of the Exchange dataset and demonstrating FALDA's high interpretability. Furthermore, a synthetic dataset provided by Ye et al. (2025) is employed to examine uncertainty behavior over long forecasting horizons. This specific dataset features a linearly increasing standard deviation. As visualized in Figure 7, FALDA effectively captures the distribution shift between the training and test datasets whereas TMDM's estimated variance deviates substantially from the true variance on the test set. The results suggest that FALDA possesses a strong capability for non-stationary uncertainty modeling. We attribute this advantage to the DMRR framework and our tailored modeling strategy. In TMDM, the diffusion process is tasked with simultaneously modeling deterministic non-stationary terms, time-varying uncertainty (such as time-varying variance), and other temporal components like stationary terms. This conflated objective hinders TMDM's ability to focus specifically on non-stationary uncertainty modeling. In contrast, FALDA explicitly models the deterministic non-stationary and stationary components with the NS-Adapter and TS-Backbone, allowing the diffusion process to focus on capturing the non-stationary uncertainty inherent in the data. Consequently, FALDA naturally yields a distinct advantage in modeling non-stationary uncertainty. More details are provided in Appendix H.1.

## 4 EXPERIMENTS

### 4.1 EXPERIMENT SETUP

Six widely recognized real-world datasets are utilized for evaluation: ILI, Exchange-Rate, ETTm2, Electricity, Traffic, and Weather. More details are provided in Appendix E.1. 13 state-of-the-art TSF

models are included in our baselines including both point forecasting and probabilistic forecasting methods: Informer (Zhou et al., 2021), Autoformer (Wu et al., 2021), FEDformer (Zhou et al., 2022), DLinear (Zeng et al., 2023), TimesNet (Wu et al., 2023), PatchTST (Nie et al., 2023), iTransformer (Liu et al., 2024), TimeGrad (Rasul et al., 2021), CSDI (Tashiro et al., 2021), SSSD (Alcaraz & Strodthoff, 2023), TimeDiff (Shen & Kwok, 2023), TMDM (Li et al., 2024b), $D^3U$ (Li et al., 2025).

We set the lookback window $T = 96$ and prediction length $S = 192$, except for ILI where $T = S = 36$. Following Ho et al. (2020), we use $K = 1000$ diffusion timesteps with a linear noise schedule. FALDA employs iTransformer as its default backbone if not stated otherwise, with DDIM (Song et al., 2021) for inference acceleration. Implementation details are fully provided in Appendix E.4.

Table 1: Comparison of MAE and MSE across six real-world datasets. **Bold** denotes the best-performing method for each metric-dataset combination, while underlined indicates the second-best.

| Methods | ILI | | Exchange | | Electricity | | Traffic | | ETTm2 | | Weather | |
|---|---|---|---|---|---|---|---|---|---|---|---|---|
| Metric | MSE | MAE | MSE | MAE | MSE | MAE | MSE | MAE | MSE | MAE | MSE | MAE |
| Informer | 4.620 | 1.456 | 1.092 | 0.853 | 0.319 | 0.399 | 0.696 | 0.379 | 0.494 | 0.525 | 0.598 | 0.544 |
| Autoformer | 3.366 | 1.210 | 0.537 | 0.526 | 0.227 | 0.332 | 0.616 | 0.382 | 0.269 | 0.327 | 0.276 | 0.336 |
| FEDformer | 2.679 | 1.163 | 0.276 | 0.384 | 0.198 | 0.312 | 0.606 | 0.377 | 0.269 | 0.325 | 0.276 | 0.336 |
| DLinear | 2.235 | 1.059 | 0.167 | 0.301 | 0.196 | 0.285 | 0.598 | 0.370 | 0.284 | 0.362 | 0.218 | 0.278 |
| TimesNet | 2.671 | 0.986 | 0.224 | 0.343 | 0.184 | 0.289 | 0.617 | 0.336 | 0.249 | 0.309 | 0.219 | 0.261 |
| PatchTST | 2.374 | 0.918 | 0.181 | 0.303 | 0.205 | 0.307 | 0.463 | 0.311 | 0.251 | 0.312 | 0.223 | 0.258 |
| iTransformer | 1.833 | 0.828 | 0.193 | 0.315 | 0.164 | 0.248 | 0.413 | 0.251 | 0.246 | **0.300** | 0.217 | **0.247** |
| TimeGrad | 2.644 | 1.142 | 2.429 | 0.902 | 0.645 | 0.723 | 0.932 | 0.807 | 1.385 | 0.732 | 0.885 | 0.551 |
| CSDI | 2.538 | 1.208 | 1.662 | 0.748 | 0.553 | 0.795 | 0.921 | 0.678 | 1.291 | 0.576 | 0.842 | 0.523 |
| SSSD | 2.521 | 1.079 | 0.897 | 0.861 | 0.481 | 0.607 | 0.794 | 0.498 | 0.973 | 0.559 | 0.693 | 0.501 |
| TimeDiff | 2.458 | 1.085 | 0.475 | 0.429 | 0.730 | 0.690 | 1.465 | 0.851 | 0.284 | 0.342 | 0.277 | 0.331 |
| TMDM | 1.985 | 0.846 | 0.260 | 0.365 | 0.222 | 0.329 | 0.721 | 0.411 | 0.524 | 0.493 | 0.244 | 0.286 |
| $D^3U$ | 2.103 | 0.935 | 0.254 | 0.358 | 0.179 | 0.267 | 0.468 | 0.299 | **0.241** | 0.302 | 0.222 | 0.264 |
| Ours | **1.666** | **0.821** | **0.165** | **0.296** | **0.163** | **0.248** | **0.412** | **0.251** | 0.246 | 0.301 | **0.215** | 0.255 |

## 4.2 MAIN RESULT

**Forecasting performance and computational efficiency**  We conduct a comprehensive evaluation of the proposed model against state-of-the-art baselines for four metrics: CRPS, CRPS$_{sum}$, MAE, and MSE. CRPS and CRPS$_{sum}$ assess the probabilistic forecasting performance, while MAE and MSE evaluate the point forecasting accuracy. See Appendix E.2 for detailed metric descriptions. Table 1 summarizes MAE and MSE results across six real-world datasets. Our method outperforms all baselines in four out of six datasets (ILI, Exchange, Electricity, and Traffic) for both MAE and MSE. On the remaining two datasets, our method consistently ranks among the top two performers. The most significant improvement is observed on the ILI dataset, where our model achieves a notable 9% reduction in MSE compared to iTransformer, the second-best model, demonstrating FALDA's powerful ability in point forecasting. FALDA also presents superior or comparable probabilistic forecasting performance compared to previous diffusion-based models. Table 2 shows the CRPS and CRPS$_{sum}$ metrics across 6 datasets. On Exchange, FALDA promotes an average of 9% on CRPS and 39% on CRPS$_{sum}$. In terms of efficiency, FALDA achieves an inference speed-up of up to $26.3\times$ and a training speed-up of up to $13.7\times$ compared to TMDM, as detailed in Appendix F.6.

**Plug-and-play performance**  To evaluate the generality of our framework, we integrate four well-known point forecasting models into the FALDA framework: Autoformer (Wu et al., 2021), Informer

Table 2: Comparison of CRPS and CRPS$_{sum}$ across six real-world datasets. **Bold** denotes the best-performing method for each metric-dataset combination, while underlined indicates the second-best.

| Methods | ILI | | Exchange | | ETTm2 | | Weather | | Electricity | | Traffic | |
|---|---|---|---|---|---|---|---|---|---|---|---|---|
| Metric | CRPS | CRPS$_{sum}$ | CRPS | CRPS$_{sum}$ | CRPS | CRPS$_{sum}$ | CRPS | CRPS$_{sum}$ | CRPS | CRPS$_{sum}$ | CRPS | CRPS$_{sum}$ |
| TimeGrad | 0.924 | 0.527 | 0.661 | 0.437 | 0.785 | 1.051 | 0.482 | 0.503 | 0.503 | 1.452 | 0.657 | 1.683 |
| CSDI | 1.104 | 0.607 | 0.448 | 0.469 | 0.625 | 0.782 | 0.508 | 0.465 | 0.465 | 0.823 | 0.612 | 1.275 |
| SSSD | 0.945 | 0.548 | 0.564 | 0.370 | 0.571 | 0.275 | 0.445 | 0.442 | 0.466 | 0.580 | 0.414 | 0.949 |
| TimeDiff | 1.083 | 0.610 | 0.376 | 0.275 | 0.316 | 0.180 | 0.293 | 0.400 | 0.475 | 0.594 | 0.671 | 0.823 |
| TMDM | 0.921 | 0.524 | 0.316 | 0.209 | 0.380 | 0.226 | 0.226 | 0.292 | 0.446 | **0.137** | 0.552 | 0.179 |
| D$^3$U | 0.951 | 0.566 | 0.318 | 0.210 | **0.243** | 0.141 | 0.207 | **0.283** | **0.202** | 0.160 | **0.232** | 0.186 |
| Ours | **0.721** | **0.387** | **0.289** | **0.126** | 0.244 | **0.141** | **0.207** | 0.298 | 0.231 | 0.160 | 0.245 | **0.163** |

(Zhou et al., 2021), Transformer (Vaswani et al., 2017), and iTransformer (Liu et al., 2024). Table 3 shows their performance improvements with FALDA. Results show consistent improvements in both MSE and MAE metrics across the majority of evaluated datasets. The most significant improvements are observed for Informer, which achieves maximum reductions of 66.4% in MSE and 46.2% in MAE on the same dataset. For iTransformer, which serves as a strong baseline model, FALDA still provides measurable improvements (e.g., 14.6% MSE reduction on Exchange) while maintaining competitive performance across other datasets. Notably, D$^3$U exhibits performance degradation when using iTransformer as the backbone, as evidenced in Tables 1 and 6 of Li et al. (2025). These results validate FALDA's effectiveness in enhancing forecasting performance for both relatively weaker backbones and state-of-the-art backbones, demonstrating its general applicability in TSF tasks.

Table 3: Plug-and-play performance improvement of FALDA on existing point forecasting methods. Better values are highlighted in **bold**.

| Model | Exchange | | ILI | | ETTm2 | | Electricity | |
|---|---|---|---|---|---|---|---|---|
| Metric | MSE | MAE | MSE | MAE | MSE | MAE | MSE | MAE |
| Autoformer | 0.537 | 0.526 | 3.366 | 1.210 | 0.269 | 0.327 | 0.227 | 0.332 |
| + ours | **0.232** | **0.351** | **2.655** | **1.118** | **0.247** | **0.313** | **0.209** | **0.316** |
| Promotion | **56.7%** | **33.3%** | **21.1%** | **7.5%** | **8.2%** | **4.2%** | **7.6%** | **4.7%** |
| Informer | 1.092 | 0.853 | 4.620 | 1.456 | 0.494 | 0.525 | 0.319 | 0.399 |
| + ours | **0.367** | **0.460** | **3.122** | **1.178** | **0.293** | **0.363** | **0.305** | **0.388** |
| Promotion | **66.4%** | **46.2%** | **32.4%** | **19.1%** | **40.8%** | **30.9%** | **4.5%** | **2.8%** |
| Transformer | 0.975 | 0.765 | 4.044 | 1.327 | 0.427 | 0.472 | 0.256 | 0.347 |
| + ours | **0.403** | **0.488** | **3.226** | **1.254** | **0.390** | **0.423** | **0.251** | **0.344** |
| Promotion | **58.7%** | **36.3%** | **20.2%** | **5.5%** | **8.7%** | **10.2%** | **1.8%** | **0.9%** |
| iTransformer | 0.193 | 0.315 | 1.833 | 0.828 | 0.246 | **0.300** | 0.164 | 0.248 |
| + ours | **0.165** | **0.296** | **1.666** | **0.821** | 0.246 | 0.301 | **0.163** | 0.248 |
| Promotion | **14.6%** | **6.0%** | **9.1%** | **0.8%** | **0.1%** | -0.5% | **1.1%** | **0.0%** |

Table 4: Ablation study on different condition strategies. The best results are boldfaced.

| Condition | Exchange | | | | ILI | | | | ETTm2 | | | | Weather | | | |
|---|---|---|---|---|---|---|---|---|---|---|---|---|---|---|---|---|
| | MSE | MAE | CRPS | CRPS$_{sum}$ | MSE | MAE | CRPS | CRPS$_{sum}$ | MSE | MAE | CRPS | CRPS$_{sum}$ | MSE | MAE | CRPS | CRPS$_{sum}$ |
| $X_{noise}$ | **0.165** | **0.296** | 0.289 | **0.126** | **1.666** | 0.821 | 0.721 | 0.387 | **0.246** | **0.301** | **0.244** | **0.141** | **0.215** | **0.255** | **0.207** | 0.298 |
| uncond | 0.184 | 0.311 | 0.264 | 0.141 | 1.675 | **0.785** | **0.677** | **0.342** | 0.251 | 0.307 | 0.265 | 0.153 | 0.217 | 0.260 | 0.212 | 0.314 |
| $X$ | 0.178 | 0.312 | **0.262** | 0.131 | 1.994 | 0.966 | 0.908 | 0.485 | 0.258 | 0.313 | 0.258 | 0.152 | 0.216 | 0.261 | 0.213 | 0.314 |

## 4.3 ABLATION STUDY

To further validate that our architecture enables the diffusion model to focus on aleatoric uncertainty learning, we investigate the model's performance under different conditioning strategies. Table 4 compares the results when using $X_{\text{noise}}$, $X$ as conditioning inputs, along with an unconditional case. The experiments show that the $X_{\text{noise}}$-conditioned version achieves optimal performance across all evaluated datasets, while the unconditional case performs comparably to the $X_{\text{noise}}$-conditioned scenario. In contrast, the $X$-conditioned approach shows the worst performance among the three conditioning types. These results indicate that epistemic uncertainty does not dominate the components of diffusion learning, thereby the residual estimation through $X$-conditioning provides limited benefits. In conclusion, the FALDA framework successfully achieves enhanced learning of aleatoric uncertainty while simultaneously improving point estimation capability. Additionally, Appendix F.1 presents an ablation study comparing DEMA with its variants, systematically validating the effectiveness of its time-decomposition operation. Appendix F.5 shows the impact of different fine-tuning strategies during training. Appendix F.2 demonstrates the effectiveness of the DMRR component and the NS-Adapter module. Figure 1 shows the advantage of our framework when using the same NSformer (Liu et al., 2022b) backbone. The complete experimental results are provided in Appendix F.3.

## 5 RELATED WORKS

Rasul et al. (2021) integrates RNN with a diffusion model for autoregressive forecasting, using hidden states to condition the diffusion process. Its autoregressive nature causes error accumulation and inefficiency in long-term forecasting. Tashiro et al. (2021) adopts a non-autoregressive fashion which uses self-supervised masking to guide the denoising process, with historical information and observation as conditions. Shen & Kwok (2023) introduces inductive bias to the outputs of the conditioning network through two mechanisms (future mixup and autoregressive initialization) to facilitate the denoising process. Yuan & Qiao (2024) combines seasonal-trend decomposition techniques with the design of the denoiser, which is expected to generate interpretable samples. Li et al. (2024b) and Li et al. (2025) guide the diffusion process with strong point forecasting models, enhancing point forecasting and probabilistic forecasting capability. Ye et al. (2025) integrates time-varying variances into the endpoint of the diffusion process, using this prior knowledge to enhance the model's ability to capture non-stationary uncertainty.

## 6 CONCLUSION

In this paper, we present **FALDA**, a Fourier-based diffusion framework for time series forecasting that systematically addresses both deterministic patterns and stochastic uncertainties. Our Fourier decomposition and component-specific modeling approach enable FALDA to decouple complex time series into interpretable components while clearly separating epistemic and aleatoric uncertainty. The integration of a conditional diffusion model with historical noise conditioning significantly improves stochastic component prediction, achieving enhanced computational efficiency. Our methodological DMRR framework and theoretical analysis provide formal guarantees for the mathematical foundations of FALDA. Extensive empirical evaluations across six diverse real-world datasets consistently demonstrate FALDA's strong performance on both point forecasting and probabilistic forecasting.

**Ethics Statement**   We have used publicly available datasets for training and evaluation, ensuring compliance with all relevant data usage policies and privacy regulations. Our paper is anonymous, with no personally identifiable information included in the main text or the supplementary materials. We advocate for the responsible use of our technology to prevent misuse in contexts that could infringe on privacy or other ethical standards. All methodologies and experiments have been conducted with integrity, adhering to established research ethics guidelines.

**Reproducibility Statement**   We have taken comprehensive measures to ensure the reproducibility of our results. Full implementation details are provided in Appendix E.4. The complete algorithmic descriptions are included in Appendix D. All mathematical derivations and proofs are furnished in Appendix A. We will release all source code pending release approval.

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

# A  MATHEMATICAL DERIVATIONS

## A.1  PRELIMINARY: DENOISING DIFFUSION PROBABILISTIC MODELS

Denoising Diffusion Probabilistic Models (DDPM) (Ho et al., 2020) is a canonical diffusion model consisting of the forward and reverse processes. Let $q(y_0)$ be the data distribution, the forward process is a Markov chain $\{y_0, y_1, ..., y_k, ...\}$ that gradually transforms the data distribution into a standard Gaussian distribution: $y_k \xrightarrow{d} \mathcal{N}(0, I), k \rightarrow \infty$. Here "$\xrightarrow{d}$" denotes convergence in distribution. The transition probability is $q(y_k|y_{k-1}) = \mathcal{N}(y_k; \sqrt{\alpha_k}y_{k-1}, \beta_k I)$. where $\alpha_k = 1 - \beta_k \in (0, 1)$ represents the noise schedule. The single-step transition formulation at step $k$ can be demonstrated as below using the reparameterization trick (Kingma & Welling, 2014):

$$y_k = \sqrt{\alpha_k}y_{k-1} + \sqrt{\beta_k}z_k, \quad z_k \sim \mathcal{N}(0, I). \tag{17}$$

Iterating the single-step formulation leads to the multi-step transition formulation at step $k$:

$$y_k = \sqrt{\bar{\alpha}_k}y_0 + \sqrt{1 - \bar{\alpha}_k}\bar{z}_k, \quad \bar{z}_k \sim \mathcal{N}(0, I). \tag{18}$$

Here $\bar{\alpha}_k = \prod_{s=1}^{k} \alpha_s, \bar{\beta}_k = \prod_{s=1}^{k} \beta_s$. The reverse process starts from a standard Gaussian noise $y_K$, and has the following posterior distribution at step $k$:

$$q(y_{k-1}|y_k, y_0) = \mathcal{N}(y_{k-1}; \tilde{\mu}_k, \tilde{\beta}_k I),$$
$$\tilde{\mu}_k = \frac{\sqrt{\bar{\alpha}_{k-1}}\beta_k}{1 - \bar{\alpha}_k}y_0 + \frac{\sqrt{\alpha_k}(1 - \bar{\alpha}_{k-1})}{1 - \bar{\alpha}_k}y_k, \quad \tilde{\beta}_k = \frac{1 - \bar{\alpha}_{k-1}}{1 - \bar{\alpha}_k}\beta_k. \tag{19}$$

By substituting $y_0$ with $y_0 = \frac{1}{\sqrt{\bar{\alpha}_k}}y_k - \frac{\sqrt{1-\bar{\alpha}_k}}{\sqrt{\bar{\alpha}_k}}\bar{z}_k$, we have $\tilde{\mu}(y_k, k) = \frac{1}{\sqrt{\alpha_k}}\left(y_k - \frac{\beta_k}{\sqrt{1-\bar{\alpha}_k}}\bar{z}_k\right)$. The mean $\tilde{\mu}_k$ is typically parameterized using two different strategies: (1) modeling the diffusion noise $\tilde{z}_k$ with $\hat{\epsilon}_\theta(y_k, k)$, or (2) directly parameterizing the target $y_0$ in Eq. 19 with $\hat{y}_\theta(y_k, k)$.

## A.2 EQUIVALENCE BETWEEN CARD AND DMRR

**Proposition 1.** *Let $y_k$ be the Markov chain defined in Eq. 1. Let $l_k = y_k - f_\phi(x)$, we have:*

$$q(l_k|l_{k-1}) = \mathcal{N}(l_k; \sqrt{\alpha_k}l_{k-1}, \beta_k I) \tag{20}$$

*and*

$$q(l_{k-1}|l_k, l_0) = \mathcal{N}(y_{l-1}; \tilde{\mu}_k, \tilde{\beta}_k I),$$
$$\tilde{\mu}_k = \frac{\sqrt{\bar{\alpha}_{k-1}}\beta_k}{1 - \bar{\alpha}_k}l_0 + \frac{\sqrt{\alpha_k}(1 - \bar{\alpha}_{k-1})}{1 - \bar{\alpha}_k}l_k, \quad \tilde{\beta}_k = \frac{1 - \bar{\alpha}_{k-1}}{1 - \bar{\alpha}_k}\beta_k. \tag{21}$$

*Thus, the residual process $l_t$ exhibits identical Markovian dynamics to the standard DDPM framework in both forward and reverse processes as shown in Eq. 18 and Eq. 19.*

*Proof.* **Proof of Equation 20:**
Starting from the result in Eq. 1,

$$l_k = y_k - f_\phi(x)$$
$$= \sqrt{\bar{\alpha}_k}y_0 + (1 - \sqrt{\bar{\alpha}_k})f_\phi(x) + \sqrt{1 - \bar{\alpha}_k}\bar{z}_k - f_\phi(x)$$
$$= \sqrt{\bar{\alpha}_k}(y_0 - f_\phi(x)) + \sqrt{1 - \bar{\alpha}_k}\bar{z}_k + f_\phi(x) - f_\phi(x)$$
$$= \sqrt{\bar{\alpha}_k}l_0 + \sqrt{1 - \bar{\alpha}_k}\bar{z}_k.$$

This demonstrates that $l_t$ satisfies the standard DDPM forward process formulation.

**Proof of Equation 21:**
since $l_k = y_k - f_\phi(x)$ and $q(y_{k-1}|y_k, y_0) = \mathcal{N}(y_{k-1}; \tilde{m}_k, \tilde{\beta}_k I)$, we have:

$$q(l_{k-1}|l_k, l_0) = \mathcal{N}(l_{k-1}, \tilde{m}_k - f_\phi(x), \tilde{\beta}_k I).$$

We now analyze the mean $\tilde{m}_k - f_\phi(x)$. With the definition of $\tilde{m}_k$ in Eq. 2, we have:
$$\tilde{m}_k - f_\phi(x) = A_k y_0 + B_k y_k + (C_k - 1)f_\phi(x),$$

where the coefficients are:
$$A_k := \frac{\beta_k \sqrt{\bar{\alpha}_{k-1}}}{1 - \bar{\alpha}_k}, \quad B_k := \frac{(1 - \bar{\alpha}_{k-1})\sqrt{\alpha_k}}{1 - \bar{\alpha}_k},$$
$$C_k := 1 + \frac{(\sqrt{\bar{\alpha}_k} - 1)(\sqrt{\alpha_k} + \sqrt{\bar{\alpha}_{k-1}})}{1 - \bar{\alpha}_k}.$$

Substituting $y_k = l_k + f_\phi(x)$ yields:
$$\tilde{m}_k - f_\phi(x) = A_k l_0 + B_k l_k + (A_k + B_k + C_k - 1)f_\phi(x).$$

In the following step, the coefficients of $f_\phi(x)$ can be expanded as:

$$A_k + B_k + C_k - 1 = \frac{\beta_k\sqrt{\bar{\alpha}_{k-1}} + (1 - \bar{\alpha}_{k-1})\sqrt{\alpha_k}}{1 - \bar{\alpha}_k} + \frac{(\sqrt{\bar{\alpha}_k} - 1)(\sqrt{\alpha_k} + \sqrt{\bar{\alpha}_{k-1}})}{1 - \bar{\alpha}_k}$$
$$= \frac{\sqrt{\bar{\alpha}_{k-1}} - \alpha_k\sqrt{\bar{\alpha}_{k-1}} - \sqrt{\alpha_k}\bar{\alpha}_{k-1} + \sqrt{\alpha_k\bar{\alpha}_k} + \sqrt{\bar{\alpha}_k\bar{\alpha}}_{k-1} - \sqrt{\bar{\alpha}_{k-1}}}{1 - \bar{\alpha}_k}$$
$$= \frac{-\alpha_k\sqrt{\bar{\alpha}_{k-1}} - \sqrt{\alpha_k}\bar{\alpha}_{k-1} + \sqrt{\alpha_k\bar{\alpha}_k} + \sqrt{\bar{\alpha}_k\bar{\alpha}_{k-1}}}{1 - \bar{\alpha}_k}.$$

Using the identity $\bar{\alpha}_k = \bar{\alpha}_{k-1}\alpha_k$, we have:
$$A_k + B_k + C_k - 1 = 0.$$

Therefore, the posterior mean $\tilde{m}_k - f_\phi(x)$ satisfies:
$$\tilde{m}_k - f_\phi(x) = \frac{\beta_k\sqrt{\bar{\alpha}_{k-1}}}{1 - \bar{\alpha}_k}l_0 + \frac{(1 - \bar{\alpha}_{k-1})\sqrt{\alpha_k}}{1 - \bar{\alpha}_k}l_k$$
$$= \tilde{\mu}_k.$$

We have thus established that the reverse distribution of the residual process satisfies: $q(l_{k-1}|l_k, l_0) = \mathcal{N}(l_{k-1}; \tilde{\mu}_k, \tilde{\beta}_k I)$, This completes the proof of Eq. 21. $\square$

## B    METHODOLOGY OF TMDM AND $D^3U$

In this section, we present the details of two previously developed diffusion-based time series forecasting methods: Transformer-Modulated Diffusion Model (TMDM) (Li et al., 2024b) and Diffusion-based Decoupled Deterministic and Uncertain framework ($D^3U$) (Li et al., 2025). The notation employed below is consistent with the notation used in Section 3.1.

### B.1    TMDM

TMDM employs CARD as its underlying diffusion framework. Given a conditional information $\hat{Y}$, the end point of TMDM's diffusion process is:

$$\lim_{k \to \infty} q(Y^{(k)}|\hat{Y}) = \mathcal{N}(\hat{Y}, I). \tag{22}$$

Here $Y^{(k)}$ represents the noise sample of $Y$ at step $k$. With a noise schedule $\alpha_t$ and $\beta_t$ defined in Section 2, the forward process at step k can be defined as:

$$q\left(Y^{(k)}|Y^{(k-1)}, \hat{Y}\right) \sim \mathcal{N}\left(\sqrt{\alpha_k}Y^{(k-1)} + (1 - \sqrt{1-\beta_k})\hat{Y}, \beta_k I\right). \tag{23}$$

The posterior distribution in the reverse diffusion process is:

$$q\left(Y^{(k-1)}|Y^{(k)}, Y^{(0)}, \hat{Y}\right) \sim \mathcal{N}\left(Y^{(k-1)}; \tilde{m}_k, \tilde{\beta}_k I\right), \tag{24}$$

where $\tilde{m}_k$ and $\tilde{\beta}_k$ are consistent with Eq. 2. Specifically, $\tilde{m}_k$ satisfies:

$$\tilde{m}_k = \frac{\beta_k \sqrt{\bar{\alpha}_{k-1}}}{1 - \bar{\alpha}_k}Y^{(0)} + \frac{(1 - \bar{\alpha}_{k-1})\sqrt{\alpha_k}}{1 - \bar{\alpha}_k}Y^{(k)} + (1 + \frac{(\sqrt{\bar{\alpha}_k} - 1)(\sqrt{\alpha_k} + \sqrt{\bar{\alpha}_{k-1}})}{1 - \bar{\alpha}_k})\hat{Y}. \tag{25}$$

### B.2    $D^3U$

The $D^3U$ framework builds upon the DMRR diffusion architecture. It employs a pretrained network $f_{D^3U}$ to generate preliminary estimates $\hat{Y}$, where the encoder embedding $f_{enc}(X)$ serves as the condition for the reverse diffusion process.

Defining the residual term $R = Y - \hat{Y}$, the forward diffusion process follows:

$$q\left(R^{(k)}|R^{(k-1)}, \hat{R}\right) \sim \mathcal{N}\left(\sqrt{\alpha_k}R^{(k-1)}, \beta_k I\right). \tag{26}$$

The posterior process is:

$$q\left(R^{(k-1)}|R^{(k)}, R^{(0)}, f_{enc}(X)\right) \sim \mathcal{N}\left(R^{(k-1)}; \tilde{\mu}_k, \tilde{\beta}_k I\right). \tag{27}$$

Here $\tilde{\mu}_k$ is consistent with Eq. 19:

$$\tilde{\mu}_k = \frac{\sqrt{\bar{\alpha}_{k-1}}\beta_k}{1 - \bar{\alpha}_k}R^{(0)} + \frac{\sqrt{\alpha_k}(1 - \bar{\alpha}_{k-1})}{1 - \bar{\alpha}_k}R^{(k)}. \tag{28}$$

## C    PROBABILITY VIEW OF RESIDUAL COMPONENT MODELING

As discussed in Section 3, $D^3U$ models epistemic uncertainty by conditioning on encoder outputs without intentionally decoupling it from temporal aleatoric uncertainty. This limits optimal performance scaling on more capable backbone models, which already exhibit low epistemic uncertainty. In this section, we provide a probabilistic analysis of different modeling approaches for time series forecasting. Specifically, Appendix C.1 summarizes the general case, while Appendices C.2 and C.3 respectively analyze the probabilistic modeling of $D^3U$ and FALDA, highlighting their distinct learning objectives. We demonstrate how FALDA models both types of uncertainty through time-series components decomposition, allowing both deterministic and probabilistic models to focus on learning their respective components.

### C.1 GENERAL SITUATION

In general, a time series $X$ can be decomposed into two components:

$$X = X_{\text{nf}} + \epsilon_X, \tag{29}$$

where $X_{\text{nf}}$ is the ideal noise-free part (incorporating trend, seasonality, and other structured patterns), and $\epsilon_X$ denotes the inherent zero-mean noise in the time series data. Notably, in real-world scenarios, $\epsilon_X$ often follows complex non-Gaussian distributions. This canonical decomposition naturally extends to the forecasting target: $Y = Y_{nf} + \epsilon_Y$. To simplify the notation, in the following paragraphs, the subscripts for the noises only indicate which components they are associated with. The goal of the time series forecasting task is then to learn the conditional distribution: $P(Y|X)$. Conventionally, a deterministic function $f$ is employed to estimate the posterior expectation:

$$E(Y|X) = E(Y_{nf}|X) + \mathbb{E}\left(\epsilon_Y|f_\phi(X)\right) = E(Y_{nf}|X) \approx f(X_{\text{nf}} + \epsilon_X). \tag{30}$$

This yields the following regression form for the prediction:

$$Y = f(X_{\text{nf}} + \epsilon_X) + \epsilon_{X,Y}. \tag{31}$$

In Equation 31, $\epsilon_{X,Y}$ comprises two distinct uncertainty components: aleatoric uncertainty stemming from inherent data randomness (specifically, the time series noise), and epistemic uncertainty arising from model estimation errors (Kendall & Gal, 2017).

Under ideal conditions where the point-estimation model perfectly captures $\mathbb{E}(Y|X)$, $\epsilon_{X,Y}$ would reduce to purely aleatoric uncertainty and become uncorrelated with $f(X)$, satisfying:

$$\mathbb{E}\left(\epsilon_{X,Y}|f_\phi(X)\right) = 0. \tag{32}$$

This implies the lookback window $X$ contains no additional information to improve point forecasts, resulting in $\epsilon_{X,Y} = \epsilon_Y$. However, in practice, point-estimation models rarely achieve this theoretical optimum, typically retaining some epistemic uncertainty. The subsequent discussion will examine how different time series forecasting models handle these distinct uncertainty components.

### C.2 D³U SITUATION

As established in Appendix B.2, the D³U framework leverages the encoder-derived embedding representation $f_{\text{enc}}(X)$ as a conditioning mechanism for probabilistic residual learning, subsequent to the preliminary estimation $f(X)$. Formulated within the regression expression in the previous section, this approach specifically targets the conditional expectation $E(\epsilon_{X,Y}|f_{\text{enc}}(X))$, yielding:

$$Y = f(X) + g(f_{\text{enc}}(X)) + \tilde{\epsilon}_{X,Y}. \tag{33}$$

In this context, $\tilde{\epsilon}_{X,Y}$ denotes the total uncertainty of D³U. Since the encoder of the point estimation model $f$ learns a good representation of the historical time series, $g(f_{\text{enc}}(X))$ can further model the epistemic uncertainty of $f(X)$. Comparing to $\epsilon_{X,Y}$, $\tilde{\epsilon}_{X,Y}$ may contain less epistemic uncertainty. However, due to the predominance of predictions with epistemic uncertainty, this facilitation may diminish when the backbone model is sufficiently powerful. More importantly, since the true probabilistic component, uncertainty, is not explicitly separated, diffusion models may focus on epistemic uncertainty rather than uncertainty. This undifferentiated treatment ultimately constrains their probabilistic learning capability.

### C.3 OUR SITUATION

To mitigate the epistemic uncertainty, first, we decompose the history time series into three parts $X = X_{\text{non}} + X_{\text{stat}} + X_{\text{noise}}$. Three models are jointly trained to forecast the whole future time series. Beyond the point-estimation model, we introduce an NS-adapter to improve modeling accuracy and reduce epistemic uncertainty, thereby alleviating part of the computational burden on the diffusion model. This architecture allows the diffusion model to concentrate solely on capturing aleatoric uncertainty, with the noise component $X_{\text{noise}}$ serving as the conditioning input for the diffusion process. The corresponding mathematical formulation is as follows:

$$Y = f_{\text{non}}(X_{\text{non}}) + f_{\text{stat}}(X_{\text{stat}}) + g_{\text{noise}}(X_{\text{noise}}) + \bar{\epsilon}_{X,Y}. \tag{34}$$

Under this formulation, $\bar{\epsilon}_{X,Y}$ contains more aleatoric uncertainty, since explicit component separation effectively mitigates epistemic uncertainty. Compared to the expression $g(f_{\text{enc}}(X)) + \tilde{\epsilon}_{X,Y}$ in Eq. 33, our approach shows superior properties. First, the composite term $g_{\text{noise}}(X_{\text{noise}}) + \bar{\epsilon}_{X,Y}$ is not dominated by epistemic uncertainty, since $f_{\text{non}}$ already takes into account most of the non-smooth patterns. Second, this decomposition allows the diffusion model to focus more effectively on capturing pure uncertainty without interference from the cognitive uncertainty component.

# D  ALGORITHMS

We formally present the complete algorithmic procedures of FALDA. Algorithm 1 details the end-to-end training protocol with multi-task optimization. The corresponding inference procedure is specified in Algorithm 2.

---

**Algorithm 1** FALDA Training Procedure

---

1: **Require**: TS-backbone $g_\phi$, NS-adapter $f_w$, denoiser $\hat{R}_\theta^{(0)}$
2: **Hyperparameters**: Threshold $\delta$, period $\Delta$, $k'$, noise schedule: $\alpha_t, \beta_t$, max diffusion step $K$
3: **Input**: Lookback window $X \in \mathbb{R}^{T \times D}$, future ground truth $Y \in \mathbb{R}^{S \times D}$
4: Initialize the parameteres
5: **repeat**
6:     **Decomposition via Fourier Transform**          ▷ Eq. equation 4, equation 5
7:     $X_{\text{non}}, X_{\text{stat}}, X_{\text{noise}} \leftarrow X$
8:     $Y_{\text{non}}, Y_{\text{stat}}, Y_{\text{noise}} \leftarrow Y$
9:     **Non-stationary & Stationary Components modeling:**
10:     $\hat{Y}_{\text{non}} \leftarrow f_w(X_{\text{non}})$                              ▷ Eq. equation 6
11:     $\hat{Y}_{\text{stat}} \leftarrow g_\phi(X_{\text{stat}})$
12:     **Residual Learning:**
13:     $R \leftarrow Y - \hat{Y}_{\text{non}} - \hat{Y}_{\text{stat}}$
14:     $k \sim \mathcal{U}(\{1, 2, ..., K\})$
15:     $\epsilon \sim \mathcal{N}(0, I)$
16:     $R^{(k)} \leftarrow \sqrt{\bar{\alpha}_k} R + \sqrt{1 - \bar{\alpha}_k}\epsilon, \; R^{(k')} \leftarrow \sqrt{\bar{\alpha}_{k'}} R + \sqrt{1 - \bar{\alpha}_{k'}}\epsilon,$
17:     Predict residual: $\hat{R}_\theta^{(0)}(R^{(k)}, k, X_{\text{noise}}), \hat{R}_\theta^{(0)}(R^{(k')}, k', X_{\text{noise}})$
18:     **Loss Computation:**
19:     Compute the loss $\mathcal{L}$ in Eq. equation 11
20:     Take gradient descent step on: $\nabla \mathcal{L}$
21: **until** converged

---

# E  EXPERIMENT DETAILS

## E.1  DATASETS

Experiments are performed on seven widely-used real-world time series datasets: (1) influenza-like illness (ILI) reports the weekly ratio of patients presenting influenza-like symptoms to total clinical visits, obtained from U.S. CDC surveillance data from 2002 to 2021. [1] (2) Exchange-Rate (Lai et al., 2018) provides daily currency exchange rates for eight countries from 1990 to 2016. [2] (3) ETTm2 and ETTm1 (Zhou et al., 2021) contains 7 factors of electricity transformer from July 2016 to July 2018, which is recorded by 15 minutes. [3] (4) Electricity (Li et al., 2019) collects hourly power consumption from 321 customers from 2012 through 2014. [4] (5) Traffic (Wu et al., 2023) collates hourly road occupancy rates measured by 862 sensors on San Francisco Bay Area freeways between January 2015 and December 2016. [5] (6) Weather (Zhou et al., 2021) includes meteorological time series

---

[1]ILI: https://gis.cdc.gov/grasp/fluview/fluportaldashboard.html
[2]Exchange: https://github.com/laiguokun/multivariate-time-series-data
[3]ETTm2: https://github.com/zhouhaoyi/ETDataset
[4]Electricity:https://archive.ics.uci.edu/dataset/321
[5]Traffic: https://zenodo.org/record/4656132

---

**Algorithm 2** FALDA Inference Procedure

---

1: **Require**: Pretrained TS-backbone $g_\phi$, NS-adapter $f_w$ and denoiser $\hat{R}_\theta^{(0)}$
2: **Input**: Lookback window $X \in \mathbb{R}^{T \times D}$
3: **Decomposition via Fourier Transform:**        ▷ Eq. equation 4, equation 5
4: $X_{\text{non}}, X_{\text{stat}}, X_{\text{noise}} \leftarrow X$
5: **Predict Non-stationary & Stationary Terms:**
6: $\hat{Y}_{\text{non}} \leftarrow f_w(X_{\text{non}})$
7: $\hat{Y}_{\text{stat}} \leftarrow g_\phi(X_{\text{stat}})$
8: **Generate Residual Prediction via Reverse Diffusion:**
9: Sample $R^{(K)} \sim \mathcal{N}(0, I)$
10: **for** $k = K$ **down to** 1 **do**
11:      Predict residual: $\hat{R}^{(0)} \leftarrow \hat{R}_\theta^{(0)}(R^{(k)}, k, X_{\text{noise}})$
12:      Compute posterior mean
13:         $\tilde{\mu}_\theta \leftarrow \frac{\sqrt{\bar{\alpha}_{k-1}}\beta_k}{1-\bar{\alpha}_k}\hat{R}^{(0)} + \frac{\sqrt{\alpha_k}(1-\bar{\alpha}_{k-1})}{1-\bar{\alpha}_k}R^{(k)}$
14:      Sample $R^{(k-1)} \sim \mathcal{N}(\tilde{\mu}_\theta, \tilde{\beta}_k I)$        ▷ Eq. equation 19
15: **end for**
16: $\hat{R} \leftarrow R^{(0)}$
17: **Final Prediction:**
18: $\hat{Y} \leftarrow \hat{Y}_{\text{non}} + \hat{Y}_{\text{stat}} + \hat{R}$
19: **Return** $\hat{Y}$

---

collected from the Weather Station of the Max Planck Biogeochemistry Institute in 2020, with 21 meteorological indicators collected every 10 minutes. [6] (7) PEMS provides California traffic network data recorded in 5-minute windows, from which we use four public subsets (PEMS03, PEMS04, PEMS07, and PEMS08) following SCINet (Liu et al., 2022a). [7]

We follow the data processing protocol and split configurations from Wu et al. (2021) and Li et al. (2024b). The lookback length is fixed to 96, and the prediction length is fixed to 192, with the exception of the ILI dataset, where the lookback length and prediction length are both set to 36. For short-term forecasting tasks, the prediction lengths are set to 12, 24, 48, and 96 following Liu et al. (2024). The details of all the datasets are provided in Table 5.

Table 5: Detailed dataset descriptions, including dimension, context length, label length, prediction length, and frequency.

| Dataset | Dim | Context length | Label length | Prediction length | Frequency |
|---|---|---|---|---|---|
| ILI | 7 | 36 | 16 | 36 | 1 week |
| Exchange | 8 | 96 | 48 | 192 | 1 day |
| Electricity | 321 | 96 | 48 | 192 | 1 hour |
| Traffic | 862 | 96 | 48 | 192 | 1 hour |
| ETTm2, ETTm1 | 7 | 96 | 48 | 192 | 15 mins |
| Weather | 21 | 96 | 48 | 192 | 10 mins |
| PEMS03 | 358 | 96 | 48 | {12, 24, 48, 96} | 5 mins |
| PEMS04 | 307 | 96 | 48 | {12, 24, 48, 96} | 5 mins |
| PEMS07 | 883 | 96 | 48 | {12, 24, 48, 96} | 5 mins |
| PEMS08 | 170 | 96 | 48 | {12, 24, 48, 96} | 5 mins |

### E.2 EVALUATION METRICS

We employ two categories of evaluation metrics: deterministic metrics for point forecasts and probabilistic metrics for uncertainty estimation. Let $x \in \mathbb{R}^d$ denote the ground truth values and $\hat{x} \in \mathbb{R}^d$ represent the predicted values.

---

[6]Weather: https://www.bgc-jena.mpg.de/wetter/
[7]PEMS: http://pems.dot.ca.gov

- **Mean Squared Error (MSE)**:

$$\text{MSE}(x, \hat{x}) = \frac{1}{d}\|x - \hat{x}\|_2^2 = \frac{1}{d}\sum_{i=1}^{d}(x_i - \hat{x}_i)^2, \tag{35}$$

where $\|\cdot\|_2$ denotes the $\ell_2$ norm.

- **Mean Absolute Error (MAE)**:

$$\text{MAE}(x, \hat{x}) = \frac{1}{d}\|x - \hat{x}\|_1 = \frac{1}{d}\sum_{i=1}^{d}|x_i - \hat{x}_i|, \tag{36}$$

where $\|\cdot\|_1$ denotes the $\ell_1$ norm.

For assessing probabilistic forecasts and uncertainty estimation, we utilize:

- **Continuous Ranked Probability Score (CRPS)** (Matheson & Winkler, 1976; Gneiting & Raftery, 2007):

$$\text{CRPS}(F, x) = \int_{-\infty}^{\infty}(F(y) - \mathbb{I}\{x \leq y\})^2 dy, \tag{37}$$

where $F(y)$ is the predicted cumulative distribution function.

- **Summed CRPS (CRPS$_{\text{sum}}$)**:

$$\text{CRPS}_{\text{sum}} = \mathbb{E}_t\left[\text{CRPS}(F_{\text{sum}}^{-1}, \sum_{i=1}^{d}x_i)\right], \tag{38}$$

where $F_{\text{sum}}^{-1}$ is obtained through dimension-wise summation of samples.

To specifically evaluate prediction intervals, we employ:

- **Prediction Interval Coverage Probability (PICP)** (Yao et al., 2019):

$$\text{PICP} = \frac{1}{N}\sum_{i=1}^{N}\mathbb{I}\{x_i \in [\hat{x}_i^{\text{low}}, \hat{x}_i^{\text{high}}]\}, \tag{39}$$

where $N$ represents the total number of observations, $x_i \in \mathbb{R}^d$ denotes the true value for the $i$-th observation, and $\hat{x}_n^{\text{low}}$ and $\hat{x}_n^{\text{high}}$ correspond to the $2.5^{th}$ and $97.5^{th}$ percentiles of the predicted distribution respectively, with $\mathbb{I}$ being the indicator function. This metric quantifies the empirical coverage probability by measuring the proportion of true observations falling within the predicted interval bounds. When the predicted distribution matches the true data distribution perfectly, the PICP should theoretically equal the nominal coverage level of 95% for the specified $2.5^{th} - 97.5^{th}$ percentile range.

- **Quantile Interval Coverage Error (QICE)** (Han et al., 2022):

$$\text{QICE} = \frac{1}{M}\sum_{m=1}^{M}\left|\rho_m - \frac{1}{M}\right|, \quad \rho_m = \frac{1}{N}\sum_{i=1}^{N}\mathbb{I}\{x_i \in [\hat{x}_i^{\text{low},m}, \hat{x}_i^{\text{high},m}]\}. \tag{40}$$

QICE can be viewed as PICP with finer granularity and without uncovered quantile ranges. Under the optimal scenario where the predicted distribution perfectly matches the target distribution, the QICE value should be equal to 0.

### E.3 IMPLEMENTATION OF NON-STATIONARY ADAPTER IN FALDA

As discussed in Section 3, we propose a non-stationary adapter $f_w$ to capture the non-stationary patterns in time series data. While a linear projection from $X_{\text{non}}$ to $\hat{Y}_{\text{non}}$ offers a straightforward approach, we enhance this design by additionally incorporating the complete lookback window $X$ as auxiliary input following the approach outlined in Ye et al. (2024). This extension enables richer temporal context utilization, improving prediction accuracy for $Y_{\text{non}}$. The output of the adapter is computed as follows:

$$\hat{Y}_{\text{non}} = f_w(X_{\text{non}}, X) = W_3 \operatorname{ReLU}\left(W_2 \operatorname{Concat}\left(\operatorname{ReLU}(W_1 X_{\text{non}}), X\right)\right), \tag{41}$$

where $W_1$, $W_2$, and $W_3$ are learnable weight matrices. The concatenation operation explicitly combines the processed non-stationary features with the original input, allowing the network to leverage both representations.

### E.4 IMPLEMENTATION DETAILS

All the experiments are conducted on a single NVIDIA L20 48GB GPU, utilizing PyTorch (Paszke et al., 2019). We set the number of diffusion steps to $K = 1000$, adopting a linear noise schedule following the configuration in Li et al. (2024b). Following DDIM (Song et al., 2021), we accelerate the sampling procedure by selecting a 10-point subsequence (with a stride of 100 steps) from the original 1000 diffusion steps, effectively skipping intermediate computations while maintaining generation quality. Correspondingly, we adjust the fine-tuning diffusion step $k'$ to align with the subsampling stride, setting $k' = 100$ to match the first sampling interval. The parameter $\eta$ controls the determinism level in DDIM sampling, where $\eta = 0$ yields a fully deterministic generation process. We utilize the Adam optimizer (Kingma & Ba, 2014) with a learning rate of $10^{-4}$ and L1 loss. Early stopping is applied after $\{5, 10, 15\}$ epochs without improvement, with a maximum of 200 epochs. The batch size is set to 32 during training and 8 for testing. The context length, label length, and prediction length are detailed in Table 5. To ensure robust statistical evaluation, we generate 100 prediction instances for each test sample to reliably compute the evaluation metrics. We show the point estimate performance and probabilistic forecasting performance in Table 1 and Table 2, respectively. The hidden dimension $H_d$ is selected from the set $\{64, 128, 256, 512\}$. Hyperparameters $K_1$ and $K_2$ are chosen from $\{0, 1, 2, \ldots, \lfloor T/2 \rfloor + 1\}$. The kernel size for the moving average operation in DEMA is fixed at $a = 25$. For reference, we provide a detailed hyperparameter configuration for FALDA with iTransformer as the backbone architecture in Table 6. Furthermore, as discussed in Section 4.2, we extend our framework to integrate with alternative backbone models (Autoformer, Transformer, and Informer), with their corresponding configurations detailed in Table 7. All relevant hyperparameters referenced in Section 3 are explicitly documented in these configuration tables.

Table 6: Hyperparameter settings for FALDA with iTransformer backbone.

|  | Exchange | ILI | ETTm2 | Electricity | Traffic | Weather |
|---|---|---|---|---|---|---|
| $\eta$ | 1.0 | 0.5 | 1.0 | 1.0 | 1.0 | 1.0 |
| $\delta$ | 0 | 0 | 1 | 2 | 1 | 0 |
| $\Delta$ | 3 | 3 | 10 | 10 | 20 | 3 |
| $K_1$ | 0 | 0 | 0 | 0 | 0 | 2 |
| $K_2$ | 32 | 2 | 5 | 20 | 3 | 25 |

Table 7: Hyperparameter settings for FALDA with other backbones.

|  | Exchange | ILI | ETTm2 | Electricity | Traffic | Weather |
|---|---|---|---|---|---|---|
| $\eta$ | 1.0 | 0.5 | 1.0 | 1.0 | 1.0 | 1.0 |
| $\delta$ | 0 | 0 | 1 | 2 | 1 | 0 |
| $\Delta$ | 3 | 3 | 10 | 10 | 20 | 3 |
| $K_1$ | 2 | 2 | 5 | 0 | 30 | 2 |
| $K_2$ | 32 | 2 | 5 | 10 | 2 | 25 |

### E.5 SELECTION OF FREQUENCY HYPERPARAMETERS

FALDA incorporates two key hyperparameters for frequency component selection: $K_1$ and $K_2$. Beyond the conventional grid search, we also explore an adaptive selection strategy to accommodate diverse datasets. Specifically, the selection of $K_1$ is guided by the dominant frequency ratio $p_1$, with a recommended range of 10 % to 20 %. The selection of $K_2$ is determined by the noise frequency ratio $p_2$, with a recommended range of 0.1 % to 1%. We select the values of $K_1$ and $K_2$ based on the

average number of frequencies with amplitudes above $p_1$ and below $p_2$ of the maximum amplitude in the training set, respectively. Table 8 shows the values of $K_1$ under different dominant frequency ratios $p_1$. Table 9 shows the values of $K_2$ under different noise frequency ratios $p_2$. The sensitivity analyses of $K_1$ and $K_2$ are presented in Figure 4.

Table 8: Values of $K_1$ under different dominant frequency ratios $p_1$.

| $p_1$ (%) | Dataset | | | | | | |
|---|---|---|---|---|---|---|---|
| | ILI | Exchange | Electricity | Traffic | ETTm2 | Weather | ETTh2 |
| 30 | 3 | 1 | 3 | 4 | 2 | 3 | 3 |
| 20 | 3 | 2 | 4 | 8 | 3 | 4 | 5 |
| 17 | 4 | 2 | 5 | 9 | 4 | 4 | 6 |
| 15 | 4 | 2 | 5 | 10 | 5 | 5 | 7 |
| 13 | 5 | 2 | 6 | 12 | 5 | 5 | 8 |
| 10 | 6 | 2 | 8 | 16 | 7 | 7 | 10 |
| 7 | 8 | 3 | 12 | 21 | 10 | 9 | 14 |
| 5 | 10 | 5 | 16 | 27 | 13 | 11 | 18 |

Table 9: Values of $K_2$ under different noise frequency ratios $p_2$.

| $p_2$ (%) | Dataset | | | | | | |
|---|---|---|---|---|---|---|---|
| | ILI | Exchange | Electricity | Traffic | ETTm2 | Weather | ETTh2 |
| 2.0 | 3 | 39 | 17 | 9 | 25 | 29 | 20 |
| 1.5 | 2 | 36 | 12 | 6 | 22 | 26 | 16 |
| 1.0 | 1 | 32 | 7 | 4 | 17 | 22 | 12 |
| 0.7 | 1 | 28 | 5 | 2 | 13 | 18 | 9 |
| 0.5 | 0 | 23 | 3 | 1 | 11 | 15 | 7 |
| 0.3 | 0 | 17 | 1 | 1 | 8 | 11 | 5 |
| 0.1 | 0 | 7 | 1 | 0 | 5 | 5 | 3 |
| 0.05 | 0 | 4 | 0 | 0 | 4 | 4 | 3 |

# F ADDITIONAL EXPERIMENTAL RESULTS

## F.1 STUDY ON DENOISER ARCHITECTURE

As described in Section 3.2, we introduce DEMA (Denoising MLP with Adaptive Layer Normalization), an MLP-based denoising module that utilizes Adaptive Layer Normalization (AdaLN) for feature transformation. The encoder layer employs a Moving Average (MA) operation to separate the latent variable into two components: seasonal and trend features. These components are then processed through independent AdaLN transformations, each governed by three trainable parameters: scale, shift, and gating coefficients, as specified in Eq. 15. To evaluate the architectural decisions in DEMA, we compare against two baseline variants in Table 10:

- **AD-MA**: This baseline removes the Moving Average decomposition in Eq. 13, applying AdaLN only to the undivided latent variable. While this configuration helps assess the importance of MA decomposition, it reduces the parameter count compared to DEMA. To address this confounding factor, we introduce a second controlled variant.

- **AD+LV**: This baseline maintains DEMA's parameter count while removing the feature decomposition step. Specifically, it implements two parallel AdaLN operations on the original latent variable (rather than on decomposed features). This design enables direct comparison of architectural contributions by isolating the effect of feature decomposition from pure parameter increases.

Experimental results demonstrate that DEMA consistently outperforms both variants in most datasets.

Diffusion-TS (Yuan & Qiao, 2024) proposes a decomposition-based denoiser architecture, which uses the seasonal-trend decomposition techniques, targeting the complete time series instead of noise

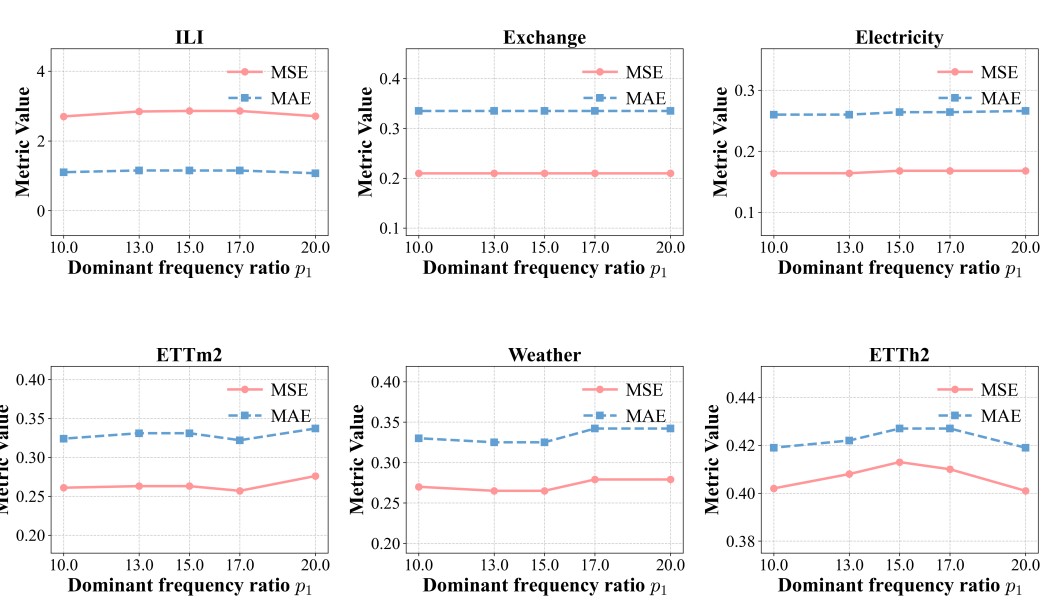

(a) Sensitivity analysis of dominant frequency ratio.

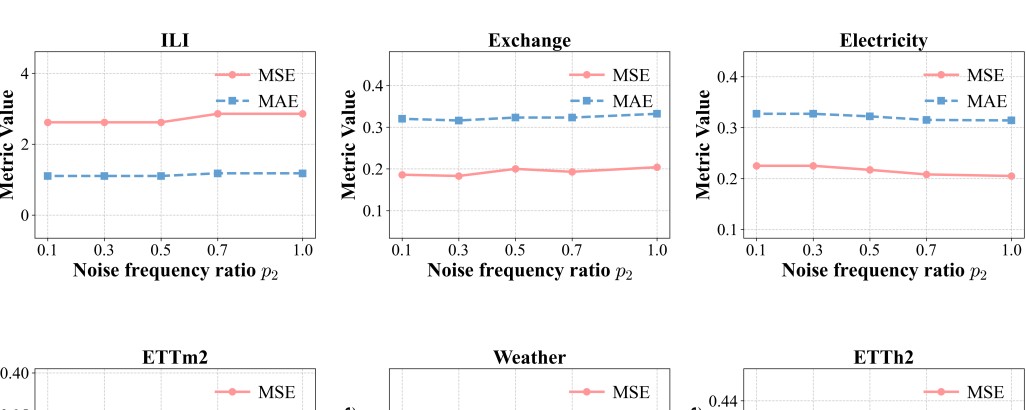

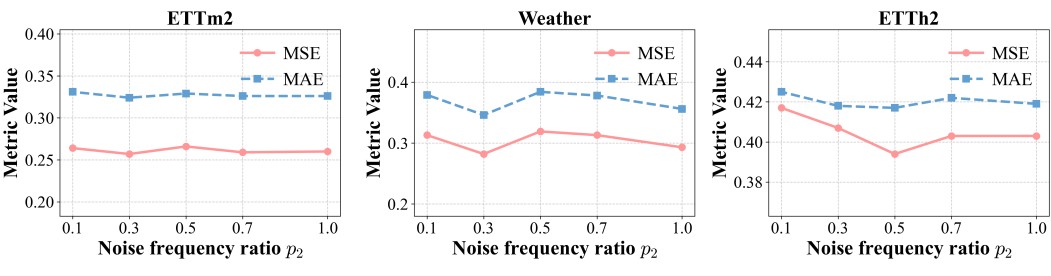

(b) Sensitivity analysis of noise frequency ratio.

Figure 4: Sensitivity Analysis of Frequency Hyperparameters. (a) Dominant frequency ratio $p_1$; (b) Noise frequency ratio $p_2$.

Table 10: Ablation study on denoiser architecture: comparison of DEMA and its variants. All experiments are repeated 10 times to compute the Means and Standard Deviation.

| Dataset | DEMA | | AD-MA | | AD+LV | |
|---|---|---|---|---|---|---|
| | MSE | MAE | MSE | MAE | MSE | MAE |
| Exchange | **0.180 ± 0.011** | **0.308 ± 0.009** | 0.197 ± 0.018 | 0.319 ± 0.014 | 0.183 ± 0.014 | 0.311 ± 0.010 |
| ILI | **1.652 ± 0.062** | 0.793 ± 0.026 | 1.735 ± 0.156 | 0.810 ± 0.058 | 1.666 ± 0.091 | **0.783 ± 0.031** |
| ETTm2 | **0.250 ± 0.003** | **0.307 ± 0.003** | 0.250 ± 0.005 | 0.307 ± 0.004 | 0.252 ± 0.004 | 0.308 ± 0.002 |
| Weather | **0.217 ± 0.003** | **0.261 ± 0.004** | 0.220 ± 0.002 | 0.264 ± 0.004 | 0.219 ± 0.005 | 0.262 ± 0.005 |

during the diffusion step. Table 11 compares the forecasting performance between Diffusion-TS and FALDA. While the denoiser in Diffusion-TS explicitly attempts to decompose and reconstruct different temporal components, the results indicate that FALDA achieves demonstrably superior performance metrics. This difference underscores the efficacy of our proposed uncertainty decomposition and DMRR framework, aligning strongly with our theoretical analysis.

Table 11: Forecasting Performance of FALDA vs. Diffusion-TS.

| Method | Exchange | | | ILI | | | ETTm2 | | | Electricity | | | Traffic | | |
|---|---|---|---|---|---|---|---|---|---|---|---|---|---|---|---|
| | MSE | MAE | CRPS | MSE | MAE | CRPS | MSE | MAE | CRPS | MSE | MAE | CRPS | MSE | MAE | CRPS |
| Diffusion-TS | 3.628 | 1.564 | 0.633 | 6.053 | 1.788 | 1.612 | 2.372 | 1.788 | 1.035 | 1.072 | 1.788 | 0.633 | 1.473 | 0.815 | 0.668 |
| FALDA | **0.165** | **0.296** | **0.289** | **1.666** | **0.821** | **0.721** | **0.246** | **0.301** | **0.244** | **0.163** | **0.248** | **0.231** | **0.412** | **0.251** | **0.245** |

## F.2 DOES DIFFUSION HELP? FREQUENCY DECOMPOSITION ABLATION STUDY

As analyzed in Appendix C.3, we introduce a temporal decomposition operation to strengthen the point forecasting capability of the backbone model, while the diffusion process primarily handles aleatoric uncertainty learning. To investigate whether probabilistic learning provides additional benefits to point forecasting, we conduct a comparative study with two deterministic models that exclude the diffusion component:

- **NDB (Non-decomposed Backbone)**: The baseline backbone model without temporal decomposition operation.
- **DB (Decomposed Backbone)**: An enhanced architecture that incorporates (1) input decomposition that separates low-frequency noise components, and (2) an NS-adapter module for non-stationary feature learning.

As shown in Table 12, the complete FALDA framework demonstrates superior performance compared to both deterministic variants (NDB and DB). These results suggest that: this decomposition operation effectively improves forecasting accuracy. Additionally, the diffusion component in FALDA provides additional performance gains beyond what can be achieved through decomposition alone. This empirical evidence confirms that probabilistic learning through diffusion modeling contributes positively to point forecasting performance when combined with our proposed decomposition architecture.

Table 12: Ablation study on the benefits of probabilistic residual learning in forecasting performance.

| Method | Exchange | | ILI | | Electricity | | Traffic | |
|---|---|---|---|---|---|---|---|---|
| | MAE | MSE | MAE | MSE | MAE | MSE | MAE | MSE |
| Ours | **0.165** | **0.296** | **1.666** | **0.821** | **0.163** | **0.248** | **0.412** | **0.251** |
| NDB | 0.194 | 0.315 | 1.786 | 0.826 | 0.165 | 0.249 | 0.439 | 0.276 |
| DB | 0.194 | 0.316 | 1.791 | 0.828 | 0.165 | 0.250 | 0.439 | 0.276 |

In addition to the plug-and-play experiments presented in the main text (shown in Table 3), we also conducted further plug-and-play experiments using TimeXer (Wang et al., 2024) as the backbone to evaluate the integration performance of FALDA with the latest TSF models (see Table 13).

Table 13: Plug-and-play performance with TimeXer.

| Method | ETTm1 | | ETTm2 | | Weather | | Electricity | | Traffic | | ILI | | Exchange | |
|--------|-------|-----|-------|-----|---------|-----|-------------|-----|---------|-----|-----|-----|----------|-----|
| | MSE | MAE | MSE | MAE | MSE | MAE | MSE | MAE | MSE | MAE | MSE | MAE | MSE | MAE |
| TimeXer | 0.373 | 0.389 | 0.243 | 0.303 | 0.213 | 0.254 | **0.179** | **0.283** | 0.507 | 0.352 | 2.711 | 1.063 | **0.181** | **0.302** |
| FALDA | **0.367** | **0.374** | **0.241** | **0.294** | **0.204** | **0.248** | 0.188 | 0.293 | **0.495** | **0.323** | **1.889** | **0.855** | 0.199 | 0.322 |

However, under some specific parameter configurations, it has been observed that integrating probabilistic residual learning frameworks with iTransformer may lead to a marginal decline in point estimation performance. To demonstrate this observation, experiments are conducted on $D^3U$, FALDA, and iTransformer under unified parameter settings (see Table 18), with the results presented in Table 14. Although FALDA achieved 6 first-place rankings, surpassing iTransformer's 4 first-place rankings, a performance decline was observed on the Electricity dataset. Furthermore, $D^3U$ consistently exhibits inferior point estimation performance compared to iTransformer across the majority of datasets.

Table 14: Residual probabilistic learning frameworks vs. their backbone models: point forecasting performance comparison.

| Dataset | iTransformer | | D3U | | FALDA | |
|---------|--------------|-----|-----|-----|-------|-----|
| | MSE | MAE | MSE | MAE | MSE | MAE |
| ETTm1 | **0.377** | 0.390 | 0.387 | 0.399 | 0.378 | **0.380** |
| ETTm2 | **0.251** | 0.311 | 0.256 | 0.316 | 0.255 | **0.310** |
| Weather | 0.237 | 0.268 | 0.232 | 0.279 | **0.220** | **0.258** |
| Electricity | **0.162** | **0.253** | 0.168 | 0.261 | 0.167 | 0.253 |
| Traffic | 0.460 | 0.312 | 0.421 | 0.290 | **0.420** | **0.265** |
| 1st Count | 4 | | 0 | | **6** | |

The phenomenon is now analyzed. During the inference process, 100 prediction instances are generated from the learned residual distribution, denoted as $\{r_i\}_{i=1}^{100}$, and the mean of these 100 samples is calculated as $m = \frac{1}{100}\sum_{i=1}^{100} r_i$. The final prediction $\{r_i + \hat{Y}\}_{i=1}^{100}$ is then obtained by adding the preliminary estimation $\hat{Y}$ from the deterministic model to the residual samples, with the mean of the final predictions being: $m + \hat{Y}$. In an ideal scenario, where the residuals only contain random uncertainty, the mean of the residual distribution should be zero. However, in practice, $m \neq 0$, which may stem from two factors: first, the point estimate model always involves estimation error and second, the limited number of samples introduces a bias in the mean.

A direct solution to this issue is to correct the final result by manually removing this bias, with the final prediction becomes: $\{r_i - m + \hat{Y}\}_{i=1}^{100}$ with the mean: $\hat{Y}$. This correction ensures that point estimation performance does not degrade when integrated with the backbone, while simultaneously highlighting the flexibility of residual probabilistic learning, as it enables further adjustment of the learned bias without any additional training cost.

### F.3 RESIDUAL FRAMEWORK COMPARISON WITH IDENTICAL BACKBONE

In this section, we evaluate the performance of TMDM, $D^3U$, and FALDA with the NSformer backbone. The parameter configuration follows Li et al. (2024b), while the correlation results are reported in accordance with Li et al. (2025). Our experimental setup maintains consistency between the training and evaluation phases. Table 15 presents the point forecasting performance, measured by MAE and MSE. Meanwhile, Table 16 summarizes the probabilistic forecasting performance using CRPS and CRPS$_{\text{sum}}$ metrics. The experimental results demonstrate that FALDA achieves superior performance in both point and probabilistic forecasting tasks, validating the effectiveness of our proposed framework. By incorporating a time series decomposition mechanism to decouple distinct temporal components, our method facilitates more balanced learning of both epistemic and aleatoric uncertainties, thereby contributing to enhanced forecasting performance.

Table 15: Point forecasting performance comparison of different residual learning frameworks with NSformer backbone.

| Method | Exchange | | ILI | | ETTm2 | | Electricity | | Traffic | | Weather | |
|---|---|---|---|---|---|---|---|---|---|---|---|---|
| | MSE | MAE | MSE | MAE | MSE | MAE | MSE | MAE | MSE | MAE | MSE | MAE |
| TMDM | 0.260 | 0.365 | 1.985 | 0.846 | 0.524 | 0.493 | 0.222 | 0.329 | 0.721 | 0.411 | 0.244 | 0.286 |
| D3U | 0.268 | 0.378 | 2.220 | 0.920 | **0.317** | 0.399 | 0.216 | 0.328 | 0.678 | 0.402 | **0.215** | **0.267** |
| Ours | **0.238** | **0.342** | **1.918** | **0.803** | 0.324 | **0.356** | **0.180** | **0.278** | **0.625** | **0.317** | 0.244 | 0.278 |

Table 16: Probabilistic forecasting performance comparison of different residual learning frameworks with NSformer Backbone.

| Method | Exchange | | ILI | | ETTm2 | | Electricity | | Traffic | | Weather | |
|---|---|---|---|---|---|---|---|---|---|---|---|---|
| | CRPS | CRPS$_{sum}$ | CRPS | CRPS$_{sum}$ | CRPS | CRPS$_{sum}$ | CRPS | CRPS$_{sum}$ | CRPS | CRPS$_{sum}$ | CRPS | CRPS$_{sum}$ |
| TMDM | 0.316 | 0.209 | 0.921 | 0.524 | 0.380 | 0.226 | 0.446 | **0.137** | 0.552 | **0.179** | 0.226 | 0.292 |
| D3U | 0.387 | 0.218 | 1.014 | 0.454 | **0.302** | **0.147** | 0.381 | 0.157 | 0.472 | 0.207 | **0.196** | **0.273** |
| Ours | **0.299** | **0.171** | **0.674** | **0.349** | 0.334 | 0.195 | **0.269** | 0.167 | **0.312** | 0.195 | 0.235 | 0.333 |

We further assess the performance of D$^3$U and FALDA with iTransformer as the backbone under the unified hyperparameter settings (Table 18). The results in Table 17 show that FALDA achieved 13 first-place rankings across five datasets, while D$^3$U secured 7 first-place rankings, indicating that FALDA outperforms D$^3$U overall.

Table 17: Comparison between D$^3$U and FALDA with iTransformer backbone.

| Dataset | D3U | | | | FALDA | | | |
|---|---|---|---|---|---|---|---|---|
| | MSE | MAE | CRPS | CRPS_sum | MSE | MAE | CRPS | CRPS_sum |
| ETTm1 | 0.387 | 0.399 | **0.295** | 0.805 | **0.378** | **0.380** | 0.312 | **0.640** |
| ETTm2 | 0.256 | 0.316 | **0.246** | **0.101** | **0.255** | **0.310** | 0.277 | 0.160 |
| Weather | 0.232 | 0.279 | 0.213 | **0.224** | **0.220** | **0.258** | **0.210** | 0.319 |
| Electricity | 0.168 | 0.261 | **0.195** | **0.151** | **0.167** | **0.253** | 0.238 | 0.157 |
| Traffic | 0.421 | 0.290 | **0.222** | 0.169 | **0.420** | **0.265** | 0.255 | **0.168** |
| 1st Count | 7 | | | | 13 | | | |

## F.4 COMPARISON WITH CLASSICAL TIME SERIES FORECASTING METHODS

To further strengthen the validity of FALDA, we include comparison to two classical statistical TSF baselines: Naive and Seasonal Naive (Hyndman et al.). As shown in Table 19, we report the point forecasting performance of Naive, Seasonal Naive, DLinear, and FALDA across six standard benchmark datasets, where **bold** denotes the best performance, and underline denotes the second-best. We include DLinear as a representative neural network-based TSF model with a linear architecture. Among the four methods, FALDA achieves the best performance, significantly surpassing all baselines. Specifically, compared with the Naive method, FALDA achieves a 77.7% improvement in MSE and a 55.6% improvement in MAE. When compared with the Seasonal Naive method, FALDA yields a 43.1% improvement in MSE and a 28.2% improvement in MAE. We also evaluate the probabilistic performance of the Naive and Seasonal Naive models when combined with prediction intervals, labeled as "Naive-I" and "Seasonal Naive-I" in Table 20. The results show that FALDA improves by up to 71.3% in CRPS$_{sum}$, demonstrating its significant advantage over traditional probabilistic approaches. The point and probabilistic improvements show the effectiveness of FALDA in modeling temporal patterns over classical TSF methods.

## F.5 TRAINING STRATEGY EXPERIMENTS

As defined in Eq. 9, our loss function incorporates both a diffusion loss for denoiser optimization and a fine-tuning loss $\mathcal{L}_{\text{finetune}} = \|R - \text{sg}(\hat{R}_\theta^{(0)}(R^{(k')}, k', c))\|^2$ to simultaneously enhance the point estimate

Table 18: Unified hyperparameter settings for D³U and FALDA in Table 17.

| Setting | ETTm1 | ETTm2 | Electricity | Traffic | Weather |
|---|---|---|---|---|---|
| Train batch_size | 128 | 128 | 32 | 32 | 128 |
| Test batch_size | 64 | 64 | 8 | 2 | 16 |
| Backbone d_model | 512 | 512 | 256 | 512 | 256 |
| Backbone d_ff | 512 | 512 | 512 | 512 | 256 |
| Backbone e_layer | 2 | 2 | 3 | 3 | 2 |
| Backbone d_layer | 1 | 1 | 1 | 1 | 1 |

Table 19: Point forecasting performance comparison with classical TSF methods.

| Dataset | Seasonal Naive | | Naive | | Dlinear | | FALDA | |
|---|---|---|---|---|---|---|---|---|
| | MSE | MAE | MSE | MAE | MSE | MAE | MSE | MAE |
| ILI | 2.266 | 0.957 | 7.714 | 1.906 | 2.235 | 1.059 | **1.666** | **0.821** |
| Exchange | 0.716 | 0.643 | 0.167 | **0.289** | 0.167 | 0.301 | **0.165** | 0.296 |
| Electricity | 0.304 | 0.324 | 1.596 | 0.951 | 0.196 | 0.285 | **0.163** | **0.248** |
| Traffic | 1.093 | 0.458 | 2.747 | 1.085 | 0.598 | 0.370 | **0.412** | **0.251** |
| ETTm2 | 0.321 | 0.337 | 0.340 | 0.371 | 0.284 | 0.362 | **0.246** | **0.301** |
| Weather | 0.343 | 0.305 | 0.309 | 0.292 | 0.218 | 0.278 | **0.215** | **0.255** |
| Average | 0.841 | 0.504 | 2.146 | 0.816 | 0.616 | 0.443 | **0.478** | **0.362** |

Table 20: Probabilistic forecasting performance comparison with classical TSF methods.

| Dataset | Seasonal Naive-I | | Naive-I | | FALDA | |
|---|---|---|---|---|---|
| | CRPS | CRPSsum | CRPS | CRPSsum | CRPS | CRPSsum |
| ILI | 0.773 | 0.401 | 1.585 | 1.003 | 0.721 | 0.387 |
| Exchange | 0.513 | 0.376 | 0.216 | 0.137 | 0.289 | 0.126 |
| Electricity | 0.276 | 0.179 | 1.106 | 1.091 | 0.231 | 0.160 |
| Traffic | 0.465 | 0.410 | 1.428 | 1.218 | 0.245 | 0.163 |
| ETTm2 | 0.273 | 0.162 | 0.428 | 0.184 | 0.244 | 0.141 |
| Weather | 0.271 | 0.338 | 0.729 | 0.824 | 0.207 | 0.298 |
| Average | 0.429 | 0.310 | 0.915 | 0.743 | 0.323 | 0.213 |
| Promotion | **24.7**% | **31.3**% | **64.7**% | **71.3**% | - | - |

models. The hyperparameter $k'$ allows for flexible selection of diffusion steps during fine-tuning. To validate this choice, we perform ablation studies comparing models trained with and without fine-tuning, as well as models fine-tuned at different diffusion steps $k'$. The experimental results presented in Figure 5 demonstrate that the fine-tuning operation provides consistent improvements over the no-fine-tuning setting. Additionally, our chosen configuration with $k' = 100$ achieves competitive MSE and MAE performance among different step selections, suggesting the validity of our configuration as mentioned in Appendix E.4.

### F.6 TRAINING AND INFERENCE EFFICIENCY

As discussed in Section 3, FALDA reconstructs the sample directly, rather than learning the noise at each diffusion step during the training phase, which reduces the learning complexity of the time series component. Additionally, our denoiser DEMA, which is designed as a lightweight MLP architecture, alleviates the training burden. During the inference process, we employ DDIM to accelerate inference. These design choices collectively contribute to the efficiency of FALDA, while maintaining its effectiveness. We conduct experiments to demonstrate its efficiency. As depicted in Figure 6, FALDA exhibits superior convergence properties compared to TMDM. While TMDM requires approximately 30 epochs to converge on the Exchange dataset, FALDA achieves competitive performance after only 1 epoch. For fair comparison, we maintain identical training configurations with TMDM, including the learning rate ($1 \times 10^{-4}$) and optimization method (Adam optimizer). This accelerated

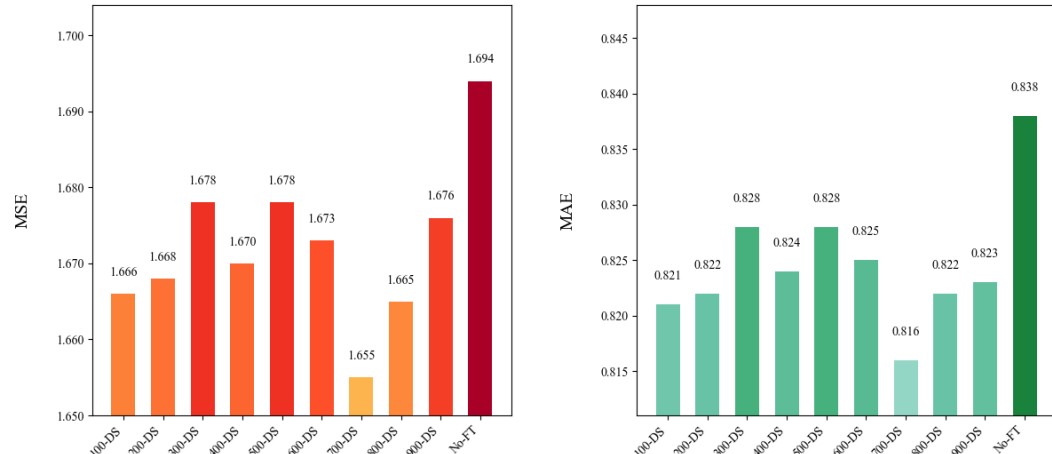

Figure 5: Evaluation of different training strategies on the ILI Dataset. The left subplot shows the MSE performance, while the right subplot shows the MAE performance. $k'$-DS: fine-tuning with diffusion step $k'$. No-FT: no fine-tuning.

convergence further underscores FALDA's computational advantages without compromising model performance.

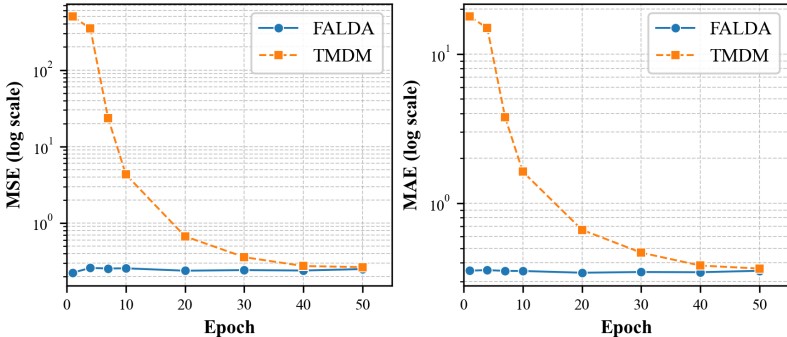

Figure 6: Training speed comparison between FALDA and TMDM on the Exchange dataset. The curves depict the evolution of metrics: MSE (left) and MAE (right) across training epochs.

Building upon these convergence advantages, we implement a reduced early stopping patience for FALDA compared to TMDM during the training process, as detailed in Appendix E.4. During inference, we employ DDIM (Denoising Diffusion Implicit Models) to accelerate the reverse diffusion process, thereby significantly reducing both inference time and memory requirements. Table 21 presents a comprehensive computational efficiency comparison between TMDM and FALDA across six benchmark datasets. The results demonstrate FALDA's consistent superiority in both training and inference phases. Specifically, FALDA achieves an inference speed improvement of up to $26.3\times$ on the ETTm2 dataset, while attaining a training speed enhancement of up to $13.7\times$ on the Exchange dataset. Furthermore, FALDA delivers a $2.1\times$ training speed-up on the Electricity dataset (from 122.9 minutes to 58.3 minutes) and a $2.9\times$ inference speed-up on the Traffic dataset (from 472.3 minutes to 160.7 minutes). These substantial improvements in computational efficiency not only validate FALDA's practical utility for real-world applications but also highlight its capability for processing high-dimensional datasets.

Additionally, we present the training convergence speed comparison of FALDA, TMDM, and D³U in Table 22 and Table 23. The one-batch sampling time comparison (with batch size set to 64) across the ETTm2 and the ETTm1 datasets is provided in Table 24. The results show that FALDA achieves the fastest convergence and sampling time on both datasets, highlighting the practical effectiveness of our framework. Additionally, both FALDA and D³U, which are based on the DMRR diffusion framework,

Table 21: Comparison of training and inference times (minutes) between TMDM and FALDA [1].

| Dataset | TMDM | | FALDA (Ours) | |
|---|---|---|---|---|
| | Training | Inference [2] | Training | Inference |
| ILI | 3.0 | 0.6 | **0.4** | **0.1** |
| Exchange Rate | 9.6 | 10.5 | **0.7** | **0.5** |
| ETTm2 | 36.6 | 194.4 | **3.4** | **7.4** |
| Weather | 69.8 | 119.1 | **6.3** | **13.5** |
| Electricity | 122.9 | 272.9 | **58.3** | **88.3** |
| Traffic | 97.0 | 472.3 | **83.6** | **160.7** |

[1] All experiments were conducted on an NVIDIA L20 GPU with 48GB memory.
[2] Inference times were measured with 100 samples per test instance.

converge substantially faster than TMDM, which relies on a vanilla DDPM latent distribution. This observation is consistent with our theoretical analysis presented in Section 3.3.

Table 22: MSE performance comparison across training epochs on ETT datasets.

| Dataset | Method | Epochs | | | | | | | |
|---|---|---|---|---|---|---|---|---|---|
| | | 2 | 4 | 6 | 8 | 10 | 20 | 30 | 40 |
| ETTm1 | TMDM | 186.236 | 16.72 | 0.544 | 0.507 | 0.544 | 0.538 | 0.543 | 0.551 |
| | D$^3$U | 0.403 | 0.412 | 0.394 | 0.384 | 0.385 | **0.369** | **0.368** | **0.368** |
| | FALDA | **0.383** | **0.375** | **0.374** | **0.371** | **0.379** | 0.378 | 0.378 | 0.378 |
| ETTm2 | TMDM | 405.603 | 21.862 | 1.095 | 0.787 | 0.618 | 0.439 | 0.404 | 0.362 |
| | D$^3$U | 0.308 | 0.252 | 0.257 | 0.264 | 0.253 | **0.254** | 0.253 | 0.253 |
| | FALDA | **0.252** | **0.247** | **0.245** | **0.245** | **0.246** | 0.256 | **0.250** | **0.251** |

Table 23: MAE performance comparison across training epochs on ETT datasets.

| Dataset | Method | Epochs | | | | | | | |
|---|---|---|---|---|---|---|---|---|---|
| | | 2 | 4 | 6 | 8 | 10 | 20 | 30 | 40 |
| ETTm1 | TMDM | 10.859 | 3.061 | 0.505 | 0.463 | 0.472 | 0.471 | 0.471 | 0.475 |
| | D$^3$U | 0.414 | 0.427 | 0.406 | 0.397 | 0.401 | 0.388 | 0.389 | 0.389 |
| | FALDA | **0.383** | **0.376** | **0.375** | **0.375** | **0.380** | **0.380** | **0.379** | **0.379** |
| ETTm2 | TMDM | 15.405 | 3.84 | 0.837 | 0.659 | 0.555 | 0.406 | 0.390 | 0.377 |
| | D$^3$U | 0.385 | 0.322 | 0.326 | 0.332 | 0.321 | 0.316 | 0.316 | 0.316 |
| | FALDA | **0.316** | **0.307** | **0.303** | **0.302** | **0.308** | **0.311** | **0.311** | **0.311** |

Table 24: Comparison of sampling times for one batch (seconds) with batch size 64.

| Method | Sampling Time (seconds) | |
|---|---|---|
| | ETTm1 | ETTm2 |
| TMDM | 27.32 | 29.14 |
| D$^3$U | 1.38 | 1.51 |
| FALDA | **0.81** | **1.21** |

## F.7 PREDICTIVE INTERVALS RESULT

We present the result of PICE and QICE in Tabel 25, which assesses the ability of the model to accurately cover the true values within its prediction intervals and the precision of the estimates for these intervals, respectively. See Appendix E.2 for specific definitions of PICE and QICE.

Table 25: Comparison of PICP and QICE metrics.

| Dataset | Metric | TimeGrad | CSDI | TimeDiff | TMDM | Ours |
|---------|--------|----------|------|----------|------|------|
| Exchange | PICP | 69.16 | 69.21 | 20.80 | 74.54 | **97.88** |
|          | QICE | 5.32 | 5.49 | 13.34 | **4.38** | 5.49 |
| ILI | PICP | 74.29 | 76.18 | 3.69 | **87.83** | 77.49 |
|     | QICE | 7.86 | 7.75 | 15.50 | 6.74 | **4.42** |
| ETTm2 | PICP | 71.62 | 71.78 | 13.16 | 73.20 | **84.08** |
|       | QICE | 5.37 | 5.07 | 14.22 | 3.75 | **2.71** |
| Electricity | PICP | 75.93 | 78.94 | 32.37 | 82.35 | **93.48** |
|             | QICE | 5.34 | 4.74 | 12.74 | 3.81 | **2.90** |
| Traffic | PICP | 82.28 | 83.51 | 9.11 | 86.83 | **92.36** |
|         | QICE | 3.80 | 3.50 | 13.53 | 2.36 | **1.90** |
| Weather | PICP | 62.79 | 62.71 | 21.60 | 72.97 | **80.75** |
|         | QICE | 7.36 | 5.14 | 13.18 | 3.87 | **3.58** |

## F.8 SHORT-TERM TSF EXPERIMENTS

In addition to long-term time series forecasting, we also investigate the capability of FALDA for short-term TSF tasks. Experiments are conducted on four PEMS public subsets (PEMS03, PEMS04, PEMS07, PEMS08). Following the experimental configuration in Liu et al. (2024), prediction lengths of {12, 24, 48, 96} are tested. The results (see Table 26) show that FALDA achieves 21 best scores out of 32 metrics, while iTransformer attains only 11, demonstrating that FALDA maintains strong point estimation accuracy even with reduced prediction horizons. We attribute this advantage to our use of Fourier decomposition to dynamically extract the top-K dominant frequencies and bottom-K frequencies from each input instance. This temporal decomposition method is instance-specific (Ye et al., 2024), thereby enabling adaptive capture of non-stationary patterns and noise in the data.

Table 26: Short-term TSF performance of FALDA.

| Dataset | | iTransformer | | FALDA | |
|---------|------|------|------|------|------|
| | | MSE | MAE | MSE | MAE |
| PEMS03 | 12 | 0.071 | 0.174 | **0.071** | **0.173** |
|        | 24 | 0.093 | 0.201 | **0.092** | **0.196** |
|        | 48 | **0.125** | **0.236** | 0.157 | 0.261 |
|        | 96 | 0.164 | 0.275 | **0.156** | **0.263** |
| PEMS04 | 12 | 0.078 | 0.183 | **0.075** | **0.176** |
|        | 24 | 0.095 | 0.205 | **0.090** | **0.193** |
|        | 48 | 0.120 | 0.233 | **0.111** | **0.216** |
|        | 96 | 0.150 | 0.262 | **0.137** | **0.239** |
| PEMS07 | 12 | **0.067** | **0.165** | 0.073 | 0.170 |
|        | 24 | **0.088** | 0.190 | 0.092 | **0.189** |
|        | 48 | **0.110** | **0.215** | 0.122 | 0.223 |
|        | 96 | **0.139** | 0.245 | 0.145 | **0.243** |
| PEMS08 | 12 | 0.079 | 0.182 | **0.077** | **0.177** |
|        | 24 | 0.115 | 0.219 | **0.115** | **0.214** |
|        | 48 | 0.186 | **0.235** | **0.174** | 0.260 |
|        | 96 | **0.221** | **0.267** | 0.312 | 0.343 |
| 1st Count | | 11 | | **21** | |

## F.9 ABLATION STUDY ON UNCERTAINTY SEPARATION

To further validate the efficacy of the Uncertainty Separation (US) mechanism within the FALDA framework, we conduct an ablation study in Table 27, comparing FALDA against a variant without the US mechanism, denoted as NUS (No Uncertainty Separation). NUS extracts non-stationary term using Eq. 4, then processed it with the NS-Adapter. The remaining deterministic component is handled by the TS-Backbone. Both FALDA and NUS utilize Autoformer as the TS-Backbone. NUS adopts a residual learning paradigm without explicit uncertainty separation, following the approach

in Li et al. (2025). The results demonstrate that FALDA consistently outperforms NUS across both point estimation metrics and probabilistic metrics, thereby confirming the effectiveness of FALDA's US mechanism.

Table 27: Comparison of NUS and FALDA methods across various datasets.

| Dataset | MSE | | MAE | | CRPS | | CRPSsum | |
|---|---|---|---|---|---|---|---|---|
| | NUS | FALDA | NUS | FALDA | NUS | FALDA | NUS | FALDA |
| Exchange | 0.208 | **0.193** | 0.338 | **0.326** | 0.156 | **0.146** | 0.244 | **0.243** |
| ILI | 2.846 | **2.537** | 1.129 | **1.022** | 1.051 | **0.904** | 0.582 | **0.507** |
| ETTm2 | 0.267 | **0.247** | 0.329 | **0.315** | 0.262 | **0.261** | 0.162 | **0.153** |
| Weather | 0.302 | **0.292** | 0.369 | **0.356** | 0.320 | **0.298** | 0.542 | **0.448** |
| Electricity | 0.165 | **0.162** | 0.261 | **0.257** | 0.248 | **0.245** | 0.161 | **0.153** |

## G  LIMITATIONS

Although FALDA demonstrates competitive forecasting performance, it inherits a common limitation of diffusion-based probabilistic TSF models: the multi-step iterative denoising process leads to significantly slower inference speed compared to deterministic point estimation methods. This computational overhead becomes increasingly pronounced as the number of diffusion steps, and also imposes higher GPU memory requirements. Such constraints could limit its applicability in scenarios demanding low-latency predictions or under limited computational resources. Future work may investigate more efficient alternatives for probabilistic modeling, such as normalizing flow and rectified flow, to achieve a more favorable balance between predictive performance and computational efficiency.

## H  SHOWCASES

### H.1  CASE STUDY OF FALDA AND TMDM

**Real-World Dataset Showcases**    To demonstrate the superior probabilistic forecasting capability of FALDA, we present comparative visualizations of ground truth values and prediction results between FALDA and TMDM across four datasets in Figures 8, 9, 10, and 11. The figures display the predicted median along with 50% and 90% distribution intervals. Our experimental results suggest that FALDA achieves substantially improved point forecasting accuracy compared to TMDM. Moreover, our DMRR framework contributes to particularly accurate predictions for the first future time step, as clearly evidenced in Figure 8. While TMDM produces excessively wide prediction intervals for the initial future prediction, FALDA generates precise first-step forecasts with narrow confidence bounds that gradually widen over the forecasting horizon. Furthermore, the predicted uncertainty of TMDM does not exhibit a clear time-varying characteristic, which shows its limited ability to capture non-stationary uncertainty. In contrast, FALDA clearly demonstrates a distinct, time-increasing variance. This temporal behavior is highly consistent with observations in real-world financial data, where volatility typically increases over time. This visualization suggests that FALDA offers more accurate and interpretable forecasts that better capture the underlying data dynamics.

**Synthetic Dataset Showcases**    To further visually illustrate FALDA's non-stationary uncertainty modeling capability, we conduct an experiment on the synthetic dataset provided in Ye et al. (2025). Specifically, this synthetic dataset is a univariate Gaussian noise data where both the mean and the standard deviation increase linearly from 1 to 10. Figure 7 visualizes the estimated standard deviation of TMDM and FALDA on this linear synthetic dataset. It is observed that TMDM's estimated standard deviation appears lower than the true standard deviation across the test set, suggesting its limited capacity to fully capture the accumulating uncertainty. In contrast, FALDA effectively captures the distribution shift between the training and test datasets. We note that NsDiff (Ye et al., 2025) explicitly injects non-stationary prior knowledge into the diffusion endpoint to enhance its non-stationary uncertainty modeling capability. However, FALDA demonstrates a robust capability for uncertainty modeling even without explicitly introducing such priors into the diffusion process. We attribute this advantage to the design of our tailored modeling framework, which may allow us to

better leverage the inherent potential of diffusion models. It is well-established that diffusion models inherently possess the capacity to learn arbitrary distributions; therefore, the diffusion process is theoretically capable of learning time-varying uncertainty. However, previous work Li et al. (2024b) employed mixed learning objectives (e.g., deterministic non-stationary term, stationary term, inherent noise), which could potentially impede the model's focus on learning non-stationary uncertainty. Conversely, under our tailored modeling approach, where other time series components are handled by specific modules, FALDA is positioned to better unlock the potential of the diffusion process, allowing it to concentrate on probabilistic learning and thus effectively capturing non-stationary uncertainty even without explicitly introducing non-stationary priors. In addition, by conditioning the denoiser on $X_{\text{noise}}$, FALDA is able to effectively learn the coupled growth of variance between the history windows and the future time series in this case.

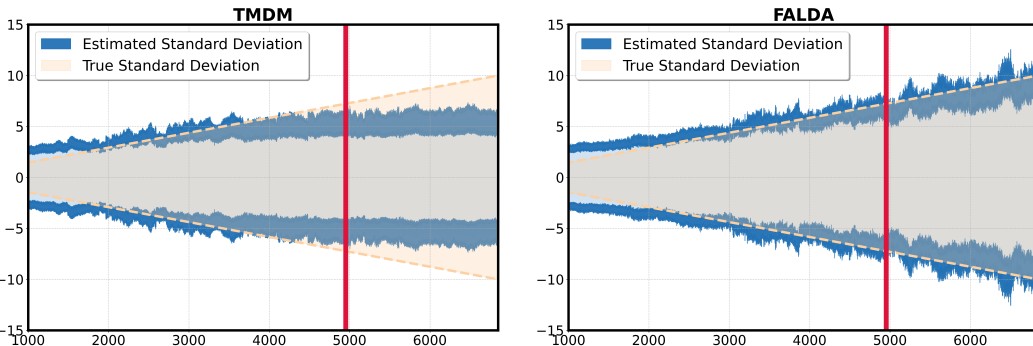

Figure 7: Estimated standard deviation of TMDM (left) and FALDA (right) on the linear synthetic dataset. The solid red line separates the training set (left) from the test set (right).

## H.2 TIME SERIES DECOMPOSITION VISUALIZATION

To illustrate our time series decomposition approach, Figures 12 and 13 visualize the distinct temporal components obtained through the decomposition method described in Eq. 4. Figures 12 and 13 demonstrate the distinct decomposition characteristics of iTransformer and other backbones, respectively. The detailed implementation settings for these decomposition strategies are provided in Appendix E.4.

## H.3 VISUALIZATION OF KEY COMPONENTS IN FALDA

To further showcase the predictive capabilities of FALDA, we visualize the outputs of its three key components across different datasets with the TS-Backbone set as Autoformer. Figures 14, 15, 16, and 17 display both the model's overall predictions and the decomposed predictions for the non-stationary term, the stationary term, and the noise term. For clarity, we sample 100 predictions to represent the probabilistic learning outcomes, with the width of the prediction intervals indicating the model's quantified uncertainty.

Figure 15 shows the progressive widening of prediction intervals over time, a pattern that aligns with the inherent characteristics typically observed in financial data. A comparative analysis of these figures reveals an inverse correlation between prediction accuracy and the width of the prediction intervals: more precise point estimates are associated with narrower uncertainty bounds, which is consistent with our residual learning paradigm. These findings underscore FALDA's capacity to effectively model aleatoric uncertainty across diverse datasets, while simultaneously preserving high predictive accuracy. This dual capability highlights the model's strength in both uncertainty quantification and forecasting precision.

# I THE USE OF LARGE LANGUAGE MODELS (LLMs)

In the preparation of this manuscript, the large language models were employed solely to aid or polish writing. Specifically, LLMs were used for the purpose of correcting grammatical errors. All research ideas are entirely the work of the authors.

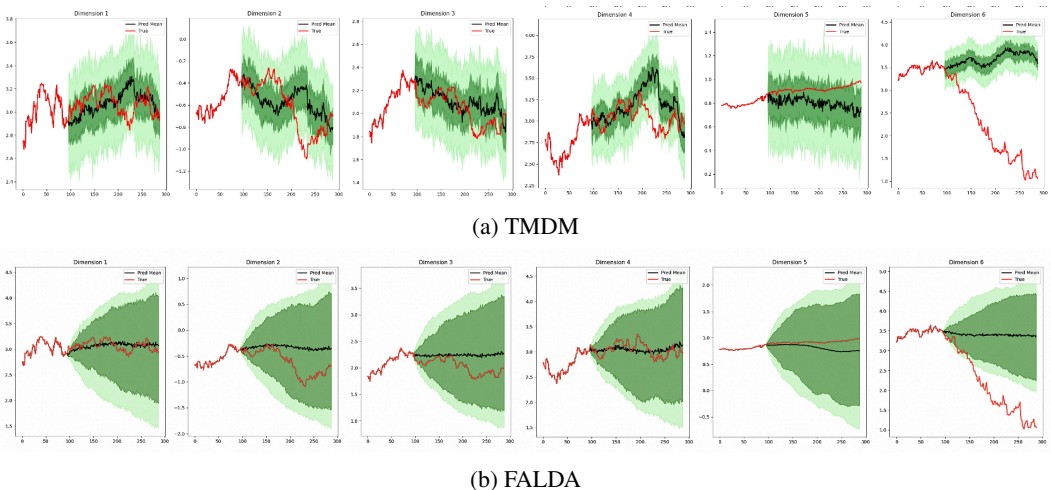

(a) TMDM

(b) FALDA

Figure 8: Comparison of prediction intervals for the Exchange dataset ($T = 96, S = 192$). The red line indicates the ground truth, and the black line represents the predicted mean. Dark green shading denotes the 50% prediction interval, and light green shading shows the 90% prediction interval.

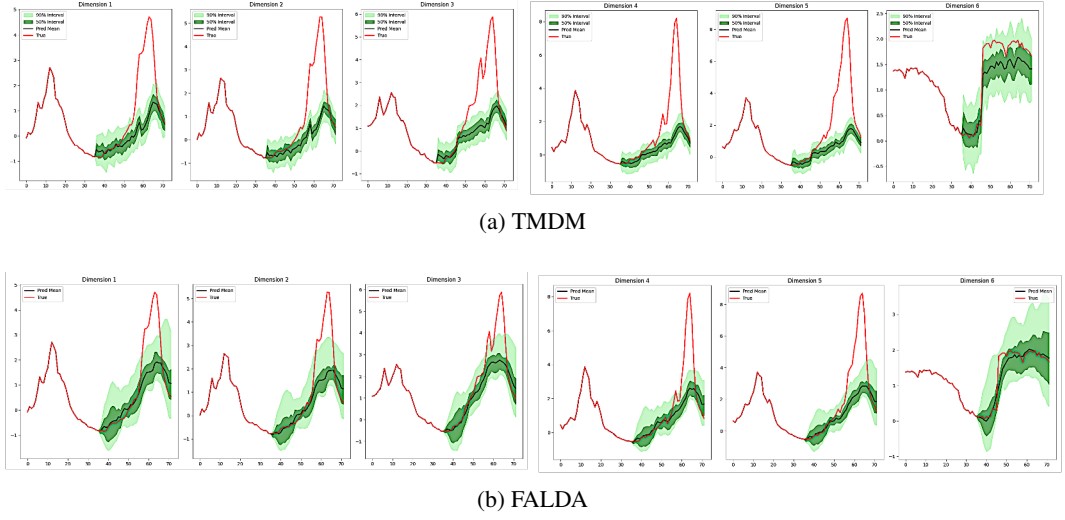

(a) TMDM

(b) FALDA

Figure 9: Comparison of prediction intervals for the ILI dataset ($T = 36, S = 36$). The red line indicates the ground truth, and the black line represents the predicted mean. Dark green shading denotes the 50% prediction interval, and light green shading shows the 90% prediction interval.

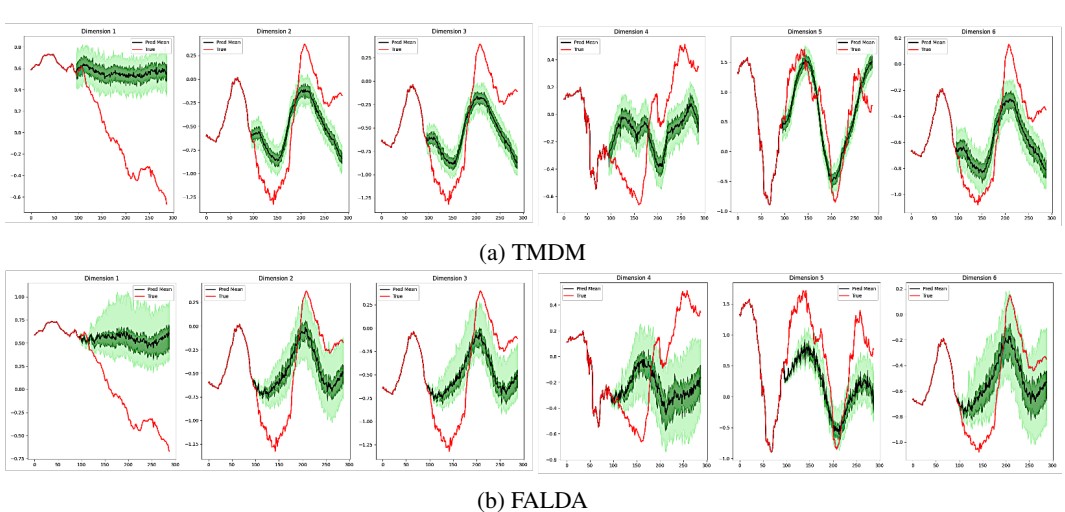

Figure 10: Comparison of prediction intervals for the Weather dataset ($T = 96, S = 192$). The red line indicates the ground truth, and the black line represents the predicted mean. Dark green shading denotes the 50% prediction interval, and light green shading shows the 90% prediction interval.

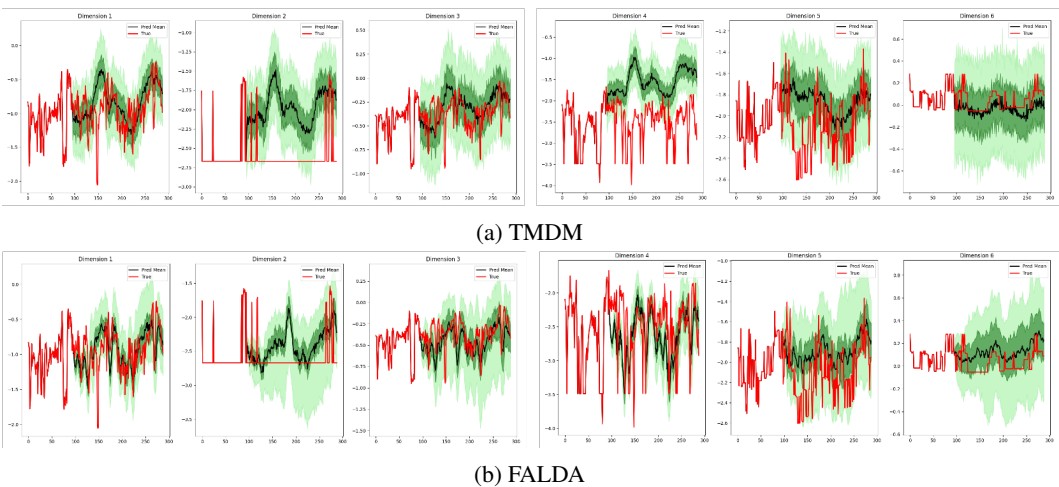

Figure 11: Comparison of prediction intervals for the ETTm2 dataset ($T = 96, S = 192$). The red line indicates the ground truth, and the black line represents the predicted mean. Dark green shading denotes the 50% prediction interval, and light green shading shows the 90% prediction interval.

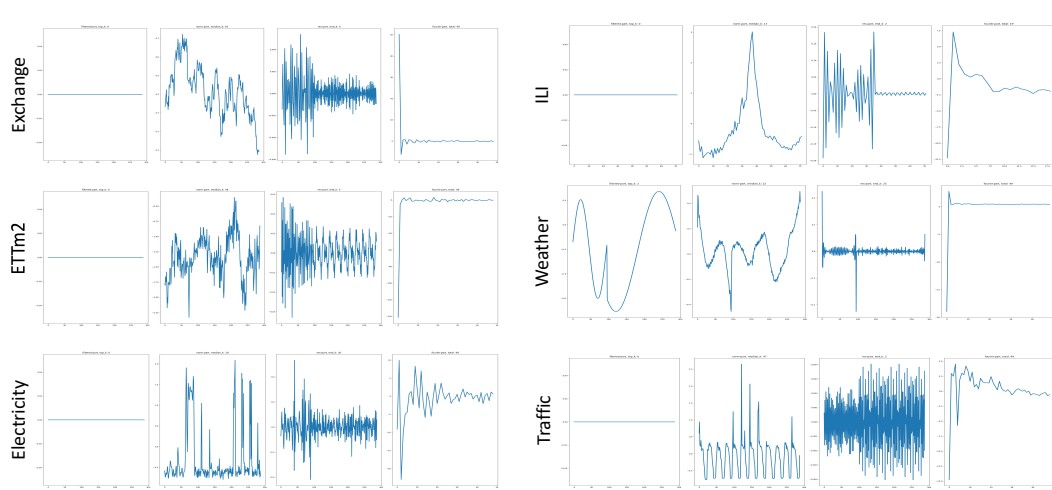

Figure 12: Time series decomposition strategy for the iTransformer backbone. From left to right, the subfigures present: (1) the non-stationary term, (2) the stationary term, (3) the noise term, and (4) the frequency-domain representation obtained via Fourier transform.

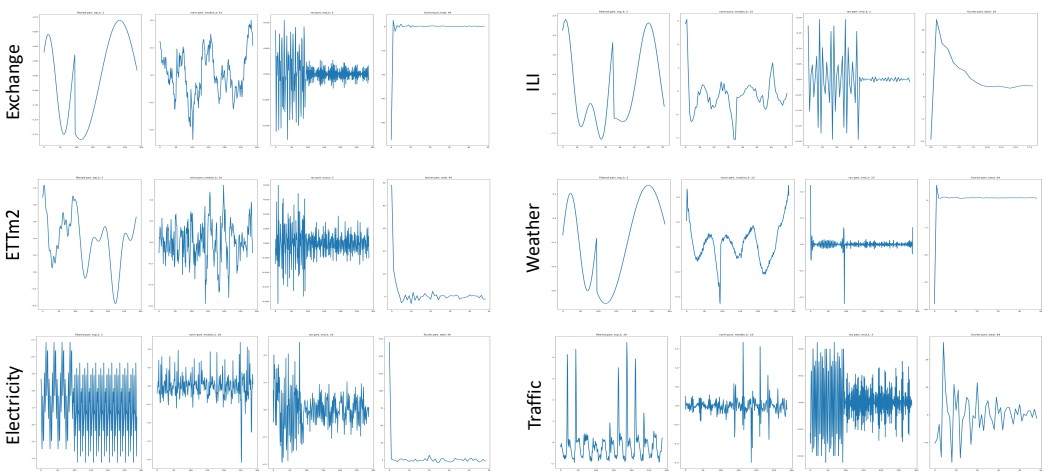

Figure 13: Time series decomposition strategy for other backbones. From left to right, the subfigures present: (1) the non-stationary term, (2) the stationary term, (3) the noise term, and (4) the frequency-domain representation obtained via the Fourier transform.

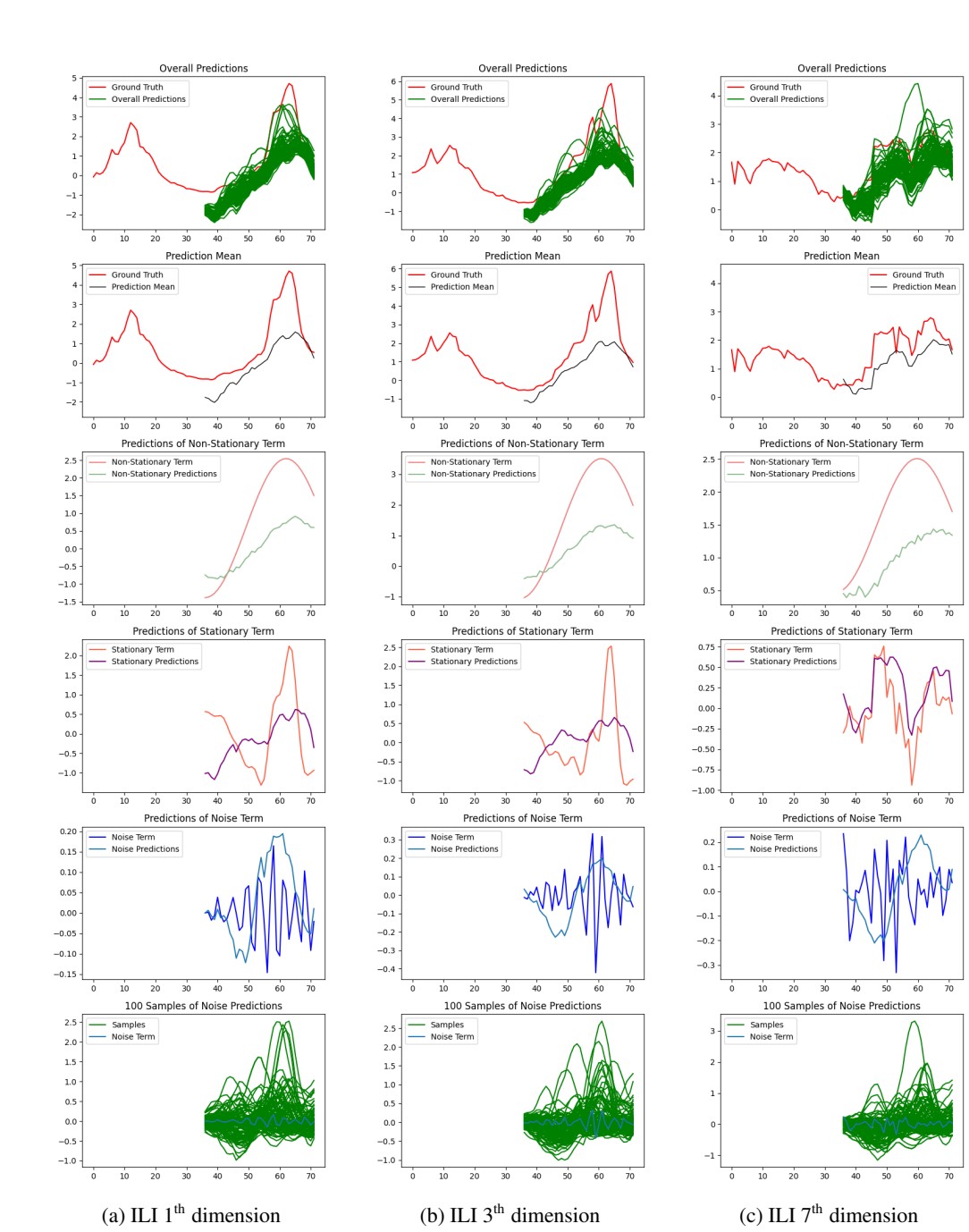

(a) ILI 1$^{th}$ dimension     (b) ILI 3$^{th}$ dimension     (c) ILI 7$^{th}$ dimension

Figure 14: Visualization of the prediction results from the different components (NS-Adapter, TS-Backbone, and DEMA) on the ILI dataset.

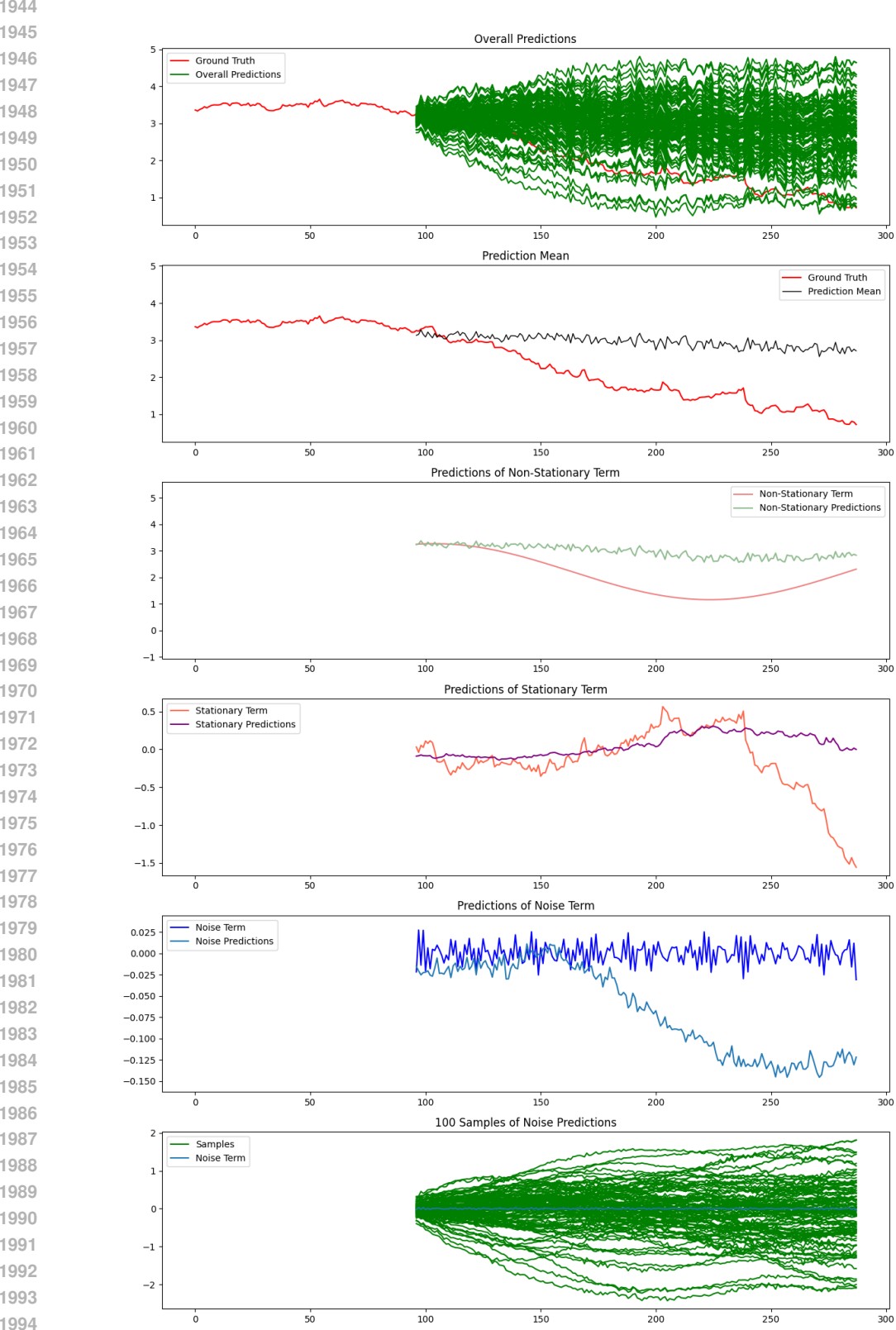

Figure 15: Visualization of the prediction results from the different components (NS-Adapter, TS-Backbone, and DEMA) on the Exchange dataset (6 [th] dimension).

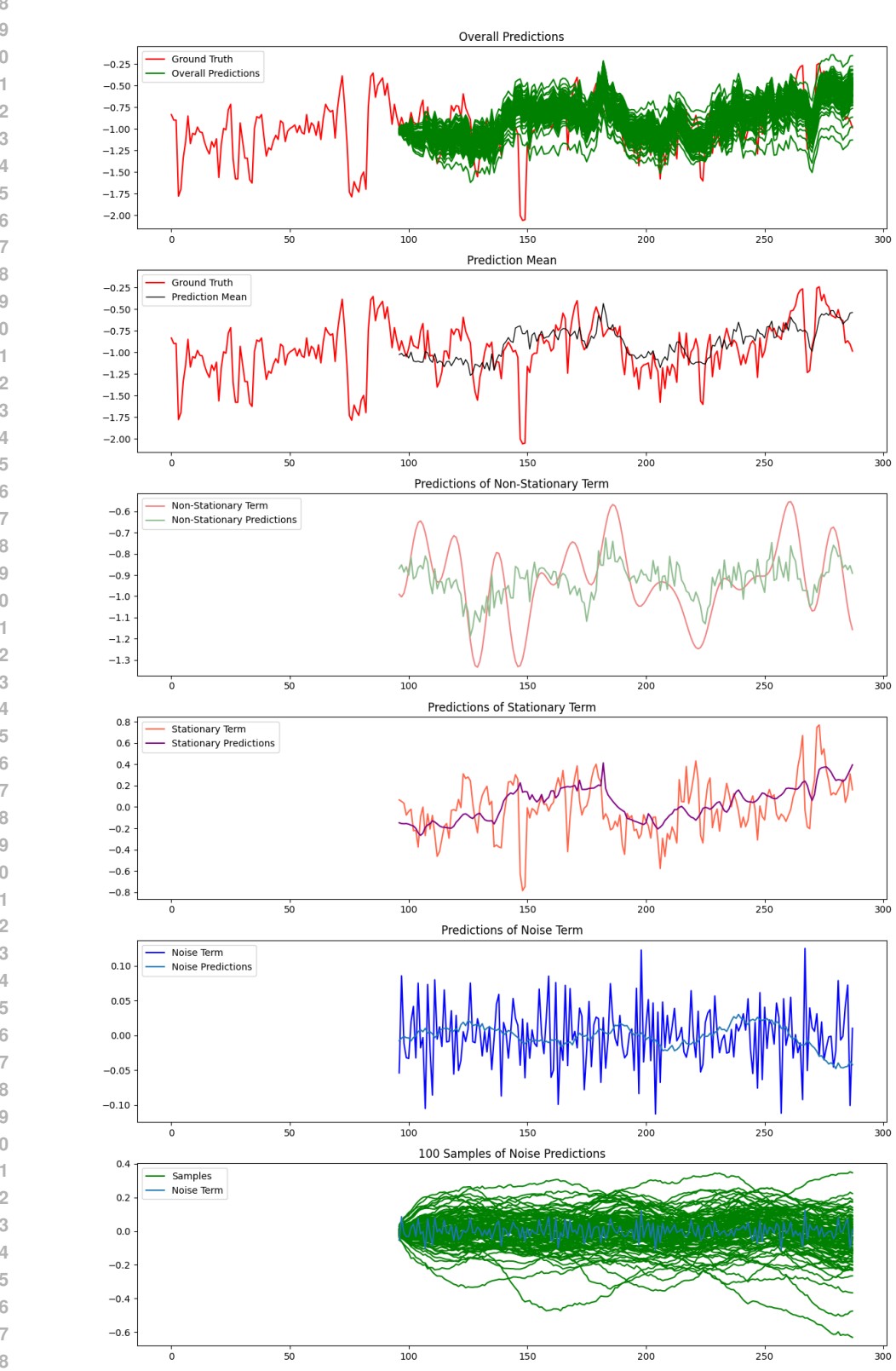

Figure 16: Visualization of the prediction results from the different components (NS-Adapter, TS-Backbone, and DEMA) on the ETTm2 dataset (1 th dimension).

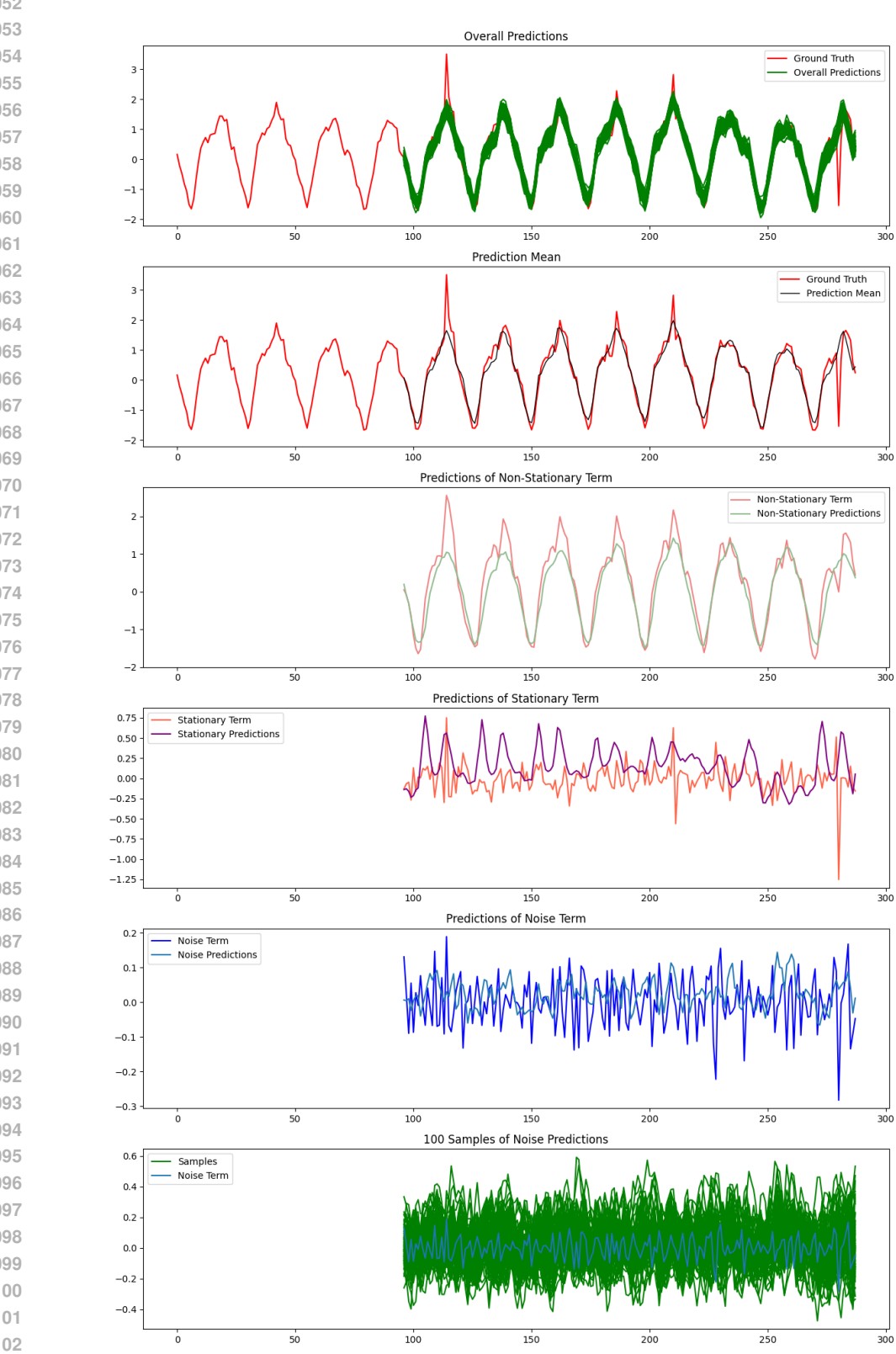

Figure 17: Visualization of the prediction results from the different components (NS-Adapter, TS-Backbone, and DEMA) on the Traffic dataset (800 th dimension).

