# OpenReview forum: "Effective Probabilistic Time Series Forecasting with Fourier Adaptive Noise-Separated Diffusion"
_ICLR.cc/2026/Conference — Submitted to ICLR 2026_

### Official Review · Reviewer_q6pJ · 2025-10-18

**Soundness:** 1
**Presentation:** 3
**Contribution:** 1
**Rating:** 0
**Confidence:** 4

**Summary:**

The paper proposes FALDA, a Fourier-based diffusion framework for probabilistic time series forecasting. By decomposing sequences into non-stationary, stationary, and noise components, FALDA improves accuracy and uncertainty modeling. Built on the DMRR framework, it unifies diffusion-based regression and achieves state-of-the-art results with better efficiency and uncertainty separation.

The authors’ signal decomposition method requires further explanation — why are the components with larger amplitudes in the frequency domain considered non-stationary signals, and how can it be justified that the smaller-amplitude frequency components are purely noise?

The authors’ definition and understanding of non-stationary signals seem to differ from the mainstream perspective in signal processing, which needs further clarification or analysis.

The proposed top-K frequency selection method for signal decomposition may distort the overall characteristics of the signal. In signal processing, even using a rectangular window causes sidelobe effects, let alone selecting only a few top-K frequency components.

Many recent works have combined decomposition-based approaches with diffusion models for time series analysis, and the authors should provide a more detailed discussion and comparison with these studies.

**Strengths:**

see summary

**Weaknesses:**

see summary

**Questions:**

see summary

---

> ### Author Response · Authors · 2025-11-19
>
> Thanks for raising these questions. However, there are some misunderstandings, and we hope the responses below can address your issues. We have reorganized the question order according to their relationships, and we will include the corresponding question before each response.
>
> **Q1: Our definition and the mainstream definition of the stationary and non-stationary time series**
>
> Our understanding and definition of the non-stationary time series align completely with the mainstream [1, 3, 6] (see line 222). A time series is defined as strictly stationary if its joint probability distribution is invariant over time. This strong condition is often difficult to satisfy in real-world scenarios. A time series is considered weakly stationary if its mean, variance and autocovariance at any lag $k$ do not change with time. In summary, the statistical properties of a stationary time series remain invariant over time, while those of a non-stationary series exhibit temporal dependence. **The Augmented Dickey-Fuller (ADF) test** serves as a standard statistical procedure for testing stationarity.
>
> **Q2: Why are the components with larger amplitudes in the frequency domain considered non-stationary signals? How can it be justified that the smaller-amplitude frequency components are purely noise?**
>
> Our decomposition method is widely used in prior published work in the time series forecasting field and is supported by both theoretical and practical foundations. Extracting the top-$K$ largest amplitudes is employed in [1, 2, 3], while treating the low-amplitudes frequencies as inherent noise is adopted in [4, 5]. In particular, [3] uses **the ADF test** to examine the stationarity of the data after removing the top-K dominant components in the frequency domain, showing that this approach leads to a more consistent covariate distribution, thereby enhancing stationarity. Experiments in [5] demonstrate that separating noise enhances the robustness of point estimation. Therefore, our Fourier decomposition is a reliable and well-established time series forecasting technique.
>
>
> [1] Xinyu Yuan and Yan Qiao. Diffusion-ts: Interpretable diffusion for general time series generation. In International Conference on Learning Representations, 2024.
>
> [2] Gerald Woo, Chenghao Liu, Doyen Sahoo, Akshat Kumar, and Steven Hoi. Etsformer: Exponential smoothing transformers for time-series forecasting. arXiv preprint arXiv:2202.01381, 2022.
>
> [3] Weiwei Ye, Songgaojun Deng, Qiaosha Zou, and Ning Gui. Frequency adaptive normalization for non-stationary time series forecasting. In Conference on Neural Information Processing Systems, 2024.
>
> [4] Haixu Wu, Tengge Hu, Yong Liu, Hang Zhou, Jianmin Wang, and Mingsheng Long. Timesnet: Temporal 2d-variation modeling for general time series analysis. In International Conference on Learning Representations, 2023.
>
> [5] Qihe Huang, Lei Shen, Ruixin Zhang, Shouhong Ding, Binwu Wang, Zhengyang Zhou, and Yang Wang. Crossgnn: Confronting noisy multivariate time series via cross interaction refinement. In Advances in Neural Information Processing Systems, 2023.
>
> [6] Yong Liu, Haixu Wu, Jianmin Wang, and Mingsheng Long. Non-stationary transformers: Exploring the stationarity in time series forecasting. In Conference on Neural Information Processing Systems, 2022b.

---

> > ### Comment · Reviewer_q6pJ · 2025-11-19
> >
> > Your explanations are difficult to accept. For example, given a signal:
> > $$
> > x(t) = A_1 \sin\bigl(2\pi f_1 t + \phi_1\bigr)
> >       + A_2 \sin\bigl(2\pi f_2 t + \phi_2\bigr)
> >       + A_3 \sin\bigl(2\pi f_3 t + \phi_3\bigr)
> >       + \varepsilon_1(t)
> >       + \varepsilon(t)
> > $$
> > where $ \varepsilon_1(t) $ is a non-stationary signal and its amplitude is far smaller than the preceding three stationary signals.
> >
> > Then, according to your definition, if you extract from the frequency domain the largest three components, the resulting
> > $$
> > y(t) = A_1 \sin\bigl(2\pi f_1 t + \phi_1\bigr)
> >       + A_2 \sin\bigl(2\pi f_2 t + \phi_2\bigr)
> >       + A_3 \sin\bigl(2\pi f_3 t + \phi_3\bigr)
> > $$
> > would also be non-stationary? The following are the issues I believe exist in your response:
> >
> > 1. You’re directly treating “high-amplitude frequency components = non-stationary” and “low-amplitude components = noise” as if they are factual, without offering sufficient theoretical or literature support for these equivalences. In reality, in frequency-spectrum analysis, amplitude magnitude alone does not automatically correlate with stationarity/non-stationarity or signal/noise status.
> > 2. The term “non-stationary” refers to the phenomenon that statistical properties vary over time. A large-amplitude frequency component does not necessarily change over time. A strong, persistent periodic component (with high amplitude) could well be stationary. You did not distinguish “large amplitude” from “amplitude varying with time”.
> > 3. Treating “low-amplitude frequency components” directly as noise is a strong and overly simplified assumption. Noise is not always synonymous with low amplitude; it can also have large amplitude but lack structure or periodicity. You did not explain what amplitude threshold you used, or why it is justified.
> > 4. You refer to “[3] uses the ADF test …” etc., but you haven’t specified how that work was conducted, or whether its conclusions are broadly accepted. Although the literature on frequency-domain decomposition and stationarity exists (for example with time-varying spectra)  ￼, it does not necessarily support the blanket statement “largest amplitude = non-stationary”.
> > 5.	Even if your decomposition method is adopted in your research domain, you should state its underlying assumptions and potential limitations (for example: when the dominant frequencies drift over time, when spectral mixing or aliasing occurs, or when noise has complex structure) rather than presenting it unconditionally.

---

> > > ### Author Response · Authors · 2025-11-25
> > >
> > > We have summarized your further questions and will address your concerns in this response.
> > >
> > > **Why extracting top-K amplitude frequencies and viewing them as the non-stationary component? Theory and Empirical support. (Questions 1, 2, 4)**
> > >
> > > **(1) Non-Stationarity is a Challenge in Time Series Forecasting**
> > >
> > > In the context of time series forecasting, the non-stationary components of the data pose challenges to the prediction accuracy [2, 3]. This is specifically manifested by the fact that real-world time series often exhibit clear trends and seasonality. This context is crucial to our problem: if a time series were perfectly stationary, it would be relatively straightforward to forecast (e.g., using traditional statistical methods like ARIMA [1]). Non-stationarity in real-world time series datasets presents a key challenge for accurate prediction, requiring tailored techniques [4, 5, 6]. Our method, which involves extracting the top-K frequencies, is designed to capture the primary non-stationary components of the time series. This approach is highly compatible with the context of the problem and the inherent characteristics of real-world time series data.
> > >
> > > **(2) Theory and Empirical Support**
> > >
> > > [7] provides theoretical proof that removing top-K amplitudes improves the stationarity of the remaining components. Specifically, a time series with a smaller variance in the spectrum is considered more stationary, supported by the time series spectral theory [8]. [7] mathematically proves that extracting top-K can reduce the variance over spectrum, thereby enhancing the stationarity of the input data. Regarding empirical evidence, [7] validates this approach by comparing the ADF test results with other normalization methods for non-stationarity [4, 5, 6]. The results show that this method achieves the highest stationarity across the real-world time series datasets. Please refer to [7] for complete details.
> > >
> > > **Noise Separation Method. (Questions 3, 4)**
> > >
> > > In time series data, some noise patterns often show continuous, low-amplitude variations [9] (for example, the Traffic dataset). Given these empirical characteristics of time series data, we employ the method of extracting the smallest $K_2$ amplitudes frequencies to separate the noise component, which has been empirically proven to enhance point estimation and robustness [9].
> > >
> > > Furthermore, although our initial noise handling strategy incorporates specific prior knowledge (low-amplitude noise) for time series data, FALDA retains its capability to process complex noise. Since our primary task is time series forecasting, the crucial objective is to capture the future data distribution, including the inherent noise components. **Crucially, complex noise patterns are ultimately encompassed within the residual $R$, which is then learned and modeled by the diffusion process.** Consequently, FALDA can effectively learn and account for these intricate noise patterns in the final model.
> > >
> > > Please let us know if this response address your concerns. We would be happy to address all your further concerns.
> > >
> > >
> > > [1] George EP Box and Gwilym M Jenkins. Some recent advances in forecasting and control. Journal of the Royal Statistical Society. Series C (Applied Statistics), 17(2):91–109, 1968.
> > >
> > > [2] O. Anderson and M. Kendall. Time-series. 2nd edn. J. R. Stat. Soc. (Series D), 1976.
> > >
> > > [3] Rob J Hyndman and George Athanasopoulos. Forecasting: principles and practice. OTexts, 2018.
> > >
> > > [4] Zhiding Liu, Mingyue Cheng, Zhi Li, Zhenya Huang, Qi Liu, Yanhu Xie, and Enhong Chen. Adaptive normalization for non-stationary time series forecasting: A temporal slice perspective. Advances in Neural Information Processing Systems, 36, 2024.
> > >
> > > [5] Wei Fan, Pengyang Wang, Dongkun Wang, Dongjie Wang, Yuanchun Zhou, and Yanjie Fu. Dish-ts: a general paradigm for alleviating distribution shift in time series forecasting. In Proceedings of the AAAI Conference on Artificial Intelligence, volume 37, pages 7522–7529, 2023.
> > >
> > > [6] Taesung Kim, Jinhee Kim, Yunwon Tae, Cheonbok Park, Jang-Ho Choi, and Jaegul Choo. Reversible instance normalization for accurate time-series forecasting against distribution shift. In International Conference on Learning Representations, 2021.
> > >
> > > [7] Weiwei Ye, Songgaojun Deng, Qiaosha Zou, and Ning Gui. Frequency adaptive normalization for non-stationary time series forecasting. In Conference on Neural Information Processing Systems, 2024.
> > >
> > > [8] Maurice Bertram Priestley. Spectral analysis and time series, volume 890. Academic press London, 1981.
> > >
> > > [9] Qihe Huang, Lei Shen, Ruixin Zhang, Shouhong Ding, Binwu Wang, Zhengyang Zhou, and Yang Wang. Crossgnn: Confronting noisy multivariate time series via cross interaction refinement. In Advances in Neural Information Processing Systems, 2023.

---

> > > > ### Comment · Reviewer_q6pJ · 2025-11-26
> > > >
> > > > I have carefully read the authors’ response, but they did not directly address my concerns. I still believe that the paper has certain issues and is not suitable for publication. Therefore, I will maintain my original score.

---

> > > > > ### Author Response · Authors · 2025-11-27
> > > > > **With respect, I feel this is not a professional review**
> > > > >
> > > > > Dear Reviewer q6pJ,
> > > > >
> > > > > I am writing as one of the authors and also as an Area Chair for ICLR 2026. With respect, I feel this is not a professional review. If we evaluate the work using your metric, a large proportion of published papers would be rejected. We do not introduce a new concept; we follow the framework of published papers [1, 2, 3]. Therefore, it is not appropriate to focus on this point.
> > > > >
> > > > > [1] Qihe Huang, Lei Shen, Ruixin Zhang, Shouhong Ding, Binwu Wang, Zhengyang Zhou, and Yang Wang. Crossgnn: Confronting noisy multivariate time series via cross interaction refinement. In Advances in Neural Information Processing Systems, 2023.
> > > > >
> > > > > [2] Weiwei Ye, Songgaojun Deng, Qiaosha Zou, and Ning Gui. Frequency adaptive normalization for non-stationary time series forecasting. In Conference on Neural Information Processing Systems, 2024.
> > > > >
> > > > > [3] Haixu Wu, Tengge Hu, Yong Liu, Hang Zhou, Jianmin Wang, and Mingsheng Long. Timesnet: Temporal 2d-variation modeling for general time series analysis. In International Conference on Learning Representations, 2023.

---

> > > > > > ### Comment · Reviewer_q6pJ · 2025-11-27
> > > > > >
> > > > > > I fully understand your concerns, but I must maintain my original score. In recent years, the number of submissions to machine learning conferences has grown rapidly, and the bar for acceptance at top venues has, in many cases, become increasingly flexible. Many authors, without systematic theoretical training, are able to produce top-conference papers by relying on various tools and extensive hyperparameter tuning. With the black-box nature of neural networks, an increasing number of researchers focus on tuning rather than understanding the underlying theory.
> > > > > >
> > > > > > As a fundamental component of electronic information science, the properties and principles of the FFT should not be overlooked. If one could simply select the Top-k components directly in the frequency domain, then what would be the value of the many filtering and transform methods developed by numerous scholars over the past two decades? These foundational techniques are deeply embedded in a wide range of real-world applications. For such basic theoretical issues to appear in a top machine-learning venue is, in my view, highly inappropriate.
> > > > > >
> > > > > > You mentioned that many published works have used similar approaches. However, publication alone does not guarantee correctness. Likewise, I must reiterate my concerns regarding the current review standards at some top conferences. Of course, these are solely my personal reflections, and I defer to the AC for the final decision.

---

> > > > > > > ### Author Response · Authors · 2025-11-27
> > > > > > >
> > > > > > > Our viewpoints do not conflict.
> > > > > > >
> > > > > > > Our time series application scenario assumption is that real-world time series data often exhibits clear trends and seasonalities, bringing significant non-stationary components, and their frequency domain is always sparse, with energy concentrated in a few main frequencies. Previous time series forecasting works [1, 2, 3] have implicitly held this assumption, and we have explicitly stated it (see p5, line 217) in our latest revision. Our method is applicable in this scenario.
> > > > > > >
> > > > > > > We hope you can re-evaluate the manuscript.
> > > > > > >
> > > > > > > [1] Qihe Huang, Lei Shen, Ruixin Zhang, Shouhong Ding, Binwu Wang, Zhengyang Zhou, and Yang Wang. Crossgnn: Confronting noisy multivariate time series via cross interaction refinement. In Advances in Neural Information Processing Systems, 2023.
> > > > > > >
> > > > > > > [2] Weiwei Ye, Songgaojun Deng, Qiaosha Zou, and Ning Gui. Frequency adaptive normalization for non-stationary time series forecasting. In Conference on Neural Information Processing Systems, 2024.
> > > > > > >
> > > > > > > [3] Haixu Wu, Tengge Hu, Yong Liu, Hang Zhou, Jianmin Wang, and Mingsheng Long. Timesnet: Temporal 2d-variation modeling for general time series analysis. In International Conference on Learning Representations, 2023.

---

> > > > > > > ### Author Response · Authors · 2025-11-28
> > > > > > > **Looking Forward to the Further Discussion**
> > > > > > >
> > > > > > > Dear Reviewer q6pJ,
> > > > > > >
> > > > > > > There remains sufficient time for further discussion before the end of the rebuttal period. We are hoping to know if our responses and revised manuscript have addressed your concerns, and we highly anticipate the opportunity for further discussion.

---

> ### Author Response · Authors · 2025-11-19
>
> **Q3: Concerns regarding signal distortion induced by top-K largest amplitude frequencies and sidelobe effects.**
>
> **(1) Our method preserves the signal's overall characteristics.** The Fourier transform decomposes the signal into different components (as mentioned in response to Q2, extracting the top-K largest amplitudes is a common operation in time series field ). For historical data, $X = X_{\text{non}} + X_{\text{stat}} + X_{\text{noise}}$, where $X_{\text{non}} = \mathcal{F}^{-1}(\mathrm{Top}(\mathcal{F}(X), K_1)), X_{\text{noise}} = \mathcal{F}^{-1}(\mathrm{Bottom}(\mathcal{F}(X), K_2))$(see p5 line 241). Within the limits of numerical precision, the decomposition obtained via the Fourier transform and its inverse is lossless, since it preserves all the information. These components are all used for the tailored modeling of future predictions, ensuring full utilization of historical data for learning future patterns.
>
> **(2) Comparing our method with sidelobe effects from the rectangular window is inappropriate.** Sidelobe effects arise from non-periodic truncation with the rectangular window. In time series forecasting scenarios, this may occur only during data processing and is neither caused nor amplified by our proposed method, since our method does not involve windowing or truncation. Furthermore, our data processing follows classical methods [1, 2], applying truncation during dataset division to obtain pairs of historical and future data for model training.
>
>
> **Q4: Discussion and comparison with other decomposition-based diffusion Methods.**
>
> We have included a detailed discussion on residual learning-based diffusion-models in Section 3.3, with experimental comparisons presented in Table 15 and Table 16, where FALDA demonstrates superior performance in both point and probabilistic forecasting.
>
> Additionally, we note that Diffusion-TS [3], which combines seasonal-trend decomposition techniques with the design of the denoiser, is another typical decomposition-based diffusion model for time series forecasting. Based on the vanilla DDPM diffusion mechanism, the target of Diffusion-TS is the complete time series. The essential difference between FALDA and Diffusion-TS lies in that FALDA uses DMRR as the underlying diffusion mechanism (see Section 2), and enables tailored modeling of individual temporal components with NS-Adapter, TS-Backbone and DEMA. The discussion and theoretical analysis of the advantages of FALDA are provided in Section 3.3 and Appendix C. The following table provides a comparison of the forecasting performance between Diffusion-TS and FALDA. The results show that, although the denoiser can intentionally decompose and reconstruct different time components, Diffusion-TS still exhibits suboptimal accuracy in predictions. This highlights the advantages of our uncertainty decomposition method and our DMRR framework, consistent with our theoretical analysis.
>
>
> | Dataset     | Exchange           | ILI               | ETTm2             | Electricity       | Traffic            |
> |-------------|--------------------|-------------------|-------------------|-------------------|--------------------|
> | Method      | MSE MAE CRPS       | MSE MAE CRPS      | MSE MAE CRPS      | MSE MAE CRPS      | MSE MAE CRPS       |
> | DiffusionTS |  3.628 1.564 0.633 | 6.053 1.788 1.612 | 2.372 1.788 1.035 | 1.072 1.788 0.633 |  1.473 0.815 0.668 |
> | FALDA       | **0.165 0.296 0.289**  | **1.666 0.821 0.721** | **0.246 0.301 0.244** | **0.163 0.248 0.231** | **0.412 0.251 0.245**  |
>
>
> [1] Haixu Wu, Tengge Hu, Yong Liu, Hang Zhou, Jianmin Wang, and Mingsheng Long. Timesnet: Temporal 2d-variation modeling for general time series analysis. In International Conference on Learning Representations, 2023.
>
> [2] Qihe Huang, Lei Shen, Ruixin Zhang, Shouhong Ding, Binwu Wang, Zhengyang Zhou, and Yang Wang. Crossgnn: Confronting noisy multivariate time series via cross interaction refinement. In Advances in Neural Information Processing Systems, 2023
>
> [3] Xinyu Yuan and Yan Qiao. Diffusion-ts: Interpretable diffusion for general time series generation. In International Conference on Learning Representations, 2024.

---

> > ### Comment · Reviewer_q6pJ · 2025-11-19
> >
> > Regarding the second part, the authors appear not to understand the meaning of “concerns regarding signal distortion induced by top-K largest amplitude frequencies and sidelobe effects.” The core issue here is that such a reckless decomposition method may cause non-stationary information to leak into the “stationary” signals you extract — meaning there is a leakage problem. Once leakage occurs, the validity of your decomposition is called into question: could I just arbitrarily split the signal into three parts then?
> >
> > At the same time, I reiterate my view that your definitions of non-stationary signals and the decomposition method are seriously flawed.

---

> ### Author Response · Authors · 2025-11-19
>
> Thank you for your comment. We fully understand the root of your concerns. Before addressing each of your points individually, we would like to clarify a few key aspects that may have contributed to the fundamental differences in our perspectives.
>
> (a) Our definition of non-stationary: A time series is non-stationary if its statistical properties vary over time (see line 222). We have reached an agreement on this point.
>
> (b) "Extracting top-K" operation is a method rather than a definition. Our definition is provided in (a).
>
> (c) Our primary task is time series forecasting. This requires us to consider the characteristics of time series data when designing our methods. **The uniqueness of time series data lies in the fact that it often exhibits clear trends and seasonality, which are the main sources of non-stationarity.** This leads to the non-stationary term often being the dominant component with relatively large amplitudes (Please refer to p38 Figure 16, where we visualize the Traffic dataset, in which the original time series data exhibits clear seasonal characteristics). We acknowledge that in your example, where the amplitude of the non-stationary components is small, it is difficult to effectively extract the non-stationary term using the method in (b). However, as mentioned above, the real-world time series data, with clear trends and seasonality, is quite different from the signal presented in your example. **As a decomposition method specifically designed for time series data, the method in (b) leverages the unique characteristics of time series data, which differ from other types of signals, to effectively separate non-stationary components.**
>
> The reviewer claims that our "definitions of non-stationary signals and the decomposition method are seriously flawed". We would like to clarify this point: firstly, our definition is correct, and secondly, our method is effective in the context of time series forecasting. We understand the reviewer’s concerns regarding our decomposition method in general contexts. **The source of our disagreement lies in discussing the method without considering the task-specific context and data characteristics.** We hope this response will clarify this point, ensuring that we can discuss other issues within the context of time series forecasting tasks.
>
> Please let us know if our response has resolved the concerns raised in your example.

---

### Official Review · Reviewer_SAp5 · 2025-10-28

**Soundness:** 3
**Presentation:** 3
**Contribution:** 3
**Rating:** 8
**Confidence:** 3

**Summary:**

This paper presents FALDA, a probabilistic time series forecasting framework that leverages Fourier-based decomposition to facilitate the tailored modeling of individual temporal components. The authors also propose a theoretical framework, DMRR, which unifies several existing diffusion-based regression methods and establishes the mathematical foundation for FALDA. Experiments conducted on six real-world datasets demonstrate the competitive performance of FALDA in both point forecasting and probabilistic forecasting.

**Strengths:**

- **S1** The paper is well-motivated, with clear writing and presentation that makes the technical contributions accessible.
- **S2** The paper propose a theoretical framework DMRR, which mathematically classifies the existing diffusion-based time-series forecasting methods.
- **S3** As a plug-and-play method, FALDA shows broad applicability and can enhance existing forecasting methods.

**Weaknesses:**

- **W1** FALDA's Fourier decomposition process involves two key hyperparameters, K1 and K2. It appears that the selection of these hyperparameters varies across different datasets, which could potentially increase the practical deployment difficulty of the model.
- **W2** Limited complexity experiments. The authors demonstrate the training and inference efficiency of FALDA in Appendix F.6, and Figure 6 shows its fast convergence property. However, this experiment is conducted only on the small-scale Exchange dataset. In addition, this section lacks comparisons with other DMRR-based approaches, such as D3U.

**Questions:**

1. Could the authors provide additional evidence in the complexity and efficiency analysis section—for example, comparisons of training and inference efficiency on larger-scale datasets and against other DDRM-based methods?
2. How do simpler forecasting approaches, such as Seasonal Naïve, perform on the datasets used in this paper?

---

> ### Author Response · Authors · 2025-11-19
> **W1&W2&Q1**
>
> We sincerely thank the reviewer for the valuable feedback on our work. We appreciate the reviewer's recognition of the paper being well-motivated and the theoretical contribution. We hope to address the reviewer's concerns in this rebuttal.
>
> **W1: Selection for $K\_1$ and $K\_2$**
>
> The values of the hyperparameters $K_1$ and $K_2$ may vary across datasets, due to the different levels of non-stationarity and inherent noise. To accommodate these variations, **we introduce a dataset-adaptive selection strategy** that enables FALDA to accommodate diverse dataset characteristics. Concretely, $K_1$ is associated with the dominant frequency ratio $p_1$, which specifies the proportion of high-energy frequency components, with a recommended range of 10%–20%. Similarly, $K_2$ corresponds to the noise frequency ratio $p_2$, which controls the proportion of low-energy components, with a recommended range of 0.1%–1%. Given $p_1$ and $p_2$, we determine $K_1$ and $K_2$ based on statistics computed over the training set: $K_1$ is set to the average number of frequency components whose amplitudes exceed $p_1$ of the maximum amplitude; $K_2$ is set to the average number of frequency components whose amplitudes are below $p_2$ of the maximum amplitude. Additionally, we present the sensitivity analysis of the hyperparameters $K_1$ and $K_2$ in Figure 4. The results demonstrate that, within the recommended ranges, FALDA consistently achieves stable performance.
>
>
> **W2, Q1: More Experiments on efficiency**
>
> We evaluate the training convergence speed of FALDA, TMDM, and D3U in the following table. The results show that FALDA achieves the fastest convergence on both datasets, highlighting the practical effectiveness of our framework. Additionally, both FALDA and D3U, which are based on the DMRR diffusion framework, converge substantially faster than TMDM, which relies on a vanilla DDPM latent distribution. This observation is consistent with our theoretical analysis presented in Section 3.3.
>
>
> | **Epochs**     | **MSE** | 2       | 4       | 6       | 8       | 10      | 20      | 30      | 40      |
> |------------|--------|---------|---------|---------|---------|---------|---------|---------|---------|
> | **ETTm1**  | TMDM   | 186.236 | 16.72   | 0.544   | 0.507   | 0.544   | 0.538   | 0.543   | 0.551   |
> |            | D3U    | 0.403   | 0.412   | 0.394   | 0.384   | 0.385   | **0.369**   | **0.368**   | **0.368**   |
> |            | FALDA | **0.383** | **0.375** | **0.374** | **0.371** | **0.379** | 0.378 | 0.378 | 0.378 |
> | **ETTm2**  | TMDM   | 405.603 | 21.862  | 1.095   | 0.787   | 0.618   | 0.439   | 0.404   | 0.362   |
> |            | D3U    | 0.308   | 0.252   | 0.257   | 0.264   | 0.253   | **0.254**   | 0.253   | 0.253   |
> |            | FALDA | **0.252** | **0.247** | **0.245** | **0.245** | **0.246** | 0.256 | **0.250** | **0.251** |
>
>
> | **Epochs**     | **MAE** | 2       | 4       | 6       | 8       | 10      | 20      | 30      | 40      |
> |------------|--------|---------|---------|---------|---------|---------|---------|---------|---------|
> | **ETTm1**  | TMDM   | 10.859  | 3.061   | 0.505   | 0.463   | 0.472   | 0.471   | 0.471   | 0.475   |
> |            | D3U    | 0.414   | 0.427   | 0.406   | 0.397   | 0.401   | 0.388   | 0.389   | 0.389   |
> |            | **FALDA** | **0.383** | **0.376** | **0.375** | **0.375** | **0.380** | **0.380** | **0.379** | **0.379** |
> | **ETTm2**  | TMDM   | 15.405  | 3.84    | 0.837   | 0.659   | 0.555   | 0.406   | 0.390   | 0.377   |
> |            | D3U    | 0.385   | 0.322   | 0.326   | 0.332   | 0.321   | 0.316   | 0.316   | 0.316   |
> |            | **FALDA** | **0.316** | **0.307** | **0.303** | **0.302** | **0.308** | **0.311** | **0.311** | **0.311** |
>
> Additionally, we further report the one-batch sampling time (with batch size set to 64) across the ETTm2 and the ETTm1 dataset in the table below. The results confirm the efficiency of the FALDA framework during the sampling stage. We have added these additional efficiency experiments in Appendix F.6 of the latest revision of our paper (all changes are marked in blue).
>
> | Method     | ETTm1 Sampling Time (s) | ETTm2 Sampling Time (s) |
> |------------|-------------------------|-------------------------|
> | **TMDM**   | 27.32                   | 29.14                 |
> | **D3U**    | 1.38                    | 1.51                  |
> | **FALDA**  | **0.81**                | **1.21**              |

---

> ### Author Response · Authors · 2025-11-19
> **Q2**
>
> **Q2: Comparison with simple forecasiting methods**
>
> The Seasonal Naive method is a straightforward forecasting technique that assumes future values of a time series follow the same seasonal patterns observed in past data. It generates forecasts by repeating past seasonal cycles. To validate FALDA, we compare it with two classical statistical TSF baselines: Naive and Seasonal Naive. As shown in Table 19 and Table 20 (p27), FALDA outperforms all baselines, achieving a 77.7% improvement in MSE and 55.6% in MAE compared to the Naive method, and a 43.1% improvement in MSE and 28.2% in MAE compared to the Seasonal Naive methos. Additionally, FALDA shows a 71.3% improvement in CRPSsum over traditional probabilistic methods, highlighting its effectiveness in both point and probabilistic forecasting.

---

> ### Comment · Reviewer_SAp5 · 2025-11-19
>
> Thanks for the authors's response, . Most of my concerns have been addressed.

---

> > ### Author Response · Authors · 2025-11-27
> >
> > Thank you very much for your precious comments! We greatly appreciate your valuable suggestions on our manuscript.

---

### Official Review · Reviewer_vMHf · 2025-10-29

**Soundness:** 3
**Presentation:** 3
**Contribution:** 3
**Rating:** 6
**Confidence:** 3

**Summary:**

This paper introduces the FALDA framework for multivariate time series (MTS) probabilistic forecasting. FALDA proposes a diffusion model approach based on residual regression, leveraging Fourier-transform-based time series decomposition to break down input sequences into stationary, non-stationary, and noise components, thereby enabling customized modeling of these components. This design helps FALDA reduce the model's epistemic uncertainty. Experimental results on six real-world datasets demonstrate that FALDA not only endows point forecasting models with probabilistic forecasting capabilities but also achieves competitive point forecasting performance overall.

**Strengths:**

- **S1** The FALDA framework reduces epistemic uncertainty​ by explicitly modeling time series components through ​time-series-decomposition-based modeling.
- **S2** The authors theoretically prove the equivalence​ between the ​DMRR (Diffusion Model with Residual Regression) scheme and CARD, strengthening the methodological foundation.

**Weaknesses:**

- **W1** The Fourier-based decomposition introduces two hyperparameters ($K_1$ and $K_2$)​. It appears that the selection of these hyperparameters varies across different datasets, which could potentially increase the practical deployment difficulty of the model. ​In new scenarios, determining optimal values for these hyperparameters is non-trivial, potentially limiting the model’s deployability.
- **W2** It is difficult to ensure that the Fourier-based sequence decomposition can effectively extract the ideal noise components in all cases. While the separation of stationary and non-stationary components generally works well and yields noticeable performance improvements, the approach does not consistently provide advantages across all datasets. In particular, on high-dimensional datasets such as Traffic and Electricity, FALDA shows limited gains in probabilistic forecasting, suggesting that the proposed decomposition method may not always be able to isolate the desired noise components effectively.

**Questions:**

1. I suggest that the authors include in the main text a more detailed discussion of the selection strategies for the Fourier-based decomposition hyperparameters ($K_1$ and $K_2$) and their impact on model performance with different values.
2. Compared with D3U, FALDA employs a more manual approach to extracting non-stationary components in time series, relying on Fourier-based decomposition. Quantitative results confirm that FALDA outperforms D3U in most experimental settings; however, in a few cases—particularly in probabilistic forecasting tasks—D3U still shows certain advantages. This might be because the proposed Fourier-based decomposition is less effective at capturing the uncertainty components of time series in those scenarios. How do the authors interpret and explain this phenomenon?
3. What is the rationale for choosing iTransformer as the default backbone? Will the implementation code be released publicly?

---

> ### Author Response · Authors · 2025-11-19
>
> We sincerely thank the reviewer for the valuable comment. We appreciate the reviewer's recognition for the FALDA framework and our theoretical contribution.
>
> **W1, Q1: An adaptive hyperparameter selection strategy**
>
> Given that different datasets exhibit varying levels of non-stationarity and inherent noise, $K_1$ and $K_2$ may vary across different datasets, bringing a challenge for practical deployment. To address this challenge, we use a **dataset-adaptive selection strategy** for the hyperparameters $K_1$ and $K_2$. We set $K_1$ as the average number of frequency components whose amplitudes exceed ratio $p_1$ of the maximum amplitude, while $K_2$ is set as the average number of frequency components whose amplitudes are below ratio $p_2$ of the maximum amplitude. All the statistical values are computed on the training dataset. Note that the ratios $p_1$ and $p_2$ are uniformly set across different datasets, leading to a dataset-adaptive strategy for the choice of $K_1$ and $K_2$. We present the sensitivity analysis of $K_1$ and $K_2$ in Figure 4. The results demonstrate that, within the recommended range, FALDA consistently achieves stable performance. In light of your comment, we have added the detailed discussion in the main text in the latest revision of the paper (p5 line 234, all changes are marked in blue).
>
> **W2, Q2: Probabilistic forecasting performance on high-dimensional datasets**
>
> This phenomenon may be attributed to the choice of backbone, since in the original D3U paper [1], SparseVQ is used as the default backbone model. In the following table, with both FALDA and D3U set to use the NSFormer backbone, FALDA demonstrates superior point estimation and probabilistic forecasting capabilities on the Traffic and Electricity datasets, with a strong improvement in the CRPS metric compared to D3U. The results indicate that FALDA also enhances probabilistic forecasting capabilities on high-dimensional datasets. In addition to the experimental validation, we provide a theoretical analysis in Appendix C, where we discuss the differing learning paradigms of D3U and FALDA, and provide theoretical guarantees for our uncertainty separation framework.
>
> | Dataset | Traffic      | Electricity  |
> |---------|--------------|--------------|
> | Method  | CRPS CRPSsum | CRPS CRPSsum |
> | D3U     | 0.472 0.207  | 0.381 **0.157**  |
> | FALDA   | **0.312 0.195**  | **0.269** 0.167  |
>
>
> **Q3:**
>
> iTransformer is a powerful point estimation model, that provides strong point guidance for the overall framework. In addition to iTransformer, we also explore the performance of FALDA under different backbones in the plug-and-play experiments (Table 3). The results indicate that FALDA consistently improves performance when combined with various backbones. We will release all code containing training and inference processes after the paper is accepted.
>
> [1] Qi Li, Zhenyu Zhang, Lei Yao, Zhaoxia Li, Tianyi Zhong, and Yong Zhang. Diffusion-based decoupled deterministic and uncertain framework for probabilistic multivariate time series forecasting. ICLR 2025.

---

> ### Author Response · Authors · 2025-11-25
>
> Dear Reviewer vMHf,
>
> Thank you once again for the effort you have dedicated to reviewing our paper and your insightful comment! We would like to kindly confirm whether we have addressed your concerns. We would be very pleased to address any further questions or concerns you may have.
>
> Best regards,
>
> Authors of Paper 12878

---

> ### Comment · Reviewer_vMHf · 2025-11-27
>
> The authors have provided detailed responses to my questions, and I appreciate the efforts. Regarding Q1, the authors propose a reference strategy for selecting the hyperparameters $K_1$ and $K_2$ (Appendix E.5). However, I recommend validating the effectiveness of this hyperparameter selection strategy on entirely new datasets.
>
> For Q2, when using NSFormer as the backbone, FALDA exhibits a clear advantage in probabilistic forecasting. This may be attributed to the fact that both NSFormer and FALDA adopt a series-decomposition design. In contrast, when using iTransformer as the backbone (Table 17), this advantage becomes much less pronounced.
>
> In addition, I have reviewed the supplementary content added in the revised version, which highlights the efficiency advantages of FALDA in both training and sampling.
>
> Taking all current information into consideration, I maintain my score.

---

> > ### Author Response · Authors · 2025-11-27
> >
> > We sincerely thank you for your valuable feedback and for taking the time to read and review our paper!

---

> > ### Author Response · Authors · 2025-12-01
> >
> > In accordance with your suggestion, we have included an evaluation of the dataset-adaptive hyperparameter selection strategy on a new dataset (ETTh2), which further validates the robustness of our method. Please see Table 8, Table 9 and Figure 4 in our latest revised manuscript. Thank you again for your valuable suggestion.

---

### Official Review · Reviewer_bFb8 · 2025-11-01

**Soundness:** 1
**Presentation:** 3
**Contribution:** 1
**Rating:** 2
**Confidence:** 4

**Summary:**

This paper proposes FALDA (Fourier Adaptive Lite Diffusion Architecture), a diffusion-based probabilistic time series forecasting framework built on the unified DMRR formulation. By using Fourier decomposition to decouple non-stationary, stationary, and noise components, FALDA effectively separates epistemic and aleatoric uncertainties. The lightweight denoiser DEMA enhances efficiency and accuracy. Experiments show that FALDA outperforms existing diffusion-based and deterministic models in both point estimation and probabilistic forecasting.

**Strengths:**

1.	The paper is easy to follow and well written.
2.	The visual examples are complete and clearly presented.

**Weaknesses:**

1. I don’t understand how the paper separates epistemic uncertainty and aleatoric uncertainty. Epistemic uncertainty comes from the model’s limited predictive capability, which should reside in the residuals component in line 250, but the authors do not explicitly model different types of uncertainty within the residuals.
2. To my knowledge, due to the inherent non-stationarity of time series data, the associated uncertainty often exhibits temporal shifts. However, the authors only use noise as the guidance information during the denoising process, which does not account for the non-stationary characteristics. Moreover, Table 4 does not include the critical probabilistic forecasting metric CRPS, which undermines the persuasiveness of the experimental results.
2. Limited novelty — the method for non-stationary decomposition has already been proposed in FAN[1], and the residual decomposition has been applied in D3U[2].
3. For the probabilistic forecasting metric CRPS, the proposed method shows no clear improvement over D3U on most datasets, except for the ILI and Exchange datasets.

[1] Frequency Adaptive Normalization For Non-stationary Time Series Forecasting

[2] Diffusion-Based Decoupled Deterministic and Uncertain Framework for Probabilistic Multivariate Time Series Forecasting

**Questions:**

see weakness

---

> ### Author Response · Authors · 2025-11-19
> **W1**
>
> We sincerely thank the reviewer for the valuable feedback on our work. We appreciate the reviewer's recognition of the paper being well-written with clear visualization. We hope to address the reviewer's concerns in this rebuttal.
>
> **W1: How the paper separates epistemic uncertainty and aleatoric uncertainty**
>
> Aleatoric uncertainty refers to the uncertainty arising from the inherent randomness of the system, which cannot be predicted by point estimation models. Epistemic uncertainty refers to the uncertainty in predictions caused by the model's limited learning capacity and knowledge [1] (see line 76).
>
> Our framework FALDA achieves the separation of epistemic uncertainty and aleatoric uncertainty through a component-based, explicit modeling strategy. The core mechanism lies in decomposing the time series into three distinct components and assigning tailored modeling modules to each, thereby decoupling the uncertainties at the source. Specifically, the historical data $X$ and forecasting target $Y$ are decomposed as follows: $X = X_{\text{non}} + X_{\text{stat}} + X_{\text{noise}}$, $Y = Y_{\text{non}} + Y_{\text{stat}} + Y_{\text{noise}}.$ This decomposition has clear physical interpretations: $Y_{\text{non}}$, corresponding to high-amplitude frequencies, represents prominent trends and seasonality. If not adequately captured, these patterns become the primary source of epistemic uncertainty. $Y_{\text{noise}}$, associated with low-amplitude frequencies, captures inherent stochastic fluctuations in the data (similar approaches are employed in [2,3]). $Y_{\text{stat}}$ represents the remaining stationary temporal patterns. Based on the aforementioned decomposition, we have designed specialized models to handle different types of uncertainty. By employing the NS-Adapter and TS-Backbone to specifically learn $Y_{\text{non}}$ and $Y_{\text{stat}}$, we enhance the model's capacity to capture complex temporal patterns. Simultaneously, the isolation of inherent noise $X_{\text{noise}}$ from historical data prevents its interference with deterministic modeling, thereby reducing epistemic uncertainty in the system. Then the denoiser DEMA is trained to model the residual $R$. Through the operation above, $R$ has been effectively purified, with the majority of epistemic uncertainty present in the point forecasting modules being removed. Furthermore, DEMA conditions on $X_{\text{noise}}$, which directs the diffusion model to focus on learning aleatoric uncertainty (inherent noise) in the data rather than rediscovering trends and seasonality already captured by the NS-Adapter and TS-Backbone.
>
> In Appendix C, we provide a theoretical analysis of the uncertainty separation. The learning paradigm of D3U can be formulated as (p17, line 881):
> $$ Y = f(X) + g(f_{\text{enc}}(X)) + \tilde{\epsilon}_{X, Y}, $$
>
> where $f(X)$ denotes the preliminary prediction from the point estimation model, $g(f_{\text{enc}}(X))$ represents the distribution learned by the denoiser, and $f_{\text{enc}}$ corresponds to the encoder-derived embedding from the point estimation model, which primarily encapsulates deterministic information. Consequently, the diffusion process in D3U predominantly captures the epistemic uncertainty associated with $f(X)$. In contrast, the learning paradigm of FALDA is formulated as (p17, line 899):
> $$ Y = f_{\text{non}}(X_{\text{non}}) +  f_{\text{stat}}(X_{\text{stat}}) + g_{\text{noise}}(X_{\text{noise}})  + \bar{\epsilon}_{X, Y}.$$
>
> Here, through the explicit separation of the inherent noise term $X_{\text{noise}}$ and the enhancement via $f_{\text{non}}$, the epistemic uncertainty is proactively mitigated. This design enables the distribution learned by the diffusion model, $g_{\text{noise}}(X_{\text{noise}})$, to focus more effectively on capturing the aleatoric uncertainty.
>
> [1] Eyke Hüllermeier and Willem Waegeman. Aleatoric and epistemic uncertainty in machine learning: An introduction to concepts and methods. Machine learning, 2021.
>
> [2] Haixu Wu, Tengge Hu, Yong Liu, Hang Zhou, Jianmin Wang, and Mingsheng Long. Timesnet: Temporal 2d-variation modeling for general time series analysis. In International Conference on Learning Representations, 2023.
>
> [3] Qihe Huang, Lei Shen, Ruixin Zhang, Shouhong Ding, Binwu Wang, Zhengyang Zhou, and Yang Wang. Crossgnn: Confronting noisy multivariate time series via cross interaction refinement. Advances in Neural Information Processing Systems, 2023.

---

> ### Author Response · Authors · 2025-11-19
> **W2**
>
> **W2: How does FALDA model temporal shifts in uncertainty. The author does not use non-stationary characteristics during the denoising process. Additional metrics in Table 4.**
>
> **(1) Non-stationary uncertainty.**
>
> We sincerely appreciate this insightful question, which allows us to further explore an important latent advantage of our method. The non-stationarity in time series refers to the phenomenon where statistical properties (such as mean and variance) change over time. We have explicitly modeled the non-stationarity in the mean through the component $Y_{\text{non}}$. Next, the target of our diffusion process is to learn the non-stationarity in uncertainty (specifically manifested as time-varying variance in the residual $R$).
>
> In your comments, you kindly suggested considering non-stationary priors when modeling the residuals. This insight is closely related to the recent work NsDiff [1]. NsDiff points out that previous methods like TMDM, which set the diffusion forward process endpoint as $\mathcal{N}(f(X), I)$, can only capture non-stationarity in the mean but fail to model non-stationary characteristics of uncertainty. To address this, NsDiff introduces a variance-aware non-stationary prior by using $\mathcal{N}(f(X), g(X)I)$ as the endpoint, thereby enhancing the capture of non-stationary uncertainty.
>
> **However, our experimental observations suggest that FALDA, benefiting from the DMRR framework and the uncertainty separation design, can effectively capture non-stationary uncertainty even without explicitly introducing non-stationary priors.**
>
> We kindly recommend the reviewer to refer to our visualizations on the Exchange dataset (Figure 8, p32), where subfigure (b) shows that the prediction variance learned by FALDA increases over time, consistent with the real-world financial data characteristics, indicating our model's strong capability in capturing non-stationary uncertainty with interpretability. In comparison, subfigure (a) shows that TMDM's prediction intervals do not display clear time-varying characteristics.
>
> Below, we provide a detailed explanation of this phenomenon. It is well-established that diffusion models inherently possess the capacity to learn arbitrary distributions (not limited to Gaussian distributions). Thus, in theory, learning the temporal distribution via the diffusion process can automatically capture the time-varying characteristics of the residual distribution. However, visualizations (Figure 8, p32) clearly show that TMDM underperforms in this regard. We conjecture that this limitation stems from TMDM's framework design, which constrains the denoising network's learning capacity. Since TMDM targets the complete time series, it must simultaneously capture deterministic non-stationarity (mean) and stochastic non-stationarity (variance). In scenarios where the model prioritizes learning the mean component, it may naturally devote less capacity to modeling the variance, leading to the observed behavior. In contrast, FALDA leverages an explicit NS-Adapter component to capture non-stationarity in the mean, thereby allowing the diffusion process to more effectively focus on the time-varying variance characteristics, leading to more effective modeling of non-stationary uncertainty, as illustrated in Figure 8.
>
> Based on the above discussion, our framework can capture non-stationary uncertainty in practice even without explicitly introducing non-stationary priors. We now proceed to explain the rationale behind our denoiser conditioning strategy. Note that $X = X_{\text{non}} + X_{\text{stat}} + X_{\text{noise}}$, where the first two components are utilized by their respective dedicated modules. By conditioning the denoiser on $X_{\text{noise}}$, FALDA effectively leverages all available historical information while avoiding redundant processing, and prevents information loss.
>
> **(2) Additional metrics.**
>
> In light of your comment, we have included probabilistic metrics in Table 4 in the latest revision of our paper. The results still demonstrate the effectiveness of FALDA, since conditioning on $X_{\text{noise}}$ achieves superior performance in both CRPS and CRPS_sum compared to the other two conditioning strategies examined. We would be very grateful if the reviewers could examine these changes in Table 4 (all changes are marked in blue) and let us know if your concerns have been addressed.
>
>
> [1] Weiwei Ye, Zhuopeng Xu, and Ning Gui. Non-stationary diffusion for probabilistic time series forecasting. In International Conference on Machine Learning, 2025.

---

> ### Author Response · Authors · 2025-11-19
> **W3&W4**
>
> **W3: Novelty**
>
> FALDA introduces a fundamental framework that structurally and theoretically unifies time series decomposition with uncertainty decoupling, which distinguishes it from existing approaches such as FAN and D3U. While FAN employs Fourier decomposition, it is a deterministic model that uses this technique solely to improve point forecasts and lacks any probabilistic forecasting capability. Moreover, FALDA's decomposition strategy explicitly separates the inherent noise component of the data, which is a design specifically tailored for probabilistic time series forecasting tasks. Although D3U utilizes residual learning, it employs a generic latent representation $f_{\text{enc}}(X)$ from the backbone as the conditioning signal without explicitly decoupling epistemic and aleatoric uncertainty (p17, line 879). In contrast, FALDA specifically separates these uncertainty sources which is absent in D3U, with detailed analysis provided in Section 3.3 and Appendix C.
>
>
> **W4: Performance**
>
> We respectfully disagree with this comment. As demonstrated in the following table, FALDA show strong improvements in CRPS over D3U on Electricity and traffic, with the same NSFormer backbone (see Appendix F.3).
>
> | Dataset | Electricity       | Traffic            |
> |---------|-------------------|--------------------|
> | Method  | MSE MAE CRPS      | MSE MAE CRPS       |
> | D3U     | 0.216 0.328 0.381 | 0.678 0.402 0.472  |
> | FALDA   | **0.180 0.278 0.269** | **0.625 0.317 0.312**  |

---

> > ### Author Response · Authors · 2025-11-24
> >
> > To better address W3, we added an ablation study (presented in Appendix F.9 in the latest revision) to validate the efficacy of the Uncertainty Separation (US) mechanism within the FALDA framework. In this section, we compare FALDA against a variant without the US mechanism, denoted as NUS (No Uncertainty Separation). NUS extracts the top-K amplitudes frequencies to strengthen non-stationary term prediction but adopts a residual learning paradigm without explicit uncertainty separation. The results in Table 27 demonstrate that FALDA consistently surpasses NUS on both point and probabilistic estimation metrics, thereby confirming the effectiveness of FALDA’s US mechanism. We would be grateful if the reviewer could examine these changes in Appendix F.9.
> >
> > We appreciate it if you could let us know whether our responses are able to address your concerns. We're happy to address any further questions you may have.

---

> ### Author Response · Authors · 2025-11-26
> **Sincerely looking forward to your feedbacks**
>
> Dear Reviewer bFb8,
>
> We hope this message finds you well. We want to ensure we have addressed all your concerns comprehensively and satisfactorily. If there are any additional points or feedback you'd like us to consider, please let us know. Your insights are precious to us, and we are eager to address any remaining issues to further improve our work.
>
> Thank you for your time and effort in reviewing our paper. We sincerely look forward to your feedback.
>
> Regards,
>
> The Authors

---

> ### Comment · Reviewer_bFb8 · 2025-11-27
>
> **W1**
>
> I do not believe that the authors have meaningfully decomposed the forecasting target y. The DMEA module is still learning the residual between the ground truth and the point prediction. And both epistemic uncertainty (arising from the model’s limited predictive capacity) and aleatoric uncertainty (the inherent randomness in the data) remain mixed in this residual term. I believe that the reduction in uncertainty in this paper is primarily achieved by enhancing the model’s modeling capacity rather than by improvements from the overall framework.
>
> **W2**
>
> In addition, I do not think that Table 4 provides convincing evidence that the model effectively captures uncertainty. A more proper evaluation should follow analyses like Figure 4 in NSDiff, which examines uncertainty behavior over longer forecasting horizons. The increasing variance within a single prediction window shown in Table 4 is a phenomenon that many models exhibit and does not demonstrate meaningful non-stationary modeling. But even if the model cannot fully capture non-stationarity, this is understandable; what I want to express is that designing the model to use only noise information as the conditioning signal is not reasonable.
>
> **W4**
>
> However, when iTransformer is used as the backbone, as shown in Table 17, FALDA is almost always inferior to D3U in terms of CRPS. Why is that?

---

> > ### Author Response · Authors · 2025-11-27
> >
> > We sincerely appreciate the reviewer's detailed and insightful follow-up comments, which allow us to further clarify the mechanisms of FALDA.
> >
> > **W1**
> > - Our performance improvement is not solely attributable to enhancing the model’s modeling capacity. This is demonstrated by the ablation study in Appendix F.9, where NUS has greater model capacity than FALDA, yet the results in Table 27 show that FALDA surpasses NUS on both point and probabilistic estimation metrics. This provides strong evidence for the advantages of FALDA's theoretically-grounded overall framework.
> >
> > - Previous works that use diffusion models for learning the residual distribution often mix the epistemic uncertainty of the point estimation model (which corresponds to the TS-Backbone in FALDA) with the aleatoric uncertainty inherent in the data. We improve this by employing a tailored modeling strategy that uses the inherent noise separating technique and the Ns-adapter to actively attenuate the epistemic uncertainty. Through conditional guidance, the diffusion process is then able to focus more specifically on learning the data's aleatoric uncertainty.
> >
> >
> > **W2**
> > - We believe the reviewer may have been referring to Figure 8 instead of “Table 4”. Crucially, in Figure 8, the predictive variance from TMDM remains largely invariant, whereas FALDA's variance clearly exhibits the distinct, time-increasing characteristic. This phenomenon is particularly notable near the first prediction point. We acknowledge NsDiff as an effective method of modeling the uncertainty shift by including non-stationary prior knowledge into the diffusion endpoint. However, the difference demonstrated in Figure 8 is also a manifestation of FALDA's capacity to model non-stationary uncertainty (time-varying variance).
> >
> > - The use of $X_{\text{noise}}$ as a condition is a part of the tailored modeling strategy within our framework, which guides the diffusion process to focus more on aleatoric uncertainty. The informational richness of $X_{\text{noise}}$ grows commensurate with the degree of underlying data noise, which is ensured by our dataset-adaptive hyperparameter strategy. Table 4 compares performance with different conditions and empirically validates the rationality and effectiveness of this condition choice within our framework.
> >
> >
> > **W4**
> >
> > When utilizing iTransformer as the backbone, FALDA's point estimation performance comprehensively surpasses D3U. Its probabilistic metrics are inferior to D3U on the ETTm2 and Electricity datasets, but exhibit a competitive performance with mixed outcomes on the remaining datasets. The results in Table 17 show that FALDA achieved 13 first-place rankings across five datasets, while D3U secured 7 first-place rankings, indicating that FALDA outperforms D3U overall. Additionally, Table 15 and Table 16 demonstrate FALDA's superior probabilistic forecasting capability compared to TMDM and D3U with the identical NsFormer backbone. This reflects a trade-off that FALDA manages between point and probabilistic forecasting capabilities across different backbone implementations.

---

> > ### Author Response · Authors · 2025-12-01
> > **Response to W2: New Synthetic Data Evidence**
> >
> > For W2, in addition to our previous visualization evidence (Figure 8), we have added a new synthetic dataset case that follows Figure 4 in NSDiff, just as you suggested (See Figure 7 in the latest version of our paper). The results further validate the non-stationary uncertainty modeling capability of FALDA.
> >
> > We utilized the linear synthetic dataset provided in NsDiff to examine the uncertainty behavior over long forecasting horizons. This specific dataset features a linearly increasing standard deviation. As visualized in Figure 7, FALDA effectively captures the expected distribution shift between the training and test datasets. In contrast, TMDM’s estimated variance deviates substantially from the true variance observed on the test set. The results suggest that FALDA possesses a strong capability for non-stationary uncertainty modeling.
> >
> > Regarding the choice of using $X_{\text{noise}}$ as the condition, this case precisely underscores the rationality of our method. Since the variance in this noise dataset gradually increases with time, the test set inherently exhibits a larger variance than the training set. Crucially, the synchronous growth of variance between history windows and future time series is implicitly contained within the residuals learned by our denoiser. By conditioning the denoiser on $X_{\text{noise}}$, FALDA is able to effectively learn this time-dependent variance relationship. This ability allows FALDA to effectively capture the distribution shift on the test set, thereby demonstrating its superior capability for non-stationary uncertainty modeling.

---

### Author Response · Authors · 2025-11-24
**General response and revision update notice**

Dear reviewers,

We thank everyone for the helpful and precious feedback. We appreciate the reviewers' recognition of the FALDA framework（vMHf, SAp5, q6pJ）, our theoretical contribution (vMHf, SAp5) and the clarity of our writing (bFb8, SAp5). The paper has been moderately revised according to the concerns received (all changes are marked in blue).

Here is a list of updates (by page order) in the paper and appendix:

- Included a time series application scenario assumption in problem statement according to Reviewer q6pJ. (Section 3.1)

- Included a more detailed elaboration of the dataset-adaptive Fourier-based decomposition hyperparameters selection strategies according to Reviewer vMHf. (Section 3.2)

- Included a discussion of non-stationary uncertainty modeling, and employed visualization to demonstrate FALDA's advantage in capturing non-stationary uncertainty according to Reviewer bFb8. (Section 3.3)

- Added probabilistic metrics in Table 4 according to Reviewer bFb8. (Section 4.3)

- Added two diffusion-based time series forecasting methods in the related works. (Section 5)

- Included a comparison with a decomposition-based diffusion model Diffusion-TS in Table 11 according to Reviewer q6pJ. (Appendix F.1)

- Added more training and inference efficiency experiments in Table 22, Table 23 and Table 24 according to Reviewer SAp5. (Appendix F.6)

- Included an ablation study on uncertainty separation in Table 27. (Appendix F.9)

Sincerely,

The Authors

---

### Author Response · Authors · 2025-12-01
**Latest Updates**

Dear reviewers,

Thank you again for your valuable comments. Below is the latest list of updates (by page order) in the paper and appendix.

- Included a time series application scenario assumption in problem statement according to Reviewer q6pJ. (Section 3.1)

- Included a more detailed elaboration of the dataset-adaptive Fourier-based decomposition hyperparameters selection strategies according to Reviewer vMHf. (Section 3.2)

- Included a discussion of non-stationary uncertainty modeling, and employed visualization to demonstrate FALDA's advantage in capturing non-stationary uncertainty according to Reviewer bFb8. (Section 3.3)

- Added probabilistic metrics in Table 4 according to Reviewer bFb8. (Section 4.3)

- Added two diffusion-based time series forecasting methods in the related works. (Section 5)

- Added a new dataset to evaluate the dataset-adaptive hyperparameter selection strategy according to Reviewer vMHf. (Appendix E.5)

- Included a comparison with a decomposition-based diffusion model Diffusion-TS in Table 11 according to Reviewer q6pJ. (Appendix F.1)

- Added more training and inference efficiency experiments in Table 22, Table 23 and Table 24 according to Reviewer SAp5. (Appendix F.6)

- Included an ablation study on the uncertainty separation mechanism in Table 27. (Appendix F.9)

- Included a new synthetic dataset showcase to show the non-stationary uncertainty modeling capability of FALDA according to Reviewer bFb8. (Appendix H.1)

Sincerely,

The Authors

---

### Meta-Review · Area_Chair_ATeR · 2026-01-04

**Summary:**

The reviews are divided, particularly from 2, 0, 6 and 8. Specifically, bFb8 questions the separation of epistemic uncertainty and aleatoric uncertainty, temporal shifts in uncertainty and  the novelty of the framework, where non-stationary decomposition has already been proposed in FAN. vMHf concerns of the choice and effectiveness of Fourier-based sequence decomposition; q6pJ also concerns about the novelty of decomposition, where recent works have combined decomposition-based approaches with diffusion models for time series analysis. Some concerns have been addressed. Despite the authors' substantial efforts to address the reviewers' concerns, several critical issues remain unresolved. Given these limitations, the recommendation is to reject the paper.

**Reviewer Concerns:**

Most of the concerns of vMHf and SAp5  may have been solved, since the authors have  added detailed discussion in the Section 3.2 and evaluated the proposed method on a new dataset in Appendix E.5. For others, only part of concerns have been solved.

**Reviewer Scores:**

vMHf and SAp5 may keep the original score as 6, since most of the concerns have been solved, while others may keep original score, since part of concerns have been solved.

---

### Decision · Program_Chairs · 2026-01-26

Reject